# A distinct monocyte transcriptional state links systemic immune dysregulation to pulmonary impairment in long COVID

Saumya Kumar [1,2], Chaofan Li [3,4], Liang Zhou[1,2], Qiuyao Zhan[1,2], Ahmed Alaswad[1,2], Sonja Volland [5], Bibiana Costa [1,2], Simon Alexander Krooss[1,2,6,7,8,9], Isabel Klefenz [10], Hagen Schmaus[1,2,11], Antonia Zeuzem[1,2,9,11], Dorothee von Witzendorff[1,2,11,12], Helena Lickei[1,2,9,11,12], Lea Pueschel [9], Anke R. M. Kraft [1,2,9,11,12], Markus Cornberg [1,2,9,11,12], Andreas Rembert Koczulla[7,13,14,15], Isabell Pink[16,17], Marius M. Hoeper [16,17], Cheng-Jian Xu[1,2,18], Susanne Häussler[12,19,20,21], Miriam Wiestler [9], Mihai G. Netea [18,22], Thomas Illig[5,17,23,24], Jie Sun [3,4,24] & Yang Li [1,2,12,17,18,23,24] ✉

The mechanisms driving immune dysregulation in long COVID disease remain elusive. Here we integrated single-cell multiome data, immunological profiling and functional assays to investigate immune alterations across multiple cohorts. A transcriptional state in circulating monocytes (LC-Mo) was enriched in individuals with mild–moderate acute infection and accompanied by persistent elevations of plasma CCL2, CXCL11 and TNF. LC-Mo showed TGFβ and WNT–β-catenin signaling and correlated with fatigue severity. Protein markers of LC-Mo were increased in individuals with pronounced fatigue or dyspnea, and those with severe respiratory symptoms showed higher LC-Mo expression. Epigenetically, LC-Mo exhibited AP-1- and NF-κB1-driven profibrotic programs. LC-Mo-like macrophages in bronchoalveolar lavage samples from individuals with severe respiratory symptoms displayed a profibrotic profile, and individuals with a high LC-Mo transcriptional state showed impaired interferon responses after stimulation. Collectively, our findings define a pathogenic monocyte transcriptional state linking systemic immune dysfunction to persistent long COVID disease, providing mechanistic insights and potential therapeutic targets.

Long COVID affects 10–20% of individuals after severe acute respiratory syndrome coronavirus 2 (SARS-CoV-2) infection, with symptoms ranging from mild discomfort to severe, long-lasting impairments such as fatigue, respiratory issues and neurological problems. These symptoms can persist for over 3 years (refs. 1–5), representing a substantial health burden and prompting efforts to better characterize long COVID (LC), including biomarker discovery for improved diagnosis[6–10].

LC presents with diverse symptoms reflecting multiorgan system abnormalities[11–13]. The evidence suggests multiple possible causes, including persistence of viral remnants or reactivation of latent viruses[7,14–17]. Yet, persistent immune dysregulation is a consistent finding in LC studies[10,11,14,16–19]. Although most LC cases follow mild-to-moderate acute illness, many studies do not stratify individuals by acute infection (AI) severity[6–8], which is crucial because severe cases, especially those treated in the intensive care unit, develop immune changes due to intensive medical interventions[20,21]. Failing to account for these differences may confound LC-associated molecular signatures, highlighting the importance of refined patient grouping.

To address this gap, we stratified individuals with LC by acute COVID-19 severity to better resolve immune heterogeneity and identify molecular features underlying chronic symptoms. We applied single-cell multiomics profiling of peripheral blood mononuclear cells (PBMCs) and measured plasma cytokines from individuals with LC with fatigue and respiratory symptoms using longitudinal and cross-sectional samples. We identified a distinct circulating CD14+ monocyte state associated with LC ('LC-Mo'), which was enriched in individuals with mild-to-moderate AI. This state coincided with persistent elevation of circulating cytokines, indicating systemic inflammation. In two independent cohorts of individuals with LC with severe respiratory symptoms and abnormal lung function, LC-Mo expression was increased in circulating CD14+ monocyte subsets. In bronchoalveolar lavage (BAL) myeloid cells from individuals with severe respiratory symptoms, LC-Mo-like macrophages showed a profibrotic gene expression profile. Functionally, CD14+ monocytes from individuals with LC-Mo enrichment showed dysregulated responses to ex vivo stimulation, indicating impaired immune regulation. Together, these findings provide systemic insight into the cellular and molecular basis of LC and highlight potential therapeutic targets.

## Results

### LC has a distinct transcriptome after mild or moderate disease

Individuals presenting with headache, dyspnea or fatigue to the pneumology outpatient clinic at Hannover Medical School (MHH) were recruited according to the German S1 guidelines[22] and the Delphi Consensus Criteria[21] for LC (4–12 weeks) and post-COVID-19 syndrome (>12 weeks). These criteria included symptoms persisting beyond the acute phase of SARS-CoV-2 infection or its treatment, new symptoms emerging after recovery and attributed to prior infection or worsening of pre-existing conditions. Because heterogeneity in LC molecular profiles may be shaped by acute disease severity and treatment, we stratified individuals with acute SARS-CoV-2 infection (AI) and LC into those with mild-to-moderate (WHO score of 1–5) AI (AI^M and LC^AM) and those with severe (WHO scores 6–9) AI (AI^S and LC^AS).

Cohort 1 included 45 individuals recruited between April 2020 and August 2021 at MHH, of which 9 gave longitudinal samples and 36 gave cross-sectional samples (n = 78 total samples), including 11 donors with AI categorized as AI^M (n = 7 donors, 42.8% women, median age = 52, range 23–66 years of age, WHO score range 1–5) and AI^S (n = 4 donors, 50% women, median age = 37, range 32–54, WHO score range 6–9), 37 donors with LC categorized as LC^AM (n = 29 donors, 8 longitudinal donors with two to three time points and 21 single-time-point donors, 58% women, median age = 49, range 31–84 years) and LC^AS (n = 8 donors, 3 with two to four time points, 5 single-time-point donors, 25% women, median age = 46, range 19–75) and 8 donors who had recovered after 4–8 months of LC (R^LC; 1 longitudinal donor with two time points and 7 single-time-point donors, 37.5% women, median age = 38, range 19–65), in addition to 6 prepandemic noninfected

control individuals (NI; 50% women, median age = 40, range 24–61). LC and R^LC samples were collected 1.7–10.2 months after infection. Cohort 2 included 117 LC^AM donors (24 donors with two to four time points, 93 single-time-point donors, 58.9% women, median age = 48, range 19–83) and 25 LC^AS donors (12 longitudinal donors, 13 single-time-point donors, 20% women, median age = 53, range 18–81), recruited between May 2020 and August 2021 at MHH, along with 33 prepandemic NI samples (48.4% women, median age = 40, range 25–65). Cohort 3 included only LC^AM donors (n = 8 donors, 62.5% women, median age = 45, range 21–63), all with respiratory postacute sequelae of SARS-CoV-2 infection (PASC) recruited between October and November 2023 at the Pulmonary Rehabilitation Clinic in Schönau am Königssee, Germany. Cohort 4 included LC^AM donors (n = 29, 58.6% women, median age = 49, range 33–72), LC^AS donors (n = 11 donors, 18% female, median age = 57, range 35–81), 8 donors recovered from AI (R^A) and 2 NI donors (60% women, median age = 41, range 29–67) recruited between August 2020 and June 2022 at MHH. Cohort 5 included LC donors (n = 9 donors, 44.4% women, median age = 64, range 62–83, including 5 with respiratory PASC) and NI donors (n = 2 donors, 50% women, median age = 77, range 73–77), recruited between October 2020 and November 2021 at Mayo Clinic, a previously published study[23] (Fig. 1a and Methods).

Clinical assessment included blood gas analysis, pulmonary function tests and standardized participant-reported outcome measures: the fatigue assessment scale (FAS), validated in chronic fatigue[24–26] and LC, and the modified medical Research Council (mMRC) dyspnea scale (0–4, where 0 indicates no breathlessness, 1 indicates breathlessness on exertion, 2 indicates breathlessness when hurrying or walking uphill, 3 indicates stopping for breath after ~100 m or a few minutes, and 4 indicates too breathless to leave the house or when dressing), along with quality-of-life metrics[27]. All clinical assessment data were systematically collected at each participant visit for cohorts 1–4 (Fig. 1b and Supplementary Tables 1–5).

To study molecular signatures of disease progression, we stratified samples in cohorts 1 and 2 by months since AI (months 1.5/1.7–2.9, 3–5.9, 6–8.9 and 9–11; Fig. 1c and Methods). For cohort 1, we generated single-nucleus RNA-sequencing (snRNA-seq) and single-nucleus assay for transposase-accessible chromatin with sequencing (snATAC-seq) data from 78 PBMC samples from NI, R^LC, AI^M, AI^S, LC^AM and LC^AS donors across all time points. In cohort 2 we measured the concentrations of 14 cytokines in plasma samples from LC^AM or LC^AS and NI donors across all time points. Validation was performed using single-cell RNA-sequencing (scRNA-seq; cohort 3), flow cytometry (cohort 4) and a published PBMC/BAL single-cell dataset[23] (cohort 5). All samples, except those from participants with AI, were PCR negative at collection. We used an integrative multistep analysis to identify cell-type-specific immune dysregulation and link and assess relevance in LC (Extended Data Fig. 1a).

Analysis of single-cell data from cohort 1 PBMCs yielded ~118,000 high-quality cells (Fig. 1d). snRNA-seq data showed distinct patterns

**Fig. 1 | Transcriptomes of circulating immune cells show heterogeneity in individuals with LC. a**, Schematic showing the distribution of samples across cohort 1, which included longitudinal and cross-sectional PBMC samples (n = 78) from NI donors (n = 6) and donors with AI (n = 11), LC^AM (WHO 1–5; n = 39), LC^AS (WHO 6–9; n = 13) and R^LC (n = 9), collected 1.7–10.2 months after infection; cohort 2, which included longitudinal and cross-sectional samples (n = 238) from NI donors (n = 33) and donors with LC^AM (n = 158) and LC^AS (n = 47) collected at 1.5–11 months after infection; cohort 3, which included PBMCs from LC^AM donors (n = 8) collected 8–42 months after infection; cohort 4, which included PBMC samples (n = 40) from R^A donors and NI donors (n = 10) and donors with LC^AM (n = 29) and LC^AS (n = 11) collected 3–14 months after infection; and cohort 5, which consisted of PBMC (n = 11) and BAL (n = 9) samples from individuals with LC with unknown acute-phase severity (LC^UN; n = 9) and NI donors (n = 2; GEO: GSE263817). **b**, Number of individuals with LC in cohorts 1–5 exceeding thresholds for fatigue (FAS > 21), respiratory symptoms (dyspnea > 0)

or cardiology symptoms (top) and number of samples with pulmonary function tests (PFT), bronchial dilation tests (BDT), blood gas analysis (BGA), electrocardiogram (ECG), FAS and mMRC scores and quality-of-life (QoL) assessments (bottom). Empty boxes denote missing data. **c**, Distribution of LC samples across months 1.7–2.9 (LC^AM, n = 10; LC^AS, n = 4), 3–5.9 (LC^AM, n = 11; LC^AS, n = 3), 6–8.9 (LC^AM, n = 10; LC^AS, n = 4) and 9–11 (LC^AM, n = 8; LC^AS, n = 2) in cohort 1 (top) and months 1.5–2.9 (LC^AM, n = 43; LC^AS, n = 8), 3–5.9 (LC^AM, n = 56; LC^AS, n = 19), 6–8.9 (LC^AM, n = 47; LC^AS, n = 13) and 9–11 (LC^AM, n = 12; LC^AS, n = 7) in cohort 2 (bottom). **d**, UMAP of snRNA-seq data from 78 PBMC samples from all donors and all time points in cohort 1, as in **a** and **c**. **e**, Expression of genes significant by two-sided Wilcoxon test (Benjamini–Hochberg method-adjusted P value of <0.05 and log₂ (fold change) > 0.8) in CD14+ monocytes, CD16+ monocytes, CD4+ T cells, CD8+ T cells, B cells and NK cells, with genes consistently upregulated across labeled LC^AM time points; HSPC, hematopoietic stem and progenitor cells; moDCs, monocyte-derived dendritic cells; pDCs, plasmacytoid dendritic cells.

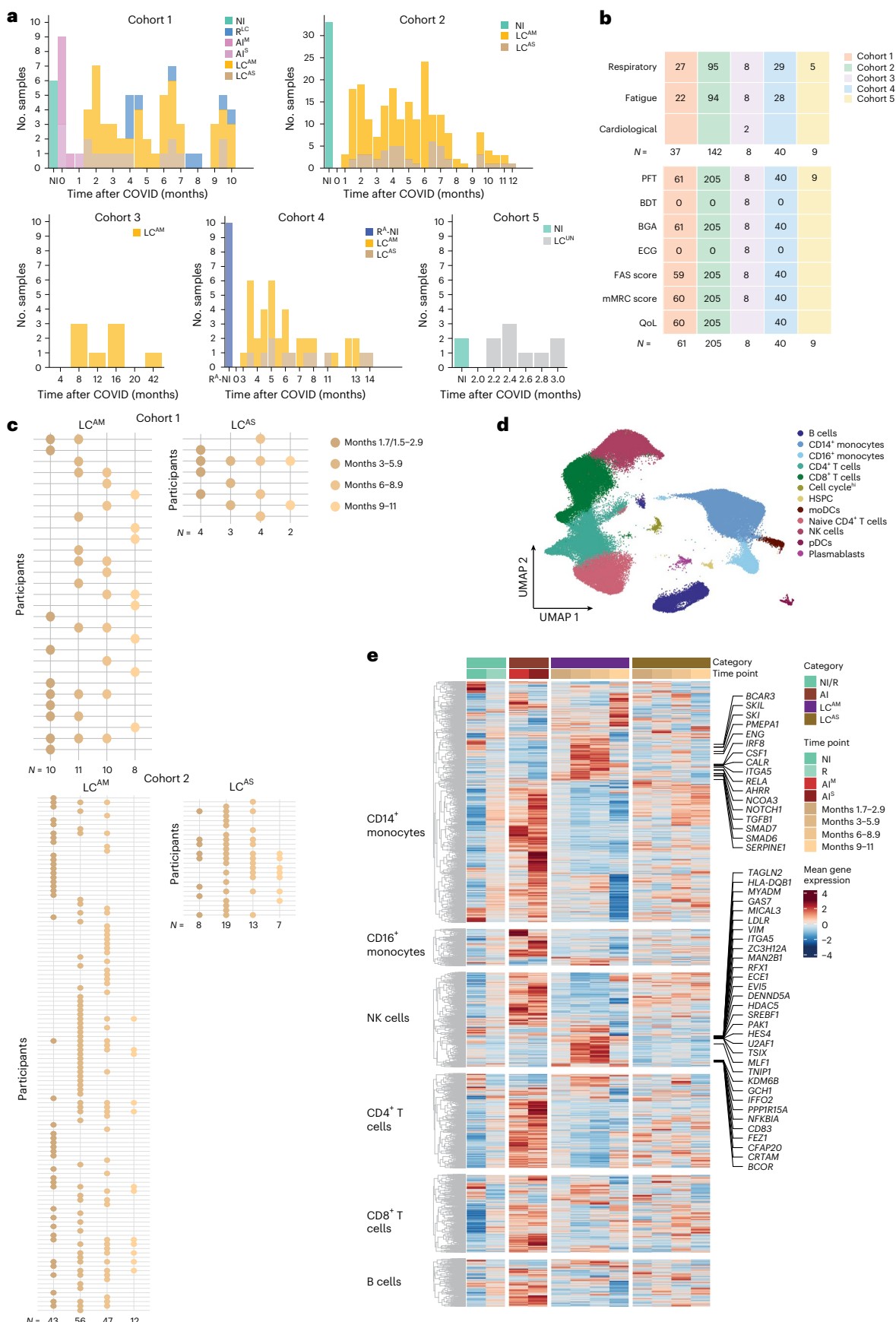

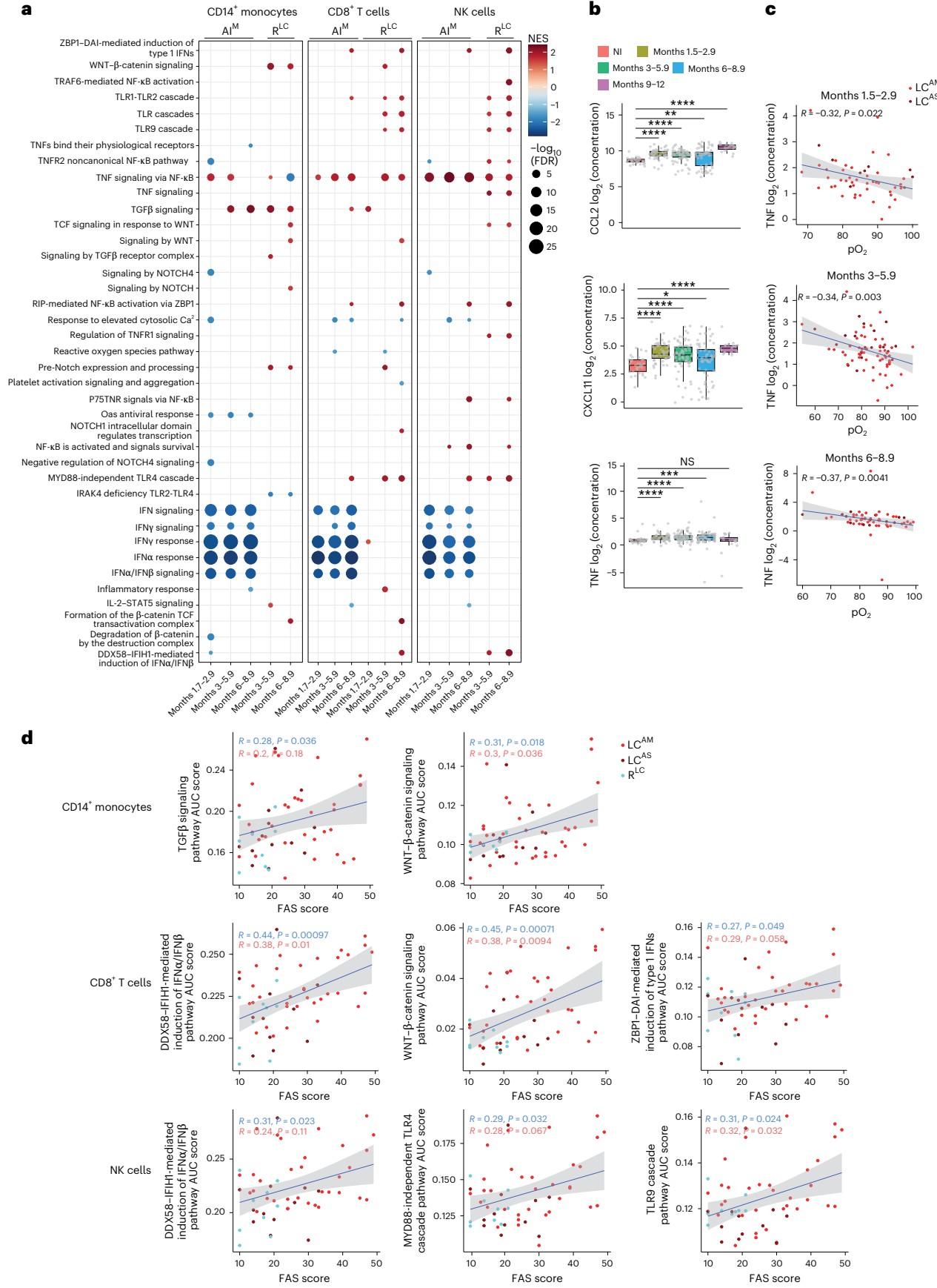

**Fig. 2 | TNF and inflammatory pathways in circulating immune cells indicate systemic inflammation in LC. a**, GSEA in CD14$^+$ monocytes, CD8$^+$ T cells and NK cells from LC$^{AM}$ samples compared to AI and R$^{LC}$ samples as in Fig. 1e. Pathways are plotted with an adjusted *P* of <0.1 (Kolmogorov–Smirnov-based test with permutation-derived *P* values, adjusted using the Benjamini–Hochberg method); NES, normalized enrichment score. **b**, Expression of CCL2, CXCL11 and TNF in the plasma of NI control individuals (*n* = 33) and individuals with LC at months 1.5–2.9 (*n* = 51), 3–5.9 (*n* = 75), 6–8.9 (*n* = 60) and 9–12 (*n* = 19) from cohort 2. Data were analyzed by two-sided Wilcoxon rank-sum test; ****P* < 0.00001, ***P* < 0.001, **P* < 0.01 and *P* < 0.05. The box plots show the median (center), first and third quartiles (bounds) and 1.5 times the interquartile range (whiskers). **c**, Correlation between the amount of TNF in the plasma and pO$_2$ levels in blood in LC donors at

months 1.5–2.9 (*n* = 51), 3–5.9 (*n* = 75), 6–8.9 (*n* = 60) and 9–12 (*n* = 19). Data were analyzed by Spearman correlation. *P* values were determined using the exact/permutation-based test. The gray shaded area indicates the 95% confidence interval. **d**, Correlation between AUC score and TGFβ and WNT–β catenin signaling pathways in CD14$^+$ monocytes, IFNα/IFNβ and TLR4 and TLR9 cascade signaling in NK cells and IFNα/IFNβ and WNT–β catenin signaling in CD8$^+$ T cells with a FAS score; red, statistics calculated using only LC samples (LC$^{AM}$ and LC$^{AS}$); blue, statistics calculated using samples from individuals with LC$^{AM}$ (*n* = 39), LC$^{AS}$ (*n* = 13) and R$^{LC}$ (*n* = 9). Data were analyzed by Spearman correlation. *P* values were determined using the exact/permutation-based test. The gray shaded area indicates the 95% confidence interval; NS, not significant.

in LC$^{AM}$ and LC$^{AS}$ compared to R$^{LC}$ and AI across major PBMCs (Fig. 1e). LC$^{AM}$ showed downregulated AI genes by months 6–8.9, whereas LC$^{AS}$ retained an acute COVID-19-like transcriptomic profile, indicating heterogeneity based on AI history (Fig. 1e). Differential gene expression (DGE) analysis identified 1,737 upregulated genes in CD14$^+$ monocytes from LC$^{AM}$ donors compared to those from AI and R$^{LC}$ (Fig. 1e), with upregulation over 1.7–8.9 months, and showed participant-specific heterogeneity (Fig. 1e and Extended Data Fig. 1b). LC$^{AM}$ CD14$^+$ monocytes showed persistent upregulation of proinflammatory (*CSF1*, *IRF8*, *RELA* and *NOTCH1*) and anti-inflammatory (*TGFB1*, *SMADs*, *ENG* and *SERPINE1*) markers (Extended Data Fig. 1c) at all time points, whereas other signature genes showed increased expression from 3 to 8.9 months (Fig. 1e). This signature diminished during months 9–11, possibly due to lower cell numbers (Extended Data Fig. 1d), but showed upregulation of a subset of acute-phase genes, including *IL1B*, *S100A4*, *PDIA3* and *MTRNR2L1*. LC$^{AM}$ natural killer (NK) cells also showed distinct increased expression of *SREBF1*, *TAGLN2*, *TNIP1*, *NFKBIA* and *CD83* among others compared to R$^{LC}$ and AI NK cells (Fig. 1e). Collectively, transcriptional profiles in individuals with LC reflected differences based on AI severity, with notable molecular changes in LC$^{AM}$ monocytes and NK cells, whereas LC$^{AS}$ displayed persistent but milder expression of acute-phase genes.

## TNF and TNF signaling genes are upregulated in LC$^{AM}$

We next performed gene set enrichment analysis (GSEA) using pseudo-bulk counts for each cell subset in LC$^{AM}$ or LC$^{AS}$ samples across all time points, comparing them to the AI and R$^{LC}$ cell samples. LC$^{AM}$ showed persistent upregulation of the TNF signaling pathway and persistent downregulation of interferon (IFN) signaling and response pathways across all major cell subsets (CD4$^+$ and CD8$^+$ T cells, B cells and CD14$^+$ and CD16$^+$ monocytes) compared to AI, up to month 8.9 (Fig. 2a and Extended Data Fig. 2a). CD8$^+$ T cells and NK cells from LC$^{AM}$ samples exhibited increased activation of the 'TLR signaling cascades' pathway

relative to R$^{LC}$ samples at months 3–8.9 (Fig. 2a). In LC$^{AM}$ CD14$^+$ monocytes, the TNF signaling pathway was transiently upregulated at months 1.7–5.9 and downregulated at months 6–8.9, whereas pathways including transforming growth factor-β (TGFβ), WNT–β-catenin and Notch signaling were upregulated at months 3–8.9 compared to in AI and R$^{LC}$ CD14$^+$ monocytes (Fig. 2a). In LC$^{AS}$, the TNF signaling pathway was sparsely activated in CD14$^+$ monocytes and CD8$^+$ T cells up to 5.9 months (Extended Data Fig. 2b). LC$^{AS}$ CD14$^+$ monocytes upregulated PD-1 signaling and MHC class II antigen presentation pathways compared to AI, but not R$^{LC}$ (Extended Data Fig. 2b, top). CD8$^+$ and CD4$^+$ T cells and NK cells from LC$^{AS}$ samples displayed increased activation of IFN response pathways compared to CD8$^+$ and CD4$^+$ T cells and NK cells from R$^{LC}$ samples (Extended Data Fig. 2b).

We also profiled 14 proinflammatory cytokines in cohort 2 plasma using a multiplex bead-based assay (Extended Data Fig. 2c), excluding interleukin-4 (IL-4) and IL-5 due to low detection. CXCL11, CCL2 and TNF were persistently elevated in individuals with LC compared to in NI donors up to month 9 (Fig. 2b). *TNF* mRNA was also persistently upregulated in individuals with LC$^{AM}$ across most immune cell types and time points (Extended Data Fig. 2d). TNF protein exhibited a statistically significant negative correlation with arterial oxygenation (pO$_2$) in individuals with LC (Fig. 2c), which remained statistically significant in LC$^{AM}$, but not in LC$^{AS}$, up to month 8.9 (Extended Data Fig. 3a). No other cytokines showed consistent correlations across all time points (Extended Data Fig. 3b,c).

Correlation analysis between key pathways upregulated in CD8$^+$ T cells, NK cells and CD14$^+$ monocytes and FAS scores indicated that TGFβ and WNT–β-catenin signaling in CD14$^+$ monocytes showed modest positive correlations with FAS scores in LC alone and stronger correlations when LC and R$^{LC}$ were combined (Fig. 2d). IFNα/IFNβ induction pathways positively correlated with FAS scores in CD8$^+$ T cells and NK cells in both LC only or LC + R$^{LC}$ combined analyses (Fig. 2d). WNT–β-catenin signaling in CD8$^+$ T cells and Toll-like receptor (TLR)

**Fig. 3 | Distinct cell subclusters drive LC signatures in NK cells, CD8$^+$ T cells and CD14$^+$ monocytes. a**, UMAP of CD8$^+$ T cells (left) and violin plots of AUC scores of TNF and TLR1–TLR2 pathways (right) within the identified subclusters *CD226$^+$* CD8$^+$ T cells (C0), *S100A4$^+$* CD8$^+$ T cells (C1), *CD69$^+$GZMK$^+$* CD8$^+$ T cells (C2), *CD69$^{hi}$GZMK$^+$* CD8$^+$ T cells (C3) and *KLRC2$^+$KLRD1$^+$* CD8$^+$ T cells (C4) from all donors and all time points in cohort 1, as in Fig. 1a. Data were analyzed by two-sided Wilcoxon rank-sum test; ****P* < 0.00001. **b**, UMAP of NK cells (left) and violin plots of AUC scores of TNF and TLR1–TLR2 pathways (right) in identified subclusters *PRF1$^+$GZMB$^+$* NK cells (C0), *GZMB$^+$KLRF1$^+$* NK cells (C1), *GZMK$^+$TGFB1$^+$* NK cells (C2), *IFNG$^+$* NK cells (C3) and *CALR$^+$S100A9$^+$* NK cells (C4) from all donors at all time points as in Fig. 1a. Data were analyzed by two-sided Wilcoxon rank-sum test; ****P* < 0.00001. **c**, UMAP of CD14$^+$ monocytes from all cohort 1 donors at all time points showing subclusters *IL1B$^+$* (MC1), *S100A4$^+$* (MC2), *FCN1$^+$* (MC3) and *TGFB1$^+$* (MC4) cells. **d**, Top significantly upregulated markers in MC1–MC4 CD14$^+$ monocyte subclusters as in **c**. Plotted genes were significant with a Benjamini–Hochberg method-adjusted *P* value of <0.05 (two-sided Wilcoxon test). **e**, Differential enrichment of neighborhoods representing transcriptional states in LC$^{AM}$ compared to AI$^M$ (top) and LC$^{AM}$ compared to R$^{LC}$ (bottom) at

months 1.7–2.9, 3–5.9, 6–8.9 and 9–11. Each dot represents a neighborhood of ~150–400 cells. Transcriptional states show significant enrichment with a spatial false discovery rate (FDR) of <0.1 (*F*-test statistic from the quasilikelihood *F*-test, graph-weighted FDR). **f**, AUC scores of TNF, TGFβ and WNT–β-catenin signaling pathways in MC1–MC4 CD14$^+$ monocyte subclusters as in **c**. Data were analyzed by two-sided Wilcoxon rank-sum test; ****P* < 0.00001. Horizontal dashed lines in **a**, **b** and **f** serve as visual reference for comparison of relative shifts in pathway AUC scores across clusters. **g**, Correlation of the percentage of MC1, MC2, MC3 or MC4 CD14$^+$ monocyte subclusters with FAS score and pO$_2$ (LC$^{AM}$, *n* = 38; LC$^{AS}$, *n* = 13; R$^{LC}$, *n* = 9). Data were analyzed by Spearman correlation. *P* values were determined using the exact/permutation-based test. The gray shaded area indicates the 95% confidence interval. **h**, Box plot showing FAS score in MC4$^{hi}$ (>10% of CD14$^+$ monocytes found within MC4, *n* = 13), MC4$^{lo}$ (<10% of CD14$^+$ monocytes within MC4, *n* = 26) and R$^{LC}$ (*n* = 7) samples from individuals with LC$^{AM}$, LC$^{AS}$ and R$^{LC}$. Data were analyzed by two-sided Wilcoxon rank-sum test; ***P* < 0.01. The box plot shows the median (center), first and third quartiles (bounds) and 1.5 times the interquartile range (whiskers).

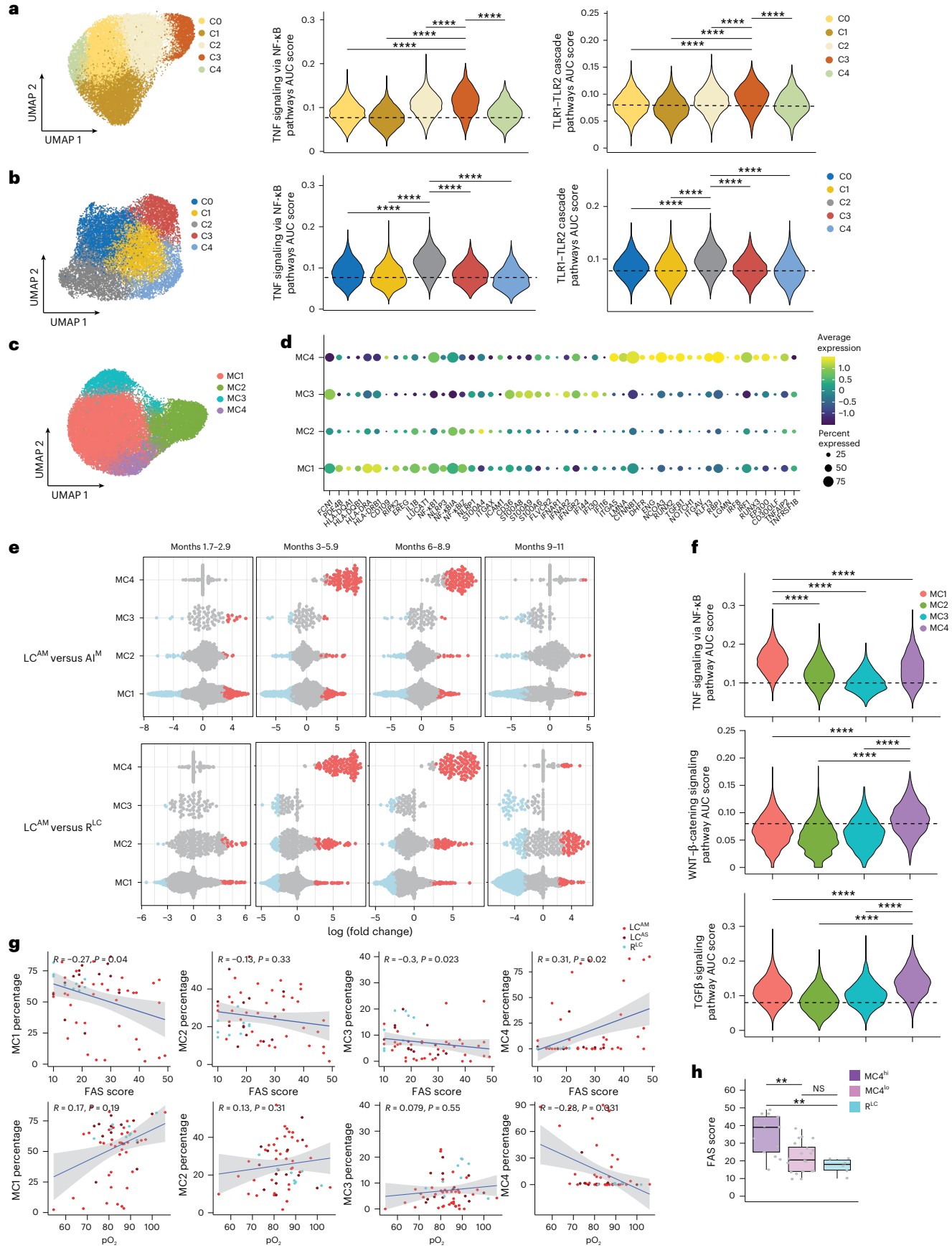

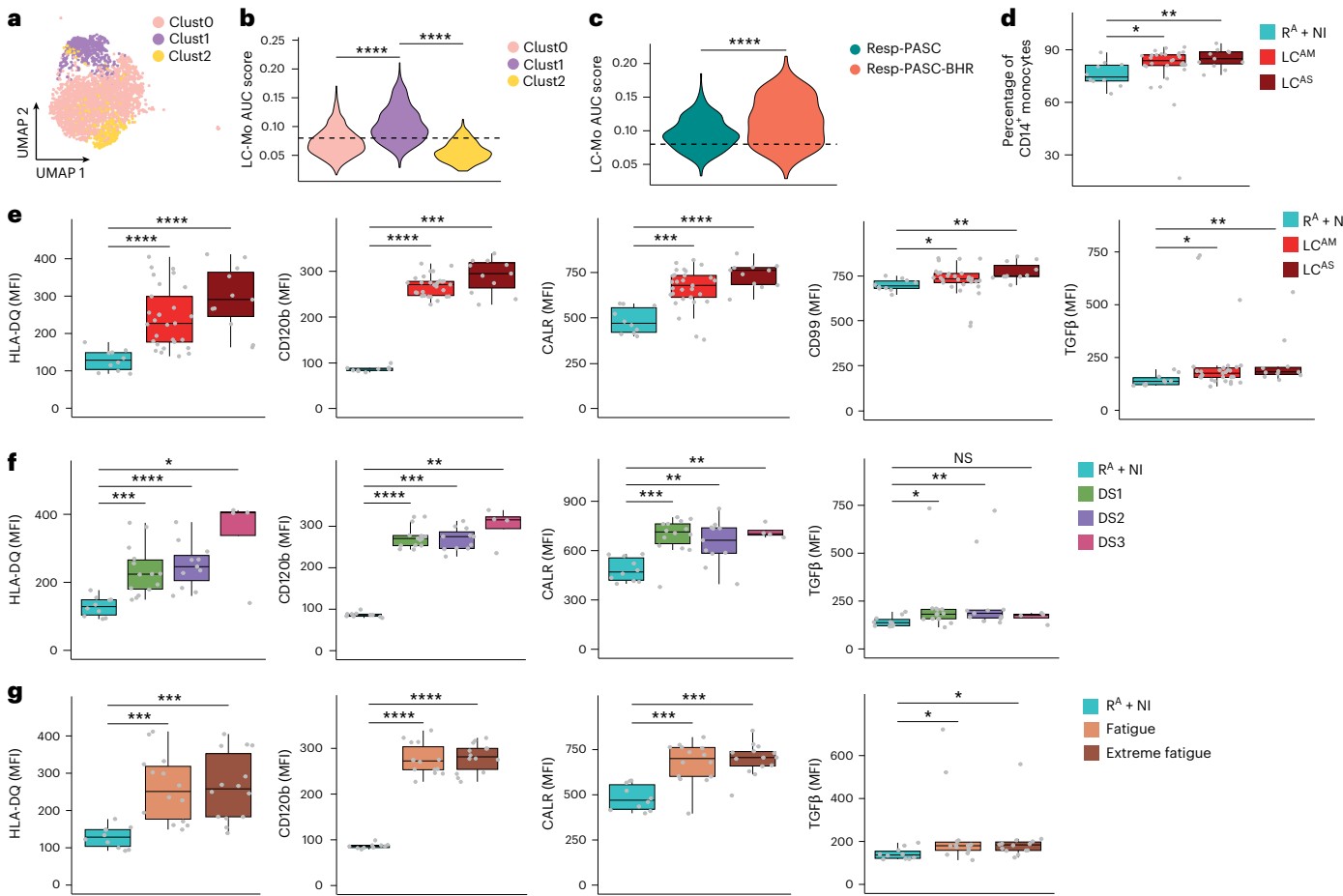

**Fig. 4 | LC-Mo-specific proteins show elevated expression in LC CD14⁺ monocytes. a**, UMAP of CD14⁺ monocytes from individuals with LC^AM (*n* = 8) from cohort 3 showing *S100A8⁺S100A9⁺* CD14⁺ monocyte (Clust0), *CTNNB1⁺EMP1⁺* CD14⁺ monocyte (Clust1) and *FCN1⁺CCL3⁺* CD14⁺ monocyte (Clust2) subclusters. **b**, LC-Mo signature AUC scores within *S100A8⁺S100A9⁺* CD14⁺ monocytes (Clust0), *CTNNB1⁺EMP1⁺* CD14⁺ monocytes (Clust1) and *FCN1⁺CCL3⁺* CD14⁺ monocytes (Clust2) as in **a**. Data were analyzed by two-sided Wilcoxon rank-sum test; \*\*\*\**P* < 0.00001. **c**, LC-Mo AUC scores in Clust1 CD14⁺ monocytes from individuals with LC^AM in cohort 3 with Resp-PASC (*n* = 5) or Resp-PASC-BHR (*n* = 3). Data were analyzed by two-sided Wilcoxon rank-sum test; \*\*\*\**P* < 0.00001. Horizontal dashed lines in **b** and **c** serve as visual reference for comparison of relative shifts in pathway AUC scores across clusters. **d**, Percentage of CD14⁺

monocytes among PBMCs in donors recovered from acute COVID-19 (R^A) combined with NI (R^A + NI, *n* = 10), donors with LC^AM (*n* = 29) and donors with LC^AS (*n* = 11) in cohort 4. **e**, MFI of HLA-DQ, CD120b, CALR, CD99 and TGFβ in samples from individuals with LC^AS, LC^AM and R^A + NI as in **d**. **f,g**, MFI of HLA-DQ, CD120b, CALR and TGFβ in R^A + NI (*n* = 10) and LC donors in cohort 4 categorized based on mMRC dyspnea score (DS) as DS1 (*n* = 14), DS2 (*n* = 11) and DS3 (*n* = 4) (**f**) or in R^A + NI (*n* = 10) and LC donors in cohort 4 with fatigue (*n* = 14, FAS score 22–34) and extreme fatigue (*n* = 14, FAS score 35–47) based on FAS category (**g**). *P* values in **d**–**g** were calculated using a two-sided Wilcoxon rank-sum test; \*\*\*\**P* < 0.00001, \*\*\**P* < 0.001, \*\**P* < 0.01 and \**P* < 0.05. Box plots show the median (center), first and third quartiles (bounds) and 1.5 times the interquartile range (whiskers).

signaling cascades in NK cells, but not TNF signaling in these cells, also correlated with FAS scores (Fig. 2d). These results indicate that persistent upregulation of inflammatory pathways and cytokines in LC^AM immune cells might contribute to the clinical symptoms in LC.

### The LC^AM monocyte signature characterizes a transcriptional state

Next, we performed a reclustering analysis of CD8⁺ T cells, NK cells and CD14⁺ monocytes from all donor samples. CD8⁺ T cells and NK cells each resolved into five clusters (Fig. 3a,b). Differential neighborhood abundance analysis (DA) comparing LC groups at each time point to AI and R^LC was performed. A neighborhood defines a small local group of cells with similar gene expression profiles, representing transitional states. LC^AM samples exhibited statistically significant increased abundance of neighborhoods in *CD69*^hi*CD27*^hi CD8⁺ T cells (C3), *GZMB⁺KLRF1⁺* NK cells (C1) and *CD69⁺TGFB1⁺* NK cells (C2; Extended Data Fig. 4a,b), whereas LC^AS samples showed increased abundance of neighborhoods in C2 NK cells at months 6–8.9 (Extended Data Fig. 4c,d). C3 CD8⁺ T cells and C2 NK cells showed

*GZMK⁺GZMB*^lo signatures (Extended Data Fig. 4e,f), reported to accumulate after SARS-CoV-2 infection and in aging[28–30]. These clusters showed higher expression of TNF and TLR signaling genes (Fig. 3a,b), suggesting the contribution of persistent TNF signaling in the expansion of *CD69*^hi*CD27*^hi*GZMK⁺* CD8⁺ T cells and *CD69⁺TGFB1⁺GZMK⁺* NK cells in individuals with LC^AM.

Within CD14⁺ monocytes, four primary clusters (MC1–MC4) were identified (Fig. 3c). MC1 showed high expression of MHC class II molecules, *IL1B* and *NFKB1*; MC2 showed elevated *NFKB1* and *S100A4*; MC3 showed increased expression of *FCN1*, IFN-stimulated genes (*IFI44*, *IFI16* and *IFI30*) and alarmins *S100A8* and *S100A9*; and MC4 displayed higher levels of *IRF1*, *IRF8*, *TGFB1*, *CTNNB1*, *ENG* and *NOTCH1*, among others (Fig. 3d). DA comparing LC samples with AI and R^LC samples across all time points showed a consistent significant increase in MC4 neighborhoods in LC^AM in both men and women (Fig. 3e and Extended Data Fig. 4g), with this becoming prominent from month 3 onward (Fig. 3e and Extended Data Fig. 5a). By contrast, MC1 neighborhoods showed a marked increase, primarily at months 1.7–2.9, and 'tapering off' by month 11 (Fig. 3e). LC^AS did not exhibit consistent

changes in MC4, except for a small number of neighborhoods at months 6–8.9 attributable to one participant (Extended Data Fig. 5b). Further, area under the curve (AUC) scores of pathways (calculated per cell from all donors) revealed that MC4, which was uniquely abundant in LC[AM], showed significantly higher expression of TGFβ and WNT–β-catenin signaling genes than MC1, MC2 and MC3 (Fig. 3f). MC1 showed higher expression of the TNF signaling genes (Fig. 3f), whereas MC1 and MC3 showed higher IFNγ response gene expression (Extended Data Fig. 5c). We further performed trajectory analysis (unstratified by disease category or groups) that revealed that lineage 3 overlapped closely with the MC4 immune program (Extended Data Fig. 5d,e), indicating that MC4 cells in LC[AM] have a distinct transcriptional profile compared to MC1–MC3. We next assessed the correlation between the frequency of MC4 within CD14+ monocytes for all LC and R[LC] samples from all time points with clinical parameters. A modest but statistically significant positive correlation was found between MC4 proportion and FAS score, whereas the correlation with $pO_2$ was negative (Fig. 3g). By contrast, a higher MC1 proportion was negatively correlated with FAS score (Fig. 3g). The modest MC4–FAS correlation likely reflected participant heterogeneity (Extended Data Fig. 5f). Individuals with LC with a high proportion of MC4 (MC4[hi]) exhibited significantly greater fatigue than those with LC with a low proportion of MC4 (MC4[lo]) or R[LC] (Fig. 3h). These findings indicate that increased MC4 abundance (referred to hereafter as LC monocyte transcriptional state (LC-Mo state)) is associated with LC, as demonstrated by its correlation with both FAS scores and $pO_2$ levels.

### LC monocytes exhibit increased LC-Mo protein marker expression

To validate the LC-Mo state, we generated and analyzed scRNA-seq data from PBMCs from cohort 3, comprising eight individuals with LC[AM] with LC symptoms reported for 8–42 months at the time of sampling (Supplementary Table 3 and Methods). All individuals with LC reported fatigue and dyspnea (classified as respiratory PASC ('Resp-PASC'), $n = 5$), and three exhibited bronchial hyper-responsiveness (BHR)[31], termed 'Resp-PASC-BHR' ($n = 3$). Three clusters (Clust0–Clust2) were identified within CD14+ monocytes (Fig. 4a). Clust1 showed significantly elevated AUC scores for the LC-Mo signature (Fig. 4b). Individuals with Resp-PASC-BHR showed significantly higher expression of the LC-Mo signature in Clust1 than those with Resp-PASC (Fig. 4c), providing independent validation of the LC-Mo state in LC[AM] and suggesting a link with progression to severe respiratory PASC.

We next performed flow cytometry analysis on PBMCs from donors in cohort 4, which included 40 LC samples 3–14 months after acute COVID-19 (Supplementary Table 4 and Methods) and 10 R[A] or NI donors. LC showed a significant increase in CD14+ monocyte percentages compared to R[A] + NI (Fig. 4d), independent of acute COVID-19 severity. We assessed the expression of 11 proteins (HLA-DR, HLA-DQ, CD105, CD51, TGFβ1, CD99, CD120b, CALR, IRF8, IFNGR1 and CD163) corresponding to LC-Mo transcripts in total CD14+ monocytes in samples from individuals with LC and R[A] + NI (Extended Data Fig. 6a). Median fluorescence intensity (MFI) of HLA-DQ, CD120b, CALR, CD99

and TGFβ1 was significantly higher in LC compared to in R[A] + NI (Fig. 4e and Extended Data Fig. 6b), whereas HLA-DR, CD51, CD105, IRF8, IFNGR1 and CD163 showed no significant difference (Extended Data Fig. 6c). Stratification by fatigue scores and dyspnea (range 1–3) revealed consistently higher MFI of CALR, CD120b, HLA-DQ and TGFβ1 in those with more severe LC symptoms (Fig. 4f,g), and TGFβ1 MFI inversely correlated with $pO_2$ (Extended Data Fig. 6d, top). MFI of both TGFβ1 and IRF8 positively correlated with each other (Extended Data Fig. 6d, bottom). Thus, protein markers specific to LC-Mo were elevated in LC, supporting an association between the LC-Mo signature and LC pathology.

### Chromatin profiling reveals AP-1/NF-κB1 activity in LC-Mo

We next investigated epigenetic regulation using snATAC-seq data from individuals with LC in cohort 1. Examination of motif signals in the chromatin landscapes of CD14+ monocytes, CD8+ T cells and NK cells from individuals with LC[AM] compared to those with R[LC] at multiple time points identified a persistent positive signal for AP-1 family activity in CD8+ T cells and NK cells (Extended Data Fig. 7a). In CD14+ monocytes, AP-1 motif accessibility was elevated up to month 5.9, after which motif enrichment shifted toward transcription factors involved in downstream TGFβ signaling, notably SP1 and KLF family of transcription factors at months 3–8.9 (Extended Data Fig. 7b). MC4 showed the highest number of differentially accessible regions (Fig. 5a). The open chromatin landscape of MC4 showed highest enrichment for motifs for ETS family transcription factors, including GABPA, ETV1, ETV4, SPI1 and SPIC (Fig. 5b). Correlating open chromatin regions with gene expression revealed significant positive associations for proangiogenic and cell adhesion genes (*VEGFA*, *ENG*, *TGFB1*, *RXRA*, *ICAM1* and *ITGA5*) and genes implicated in inflammatory/metabolic diseases (*TTC7A*, *LMNA* and *IER3*) among others (Fig. 5c). AP-1 family, SMADs, NF-κB1 and RELA transcription factor motifs showed a marked increase within the accessible chromatin regions of these genes (Fig. 5d). Within MC4, correlation of transcription factor transcripts and target gene transcripts with accessible motifs enabled pinpointing of noncoding regulatory regions associated with gene expression, such as those for *IER3* and *LMNA*, and establishment of gene–transcription factor relationships (such as NF-κB1 and AP-1 family likely regulators of *LMNA*; Fig. 5e–g and Extended Data Fig. 7c). In summary, these findings indicate that LC-Mo is driven by ETS, AP-1 and NF-κB1 transcription factors.

### BAL myeloid cells show LC-Mo and profibrotic programs

Circulating monocytes contribute to PASC pathogenesis, particularly pulmonary fibrosis[23,32]. To assess whether LC-Mo participates in fibrotic lung remodeling, we analyzed paired PBMC and BAL fluid samples from a public dataset[23] (cohort 5) consisting of nine individuals with LC of unknown severity during AI (LC[UN]), classified based on lung function as Resp-PASC ($n = 5$) or nonResp-PASC ($n = 4$), and PBMCs from NI donors ($n = 2$; Supplementary Table 5). Circulating CD14+ myeloid cells were reclustered to identify CD14+CD16− monocytes (Extended Data Fig. 8a), leading to six clusters (CL0–CL5; Fig. 6a). CL5 showed the highest enrichment of LC-Mo signature AUC scores (Fig. 6b and Extended Data Fig. 8b). Within cluster 5,

**Fig. 5 | AP-1 and NF-κB1 transcription factors regulate LC-Mo in CD14+ monocytes from individuals with LC[AM]. a**, Top significant peaks calculated from snATAC-seq data from cohort 1 ($n = 78$), with aggregated peaks from cells in MC1–MC4 subclusters; data were analyzed by two-sided Wilcoxon test (Benjamini–Hochberg method-adjusted $P < 0.05$). **b**, Top ChromVar transcription factor motif enrichment in open chromatin regions in MC4 compared to MC1–MC3; Avg diff, average difference. **c**, Expression of genes significantly correlated with open chromatin regions within the MC4 subcluster (Pearson's correlation (Benjamini–Hochberg method-adjusted $P < 0.05$)). **d**, Scatter plot of enriched transcription factors motifs (fold enrichment) within open chromatin regions correlated with the expression of genes in **c** against the ChromVar transcription

factors motif enrichment as in **b**; TF, transcription factor. Dashed horizontal lines represent −0.1 and 0.1 ChromVAR average difference; dashed vertical line represents 0.5 fold enrichment. **e**, Correlation (Corr) between the expression of genes in **c** with the expression of transcription factors identified in **b** and **d**. Data were analyzed by Pearson's correlation. Absolute correlations of >0.3 are plotted. The black asterisk indicates transcription factors with motifs in the open chromatin of the correlated gene. **f,g**, Coverage plots showing the chromatin accessibility regions and gene expression of *IER3* (**f**) and *LMNA* (**g**) in subclusters MC1–MC4 and correlations between open chromatin regions and transcription factors with binding sites (gray lined boxes); bp, base pairs; $P_{adj}$, adjusted $P$ value.

Resp-PASC exhibited significantly higher LC-Mo expression than nonResp-PASC or NI (Fig. 6c).

We next integrated CD14⁺ monocytes from PBMCs and CD163⁺ or CD14⁺ myeloid cells from BAL fluid. This integrated dataset identified CI1 with >75% cells from BAL fluid and expressing *MARCO⁺FABP4⁺*, markers for tissue-resident alveolar macrophages, two clusters (CI2 and CI3) with >65% of cells from PBMCs and expressing *LYZ⁺CD14⁺*, markers for circulating monocytes, and three mixed clusters (CI4–CI6) with comparable proportion of cells from both PBMCs and BAL (Fig. 6d and Extended Data Fig. 8c,d). PBMC monocytes in CL5 primarily localized to clusters CI4–CI6 (Fig. 6e), suggesting a macrophage-polarized phenotype. Among these, cluster CI4 had the highest LC-Mo signature

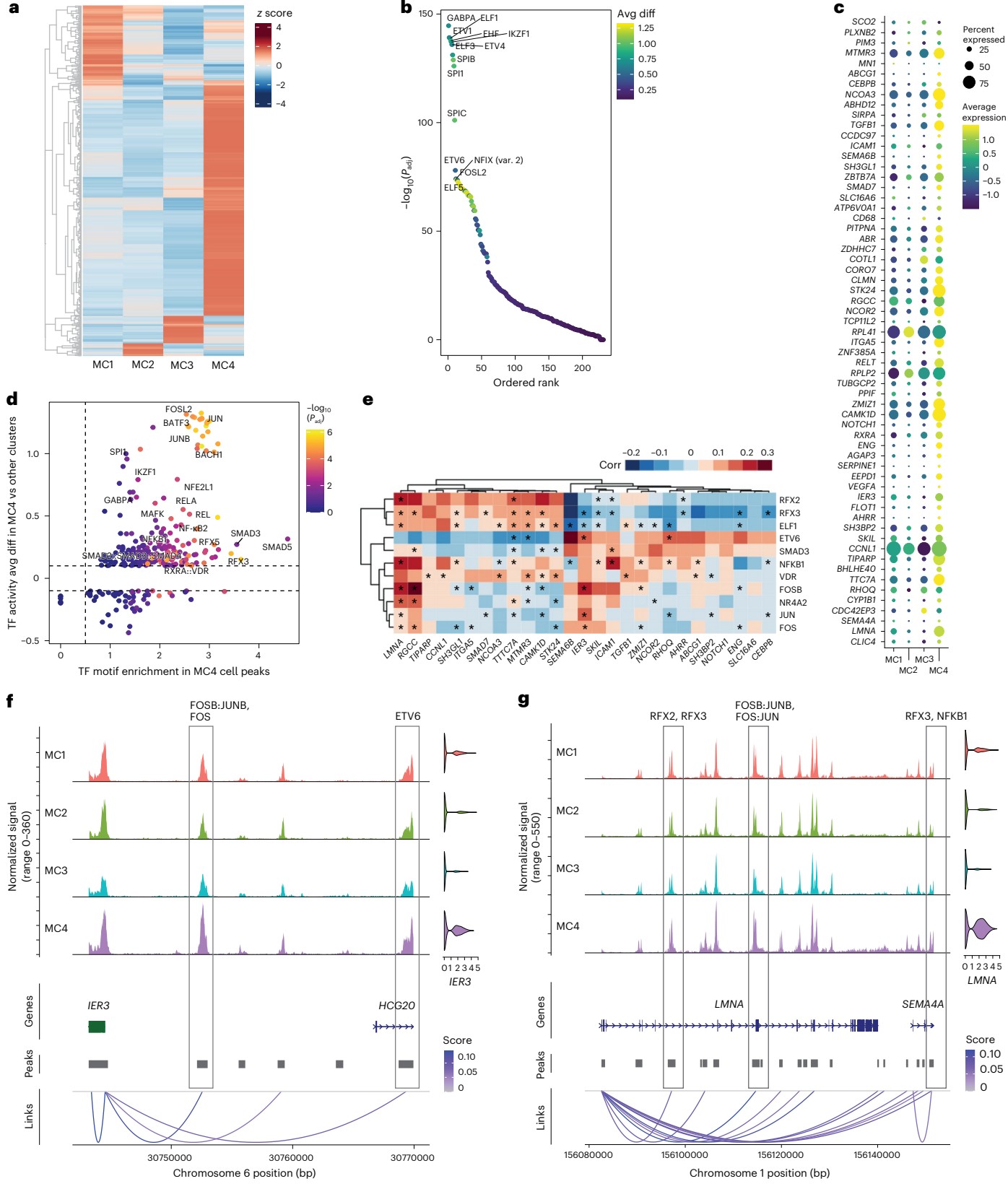

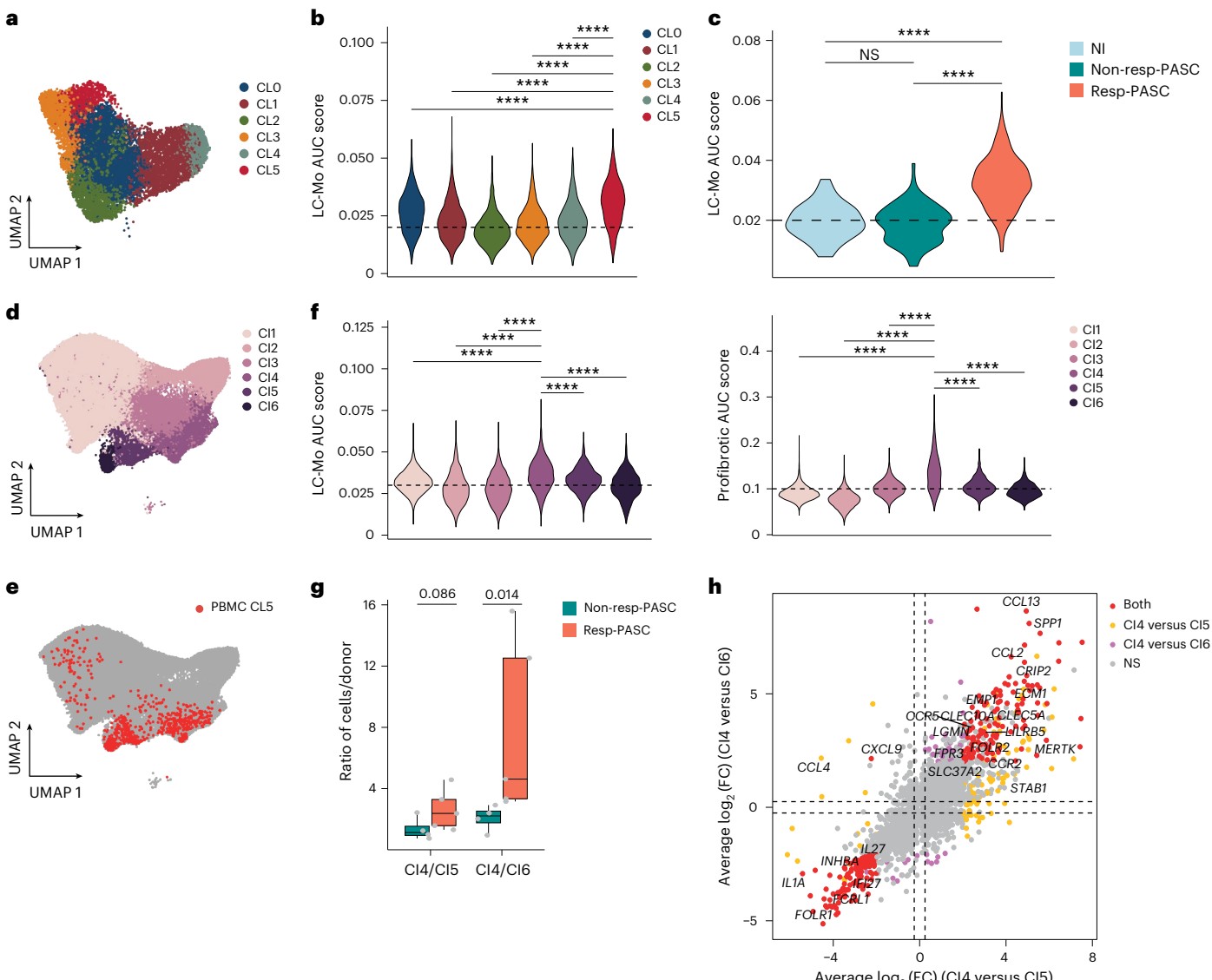

**Fig. 6 | The LC-Mo cluster is enriched in profibrotic monocyte-derived alveolar macrophages in BAL fluid from individuals with LC. a**, UMAP of CD14⁺ monocytes cells from PBMCs of cohort 5 (GEO: GSE263817)[23] subclusters *FABP4⁺C1QA⁺* CD14⁺ monocytes (CL0), *NKG7*ᵗ*GZMB⁺* CD14⁺ monocytes (CL1), *FCN1⁺S100A9⁺* CD14⁺ monocytes (CL2), *KLRC2⁺LAG3⁺* CD14⁺ monocytes (CL3), *NLRC5⁺* CD14⁺ monocytes (CL4) and *TREM2⁺CALR⁺* CD14⁺ monocytes (CL5) from NI donors (*n* = 2) and LCᵁᴺ donors (*n* = 9). **b**, AUC scores of the LC-Mo signature in CL0–CL5 as in **a**; data were analyzed by two-sided Wilcoxon rank-sum test; ****P < 0.00001. **c**, AUC scores of the LC-Mo signature in CL5 in PBMC CD14⁺ monocytes from individuals with LC Resp-PASC (*n* = 5) and nonResp-PASC (*n* = 4) and NI donors (*n* = 2); data were analyzed by two-sided Wilcoxon rank-sum test; ****P < 0.00001. **d**, Integrated UMAP of CD163⁺ or CD14⁺ myeloid cells from PBMCs and BAL samples of individuals with LCᵁᴺ (*n* = 9) showing *MARCO⁺FABP4⁺* macrophages (CI1), *LYZ⁺CD14⁺* monocytes (CI2–CI3) and mix clusters from

PBMCs and BAL samples with *TREM2⁺CCL2⁺* (CI4), *CCL23⁺*(CI5) and *NUPR8⁺* (CI6). **e**, UMAP as in **d** showing CL5 cells. **f**, LC-Mo AUC score within CI1–CI6 (left) and profibrotic gene signature[33] AUC score as in **d**. Data were analyzed by two-sided Wilcoxon rank-sum test; ****P < 0.00001. Horizontal dashed lines in **b**, **c** and **f** serve as visual reference for comparison of relative shifts in pathway AUC scores across clusters. **g**, Ratio of CI4/CI5 or CI4/CI6 cells within each individual with Resp-PASC (*n* = 5) or nonResp-PASC (*n* = 4). Data were analyzed by two-sided Wilcoxon rank-sum test. Box plots show the median (center), first and third quartiles (bounds) and 1.5 times the interquartile range (whiskers). **h**, Scatter plots showing log₂(fold change) (log₂ (FC)) of DGE in the CI4 versus CI5 and CI4 versus CI6 clusters. Genes significant with an adjusted *P* value of <0.05 in both comparisons are labeled (two-sided Wilcoxon rank-sum test, Benjamini–Hochberg method-adjusted *P* < 0.05). All data correspond to cohort 5 scRNA-seq data.

enrichment and higher expression of a profibrotic gene set defined in prior COVID-19 BAL studies[33] and including *TREM2*, *CALM1*, *LGMN* and *APOE* (Fig. 6f and Extended Data Fig. 8e). Individuals with resp-PASC showed a higher proportion of CI4 cells and higher CI4/CI5 and CI4/CI6 ratios than individuals without resp-PASC (Fig. 6g and Extended Data Fig. 8f). Differential expression analysis revealed that CI4 cells upregulated the expression of *SPP1*, *CCL13*, *CCL2* and *FOLR2* compared to CI5 or CI6 cells from both individuals with resp-PASC and non-resp-PASC (Fig. 6h). These results indicate LC-Mo enrichment in

Resp-PASC PBMCs and its association with a profibrotic transcriptional profile in lung myeloid cells.

### Individuals with LC-Mo exhibit dysregulated monocyte function

To assess the functional implications of LC-Mo during immune challenge, we stimulated PBMC samples from cohort 1 (months 1.7–2.9 and 6–8.9) with heat-inactivated *Pseudomonas aeruginosa* for 4 h and performed single-cell multiome profiling in samples from

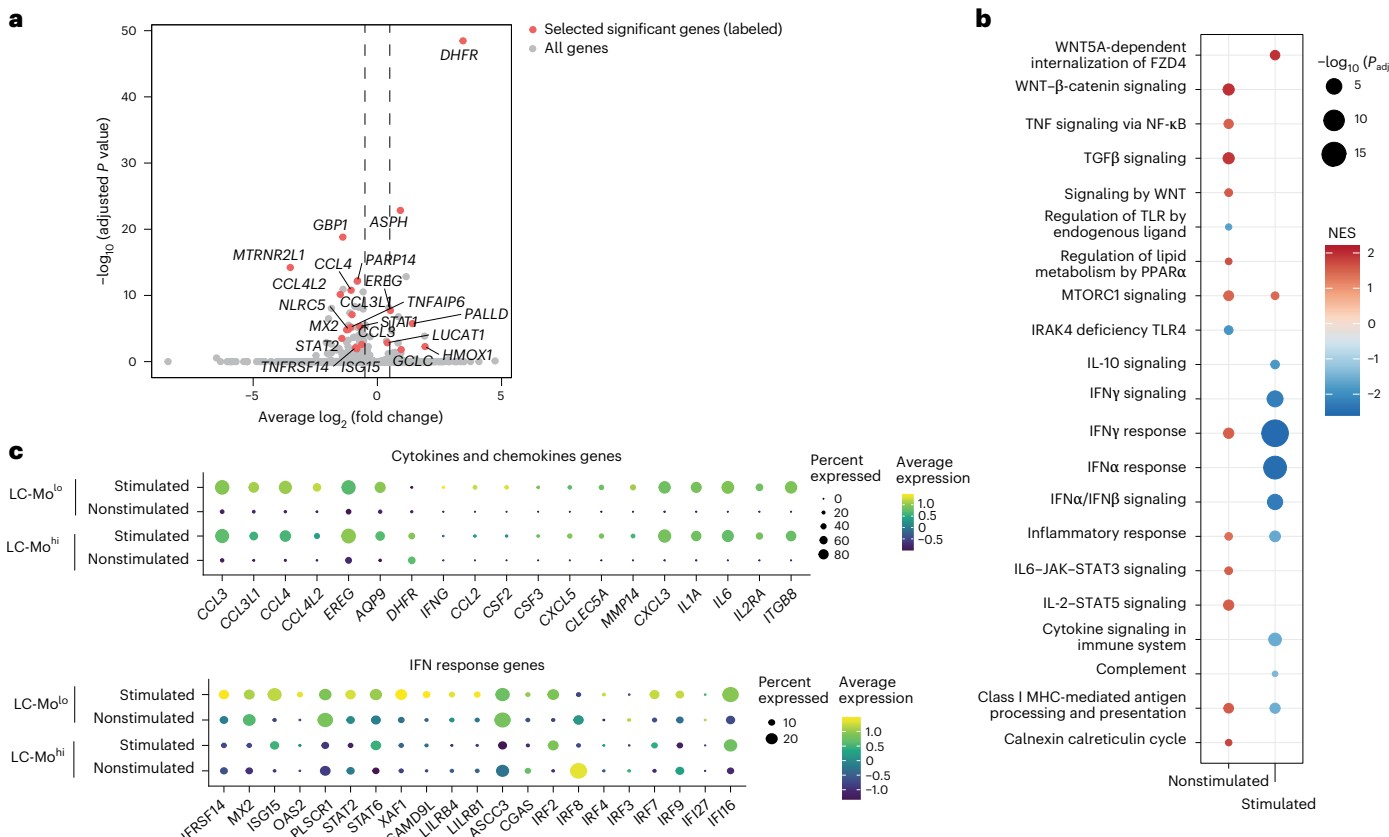

**Fig. 7 | LC-Mo is linked to dysregulation of CD14⁺ monocyte function in LC^AM. a**, Volcano plot showing DGE in CD14⁺ monocytes from LC-Mo^hi (>10% of CD14⁺ monocytes exhibiting the LC-Mo state) versus LC-Mo^lo (<10% of CD14⁺ monocytes exhibiting the LC-Mo state) LC^AM ($n = 7$), LC^AS ($n = 5$) and R^LC ($n = 6$) samples collected from cohort 1 at month 1.7–2.9 and month 6–8.9 time points and stimulated with *P. aeruginosa* for 4 h. Selected significant labeled genes with an adjusted *P* value of <0.05 are shown in red, whereas all other genes are shown in gray. Data were analyzed by Wilcoxon rank-sum test, Benjamini–Hochberg method-adjusted *P* < 0.05. **b**, GSEA enrichment of pathways in nonstimulated LC-Mo^hi versus LC-Mo^lo and stimulated LC-Mo^hi versus LC-Mo^lo (unstimulated LC-Mo^hi, $n = 11$; unstimulated LC-Mo^lo, $n = 23$; stimulated LC-Mo^hi, $n = 4$; stimulated LC-Mo^lo, $n = 14$). Pathways plotted with an adjusted *P* value of <0.1 (Kolmogorov–Smirnov-based test with permutation-derived *P* values, adjusted using the Benjamini–Hochberg method). **c**, Dot plot showing gene expression of chemokine and cytokine genes (top) and IFN response genes (bottom) in stimulated and nonstimulated CD14⁺ monocytes from LC-Mo^hi and LC-Mo^lo as in **a**.

individuals with LC^AM ($n = 7$), LC^AS ($n = 5$) and R^LC ($n = 6$; Extended Data Fig. 9a,b). Stimulation resulted in a reduction in the numbers of CD14⁺ and CD16⁺ monocytes compared to unstimulated samples (Extended Data Fig. 9c,d), consistent with prior reports[34,35]. Joint analysis of stimulated and unstimulated samples showed that stimulated LC^AM CD14⁺ monocytes exhibited significant downregulation of the inflammatory response, IFNγ signaling, IL-10 signaling, cytokine signaling and IL-6–JAK–STAT3 signaling pathways relative to stimulated R^LC CD14⁺ monocytes (Extended Data Fig. 9e). Next, we classified donors as LC-Mo^hi (>10% of CD14⁺ monocytes exhibiting the LC-Mo state) or LC-Mo^lo (<10%); all R^LC and LC^AS samples were LC-Mo^lo (Extended Data Fig. 5f). Comparison of stimulated LC-Mo^hi and LC-Mo^lo identified *DHFR*, *HMOX1*, *EREG* and *GCLC* among the top significantly upregulated DEGs (Fig. 7a). Pathways related to 'IFNα response' and 'cytokine signaling' were significantly decreased in expression (Fig. 7b) in stimulated LC-Mo^hi compared to stimulated LC-Mo^lo. At the gene level, stimulation induced cytokine and chemokine gene expression (*CCL3*, *CCL4*, *CXCL3* and *IL6*) in both stimulated LC-Mo^hi and stimulated LC-Mo^lo, whereas IFN response genes (*IRF9*, *ASCC3*, *XAF1*, *SAMD9L*, *LILRB4* and *CGAS*) were downregulated in LC-Mo^hi (Fig. 7c). Motif accessibility analysis of stimulated LC-Mo^hi and stimulated LC-Mo^lo showed that FOXO and TCF (especially TCF7L2) and ZIC motifs were more accessible in stimulated LC-Mo^hi, whereas stimulated LC-Mo^lo showed increased ETS and AP-1 motif accessibility compared to stimulated LC-Mo^hi (Extended Data Fig. 9f). Together, these data suggest

that LC-Mo might contribute to the functional immune dysregulation observed in individuals with LC.

## Discussion

Using high-resolution single-cell multiome analysis, immunological profiling and functional assays on PBMC and BAL samples from individuals with LC experiencing fatigue and dyspnea, we identified persistent elevations of proinflammatory mediators such as TNF, CCL2 and CXCL11 up to 9 months after infection. We also defined a distinct circulating CD14⁺ monocyte state (LC-Mo) associated with LC. This state, predominant in individuals with LC^AM, showed increased TGFβ/WNT–β-catenin signaling that increased over time and exhibited interindividual variability. Individuals with severe resp-PASC displayed higher LC-Mo gene expression, whereas individuals with higher LC-Mo proportions showed reduced IFN responses after in vitro stimulation, suggesting a compromised immune response.

Although 14 individuals in cohort 1 and 51 in cohort 2 were enrolled before the 3-month National Academies of Sciences, Engineering, and Medicine cutoff for LC, over 70% had symptoms extending beyond this period, aligning with established diagnostic criteria. Overlapping symptoms with post-intensive care syndrome complicate LC heterogeneity. Our data revealed molecular differences in PBMCs based on the severity of the AI. Circulating monocytes have been implicated in severe COVID-19 disease and in resp-PASC[23,32,33,36,37]. Although oxygen saturation in cohort 1 was normal, MC4 cell proportions negatively

correlated with oxygen saturation, suggesting subtle gas exchange defects. Nevertheless, increased LC-Mo expression in PBMCs and BAL fluid from individuals with severe resp-PASC in cohorts 3 and 5, together with a profibrotic BAL phenotype, support a link to lung pathology.

Sustained TNF expression, reported in post-COVID cohorts[8,13,38], paralleled persistent TNF and/or NF-κB signaling in immune subsets, driving systemic inflammation. Enrichment of *GZMK*+CD8+ T and NK cells in LC[AM], shown to expand after SARS-CoV-2 infection[30] and in chronic inflammatory diseases[28,29,39,40], and increased AP-1 accessibility were consistent with TNF-driven activation[41]. Upregulated TLR pathways in CD8+ T cells and NK cells indicated ongoing viral sensing and potential NF-κB1 activation, fitting with evidence of persistent viral reservoirs or remnants[42–45].

By contrast, CD14+ monocytes from individuals with LC[AM] showed transient TNF signaling, with persistent *TGFB1* expression and activation of TGFβ and WNT–β-catenin pathways for up to 11 months. MC4 proportions showed coexpression of TGFβ and IRF8 mRNA and protein, shown to synergistically drive neuroinflammation in the experimental autoimmune encephalomyelitis mouse model[46], and displayed motifs for ETS1, AP-1, NF-κB1 and SMAD, transcription factors linked to adhesion and fibrosis[47,48]. MC4 proportions modestly correlated with FAS scores and blood $pO_2$, and flow cytometry confirmed elevated TGFβ1 in CD14+ monocytes. LC-Mo enrichment was also validated in two independent cohorts with severe Resp-PASC patients, thereby linking LC-Mo to lung fibrosis.

Although associations between LC-Mo and symptom severity were noted, correlations were modest, leaving causality undetermined. Further functional studies are needed to clarify these mechanisms. Our study focused on respiratory symptoms and fatigue, so the involvement of LC-Mo in other organ systems remains open. Comparison to unstratified recovered controls (due to limited sample size) restricts interpretation; future work with stratified groups and consideration of vaccination or comorbidities will be important.

In conclusion, we provide a systems view of LC with fatigue and respiratory involvement, identifying a pathogenic monocyte state linked to severe symptoms and offering insights into disease mechanisms and heterogeneity.

## Online content

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

[1]Centre for Individualised Infection Medicine (CiiM), a joint venture between the Helmholtz Centre for Infection Research (HZI) and Hannover Medical School (MHH), Hannover, Germany. [2]TWINCORE, a joint venture between the Helmholtz Centre for Infection Research (HZI) and the Hannover Medical School (MHH), Hannover, Germany. [3]Beirne B. Carter Centre for Immunology Research, University of Virginia, Charlottesville, VA, USA. [4]Division of Infectious Disease and International Health, Department of Medicine, University of Virginia, Charlottesville, VA, USA. [5]Hannover Unified Biobank, Hannover Medical School (MHH), Hannover, Germany. [6]Institute of Virology, Hannover Medical School (MHH), Hannover, Germany. [7]Institute for Pulmonary Rehabilitation Research, Schön Klinik Berchtesgadener Land, Schönau am Königssee, Germany. [8]PRACTIS Clinician Scientist Program, Dean's Office for Academic Career Development, Hannover Medical School, Hannover, Germany. [9]Department of Gastroenterology, Hepatology, Infectious Diseases and Endocrinology, Hannover Medical School (MHH), Hannover, Germany. [10]Department of Human Genetics, Hannover Medical School (MHH), Hannover, Germany. [11]German Centre for Infection Research (DZIF), partner site Hannover-Braunschweig, Hannover, Germany. [12]Cluster of Excellence Resolving Infection Susceptibility (RESIST; EXC 2155), Hannover Medical School, Hannover, Germany. [13]Department of Pulmonary Rehabilitation, German Center for Lung Research (DZL), University Medical Center Giessen and Marburg, Philipps-University Marburg (UMR), Marburg, Germany. [14]Department of Medicine, Pulmonary and Critical Care Medicine, University Medical Center Giessen and Marburg, Philipps University Marburg, German Center for Lung Research (DZL), Marburg, Germany. [15]Teaching Hospital, Paracelsus Medical University Salzburg, Salzburg, Austria. [16]Department of Respiratory Medicine and Infectious Diseases, Hannover Medical School, Hannover, Germany. [17]Biomedical Research in Endstage and Obstructive Lung Disease Hannover (BREATH), German Center for Lung Research (DZL), Hannover, Germany. [18]Department of Internal Medicine and Radboud Center for Infectious Diseases, Radboud University Medical Center, Nijmegen, the Netherlands. [19]Institute for Molecular Bacteriology, TWINCORE, Centre for Experimental and Clinical Infection Research, Hannover, Germany. [20]Department of Clinical Microbiology, Rigshospitalet, Copenhagen, Denmark. [21]Department of Molecular Bacteriology, Helmholtz Centre for Infection Research, Braunschweig, Germany. [22]Department of Immunology and Metabolism, Life and Medical Sciences Institute (LIMES), University of Bonn, Bonn, Germany. [23]Lower Saxony Center for Artificial Intelligence and Causal Methods in Medicine (CAIMed), Hannover, Germany. [24]These authors jointly supervised this work: Thomas Illig, Jie Sun, Yang Li. ✉e-mail: yang.li@helmholtz-hzi.de

## Methods

### Cohorts

**Cohorts 1, 2 and 4.** Sample collection and analyses were approved by the ethics committee of the MHH (ethics vote 9001_BO_K). All participants gave written broad consent. Blood was collected from individuals who were admitted to the hospital due to COVID-19, as well as from ambulatory individuals after SARS-COV-2 infection showing symptoms indicative of LC >4 weeks after infection in accordance with the German S1 guideline for LC and post-COVID syndrome at the MHH. Cohort 1 included individuals with AI$^M$ ($n$ = 7, 42.8% women, age median = 52, range 23–66 years) and AI$^S$ ($n$ = 4, 50% women, age median = 37, range 32–54 years), individuals with LC symptoms (LC$^{AM}$ ($n$ = 29, 8 longitudinal with two to three time points, 21 single-time-point donors, 58% women, median age = 49, range 31–84) and LC$^{AS}$ ($n$ = 8, 3 longitudinal with two to four time points, 5 single-time-point donors, 25% women, median age = 46, range 19–75)), donors recovered from LC (4–8 months of LC; R$^{LC}$ ($n$ = 8, 37.5% women, median age = 38, range 19–65)) and prepandemic NI control individuals ($n$ = 6, 50% women, median age = 40, range 24–61). LC and R$^{LC}$ samples were collected 1.7–10.2 months after infection, and all groups were recruited between April 2020 and August 2021 at MHH. Cohort 2 included individuals with LC$^{AM}$ ($n$ = 117, 24 longitudinal with two to four time points, 93 single-time-point donors, 58.9% women, median age = 48, range 19–83) and LC$^{AS}$ ($n$ = 25, 12 longitudinal, 13 single-time-point donors, 20% women, median age = 53, range 18–81), recruited between May 2020 and August 2021 at MHH, along with prepandemic NI samples ($n$ = 33, 48.4% women, median age = 40, range 25–65). Cohort 4 included individuals with LC$^{AM}$ ($n$ = 29, 58.6% women, median age = 49, range 33–72) and LC$^{AS}$ ($n$ = 11, 18% women, median age = 57, range 35–81) and individuals recovered from AI (R$^A$ $n$ = 8 and NI $n$ = 2, 60% women, median age = 41, range 29–67), recruited between August 2020 and June 2022 at MHH. All individuals with AI had a positive SARS-CoV-2 PCR test at admission or externally before admission. All individuals with LC had a prior proven SARS-CoV-2 infection. The SARS-CoV-2 strain in individuals with AI or LC was not recorded. Clinical parameters, including blood gas measurements, pulmonary function tests, FAS and mMRC scores and quality-of-life assessments, were systematically collected at each visit.

**Cohort 3.** Sample collection and analyses were approved by the ethics committee of the Philipps University Marburg (Az.:24-289 'Entschlüsselung der molekularen Pathophysiologie des Post-Covid-Syndroms und prädisponierender Faktoren mit Hilfe neuer Sequenzierungstechnologien und Phänotypisierung von Immunzellen'). All participants gave written broad consent. Samples was collected from individuals with LC$^{AM}$ ($n$ = 8, 62.5% women, median age = 45, range 21–63), all with resp-PASC and recruited between October 2023 and November 2023 during their stay at the Pulmonary Rehabilitation Clinic in Schönau am Königssee, Germany. All individuals with LC had prior proven SARS-CoV-2 infection, and samples were collected >6 months after SARS-CoV-2 infection, one sample per participant. The SARS-CoV-2 strain in individuals with AI or LC was not recorded. Clinical parameters, including blood gas measurements, pulmonary function tests, FAS and mMRC scores and quality-of-life assessments, were systematically collected for each participant.

**Cohort 5.** Biosample collection for both PBMCs and BAL fluid is available at ref. [23] and included individuals with LC ($n$ = 9, 44.4% women, median age = 64, range 62–83), including five with resp-PASC, and NI donors ($n$ = 2, 50% women, median age 77, range 73–77), recruited between October 2020 and November 2021 at Mayo Clinic.

### Sample processing for PBMCs

Sample processing for cohorts 1–4 and storage was performed following the standard procedures of the Hannover Unified Biobank (HUB) as described by Kopfnagel et al.[49]. PBMCs were isolated from whole blood using Ficoll gradient centrifugation. Cohort 5 PBMC and BAL sample processing was performed similar to as described previously[23].

### Cytokine assay (cohort 2)

The Quanterix HD SP-X Imaging and Analysis System was used to measure plasma samples. The following panels were used in this study: Human Corplex cytokine panel 1 10-Plex array including IL-12p70, IL-1β, IL-4, IL-5, IL-6, IL-8, TNF, IFNγ, IL-10 and IL-22. The Simoa chemokine panel 1 4-plex kit contained four chemokines, including IP-10 (CXCL10), MCP1 (CCL2), MIP1-β (CCL19) and ITAC (CXCL11). IL-4 and IL-5 were excluded from further analysis due to being below the limit of detection. The detection values were log$_2$ transformed. All plasma samples were processed according to standard biobanking protocols and stored at a minimum temperature of –80 °C. For the experiments, the samples were randomized and measured according to the manufacturer's manual. The study protocol conformed to the ethical guidelines of the Declaration of Helsinki, and the ethics committee of MHH approved this study a priori (9001_BO_K, No. 9472_BO_K_2020, broad consent: 2923-2015). Informed consent was obtained from all participants included in the study.

### *P. aeruginosa* stock production

A *P. aeruginosa* clinical isolate CH5464 was streaked from frozen glycerol stocks onto LB agar plates and incubated overnight at 37 °C. Bacteria from single colonies were used to inoculate an overnight preculture in LB medium. This preculture was then diluted in fresh LB medium and grown at 37 °C with shaking at 180 rpm until reaching the early stationary phase. The culture was centrifuged at 10,000$g$ for 10 min, and the supernatant was discarded. The pellet was washed twice with PBS and incubated at 80 °C for 60 min to inactivate the bacteria in a waterbath. Afterward, the suspension was centrifuged again at 10,000$g$ for 10 min to remove cellular debris. The bacterial suspension was adjusted to a concentration of 10$^8$ colony-forming units (c.f.u.) per ml and stored at –20 °C for future use. To confirm complete bacterial inactivation, 100 µl of the bacterial suspension was plated on blood agar plates.

### In vitro PBMC stimulation

We conducted scMultiome-seq analysis on PBMCs from individuals with LC across five time points: the acute phase; 3, 9 and 12 months after infection and during recovery. Heat-inactivated *P. aeruginosa* strain and a mock stimulation condition were tested over the course of four experimental runs. A pilot study was performed using samples from two healthy individuals, with cells stimulated for 4 and 24 h at four different concentrations to determine the optimal conditions. Based on this pilot study, the 4-h time point and 2.5 × 10$^6$ c.f.u. per ml were identified as optimal.

For the main experiment, PBMCs were thawed according to an optimized protocol based on 10x Genomics guidelines (CG000365, Rev B). The cells were counted and resuspended at a concentration of 5 × 10$^6$ cells per ml in warm RPMI. Cell suspension (100 µl) was plated into a 96-well, round-bottom plate and rested for 1 h at 37 °C. Following this rest period, the RPMI medium was replaced with 100 µl of heat-inactivated *P. aeruginosa* corresponding to a concentration of 2.5 × 10$^6$ c.f.u. per ml. The cells were incubated at 37 °C for 4 h. After incubation, the plates were centrifuged at 300$g$ for 5 min, and the cells were collected for nuclei isolation and library preparation.

### Isolation of nuclei and library preparation (cohort 1)

scMultiome-seq analysis was performed on both directly thawed and stimulated PBMCs. To manage sequencing costs, cells from three to four donors were pooled together. After pooling, the cells were treated with DNase I to remove free DNA and centrifuged at 300$g$ for 10 min at 4 °C. The cell pellets were resuspended and incubated with 300 µl of prechilled 1× lysis buffer on ice for 3 min. Lysis was stopped by adding 1 ml of ice-cold wash buffer, followed by centrifugation at 500$g$ for

5 min at 4 °C. The nuclei were washed twice with 500 μl of wash buffer and resuspended in Diluted Nuclei Buffer. To ensure purity and dissociation of single nuclei, the suspension was passed through a 40-μm Flowmi strainer and inspected under a microscope.

Approximately 20,000 nuclei were loaded into a Chromium Controller to produce single-cell gel beads, following the 10x Genomics Chromium Next GEM Single Cell Multiome ATAC + Gene Expression protocol (CG000338, Rev C). After transposition, the nuclei were treated with a transposase enzyme, which selectively fragmented the accessible DNA regions and added adapter sequences to the fragmented DNA ends. The transposed nuclei were loaded onto a Chromium Next GEM Chip J (PN-1000234), alongside partitioning oil and barcoded gel beads. PCR amplification was performed, targeting approximately 10,000 nuclei per library. Sequencing was performed using the Illumina NovaSeq 6000 platform, with a minimum read depth of 20,000 read pairs per cell for scRNA libraries and 25,000 read pairs per cell for scATAC libraries.

### Cell capture and library preparation (cohort 3)
We thawed cells following the 10x Genomics thawing protocol (CG00039, Rev D), cells from four donors were pooled together, and approximately 29,000 cells were loaded into the Chromium X (10x Genomics) to generate single-cell gel beads in emulsion according to the 10x Genomics protocol (CG000731, Rev A). scRNA-seq libraries were prepared using a Chromium GEM-X Single Cell 3′ Reagent Kits v4 (10x Genomics) and sequenced on the NovaSeq 6000 platform (Illumina), with a minimum depth of 20,000 read pairs per cell.

### Sample preparation for ex vivo flow cytometry experiments (cohort 4)
PBMCs were isolated from fresh whole blood using standard Ficoll density gradient centrifugation and cryopreserved in liquid nitrogen for deferred use. Ex vivo phenotyping of immune cells was performed from cryopreserved PBMCs. In brief, thawed PBMCs were stained with a Zombie NIR Fixable Viability kit (Biolegend, 423106) at room temperature in PBS for 15 min. Nonspecific immunolabeling conferred by Fc receptor binding was blocked by the addition of 10% Gamunex solution (Grifols Deutschland). Surface marker immunolabeling was performed in cell staining buffer (PBS, BSA and EDTA) (Biolegend, 420201) and Brilliant Stain Buffer (BD, 563794) overnight at 4 °C with antibodies to the ontogeny markers anti-human CD3, CD14, CD16 and HLA-DR; macrophage markers CD163 and CD206 and the markers identified from the transcriptomic analysis CD51, CD99, CD105, CD120b and HLA-DQ (see antibody details in the table). After fixation and permeabilization (BD, 554714) for 30 min at room temperature, immunolabeling of intracellular markers was performed for 30 min in Permwash buffer (BD, 554714) at 4 °C with antibodies to CALR, IFNGR1, TGFB1 and IRF8. Next, cells were immunolabeled with the secondary antibodies AF488 and AF568 for 30 min in Permwash buffer (BD, 554714) at 4 °C to label the unconjugated antibodies CALR and IFNGR1, respectively. All donors were also immunolabeled with the correspondent isotype controls for the used antibodies. Cells were washed with PBS, and data were acquired on a five-laser Sony spectral analyzer (ID7000, Sony) and analyzed with FlowJo software v10.10.0 (Tree Star).

### Antibody list

| Antibody | Fluorochrome | Clone | Company | Catalog |
| --- | --- | --- | --- | --- |
| CD3 | SparkBlue | SK7 | BioLegend | 344852 |
| CD14 | PacBlue | 63D3 | BioLegend | 367122 |
| CD16 | BUV563 | 3G8 | BD | 568289 |
| CD51 | APC | NKI-M9 | BioLegend | 327913 |
| CD99 | PE | hec2 | BioLegend | 398205 |

| Antibody | Fluorochrome | Clone | Company | Catalog |
| --- | --- | --- | --- | --- |
| CD105 | BUV421 | 43A3 | BioLegend | 323219 |
| CD120b | PE-DAZZLE | 3G702 | BioLegend | 358413 |
| CD163 | FITC/PE-Cy7 | GHI/61 | BioLegend | 333618/2268070 |
| CD206 | APC-Cy7/PE-Cy7 | 15-2 | BioLegend | 321120/321124 |
| CALR | Purified | | Abcam | ab2907 |
| | AF488 | | Invitrogen | |
| HLA-DQ | BB700 | Tu169 | BD | 745976 |
| HLA-DR | AF700 | L243 | BioLegend | 307626 |
| IFNGR1 | Purified | | Abcam | ab154400 |
| | AF568 | | Invitrogen | |
| IRF8 | PE | REA516 | Miltenyi | 130-122-927 |
| TGFB1 | PE-CF594 | TW4-9E7 | BD | 562422 |

### Statistical methods
No statistical method was used to predetermine sample size. The samples were randomized before processing for single-cell experiments. The investigators were not blinded to allocation during the experiments or during outcome assessment. All statistics in the manuscript are reported as specified in the figure legends.

### Genotyping
Genotyping of DNA samples isolated from participants in the current study was performed using the GSA-MDv3 array (Infinium, Illumina) following the manufacturer's instructions. In total, 725,875 variants of 48 individuals were called by Optical 7.0 with default settings

### Genotype processing for demultiplexing
Genotype data were reformatted into PLINK binary format files[50]. Quality control was performed at both the sample and single-nucleotide polymorphism (SNP) levels. Samples were excluded if they exhibited sex mismatches, missing genotyping rates of ≥0.05, heterozygosity rates beyond three standard deviations from the mean or relatedness across samples. A total of 58 samples passed these filters. SNPs were further filtered based on a minor allele frequency of >0.01 and an SNP missingness rate of <0.05. Genotype imputation was conducted using the Minimac4 server[51], using the TOPMed r3 reference panel[52] and EAGLE v2.4 for phasing. The final analysis included a total of 6,050,031 variants.

### Data preprocessing for multiome datasets and demultiplexing
BCL files from each library were converted to FASTQ files using cellranger-arc mkfastq with default parameters and using the respective sample sheet with the 10x barcodes. The 10x Genomics cellranger-arc count pipeline (v2.0.2) was used with default parameters using the human reference genome GRCh38-2020-A-2.0.0 obtained from 10x Genomics website. Demultiplexing was performed using Souporcell (v2.4)[53]. To assess the concordance between the genotypes of each donor in Souporcell-generated VCF and a reference VCF, BCFtools was used to perform a genotype check with parameter 'gtcheck' and the '-u GT' option to compare the genotype fields in the two VCF files.

### Quality control and integration of multiome datasets (cohort 1)
Once the donor for each cell was assigned after demultiplexing, only single cells with both RNA and ATAC data were considered. Seurat version 5.0 (ref. 54) was used for downstream analysis. The following filtration criteria were used: 'nCount_RNA < 6,000 and nCount_ATAC < 15,000, mitochondrial percentage < 20, RNA features < 3,000, TSS enrichment > 1 and <10, while nucleosome_signal < 2'. RNA integration across libraries was performed using 'RPCAIntegration' and the top 30 dimensions for both clustering and UMAP generation. Further,

multiple resolutions varying from 0.2 to 0.8 were performed to get clusters, and canonical markers were used for identifying cell subsets (using a combination of known markers and those used by Azimuth celltype. l2). For ATAC integration across libraries, signac version 1.13 (ref. 55) was used. Integration anchors were found by using 'rlsi' and 2–30 top dimensions, followed by integration using the top 30 dimensions.

### Participant and sample stratification and sample category classifications (cohorts 1, 2 and 3)

We first stratified individuals with LC into two groups based on their acute COVID-19 disease WHO scores, where LC samples from individuals with acute COVID-19 WHO scores between 1 and 5 were classified as LC[AM] and LC samples from individuals with acute COVID-19 WHO scores between 6 and 9 were classified as LC[AS]. Further samples were also stratified based on time points of collection resulting in NI, AI, T2:1.5/1.7–2.9 months, T3: 3–5.9 months, T4: 6–8.9 months and T5: 9–12 months.

For cohort 1, to ensure that our findings were not convoluted by COVID-19 infection imprinting on immune cells and were unique to LC, we performed transcriptome comparisons either to AI samples or to R[LC]. All recovered samples were considered as one category. Consequently, within each group of participants (LC[AM] or LC[AS]), comparisons were performed as T2 versus AI, T3 versus AI, T4 versus AI or T2 versus R[LC], T3 versus R[LC] and so on. In heat maps, the transcriptome signatures were plotted for all categories, including pre-pandemic healthy controls.

For cohort 3, the LC-MO signature was checked in CD14+ monocytes of each individual with LC. Participants were further grouped into Resp-PASC-BHR ($N = 3$) and Resp-PASC ($N = 5$) groups based on their pulmonary function test results, as shown in Supplementary Table 3.

### DGE analysis per cell type

For each of the major cell types (CD14+ monocytes, CD16+ monocytes, CD8+ T cells, NK cells, CD4+ T cells and B cells), DGE analysis was performed for LC samples separately for mild/moderate and severe samples. Comparisons were made against acute COVID-19 samples and against recovered samples using Seurat FindMarkers. Genes upregulated and downregulated in these comparisons with an adjusted $P$ value of <0.05 and log$_2$ (fold change) of >0.8 were considered for each cell-type analysis.

### Pathway analysis per cell type

Pseudobulk of each donor at each time point was calculated, followed by similar comparisons as described in the previous section using DESeq2. GSEA using Hallmark and REACTOME pathways as background was performed using the fgsea R package. Furthermore, immune-related pathways that showed statistical significance in any comparisons were plotted. The whole list of statistically significant pathways resulting from all comparisons is shown in Source Data Fig. 2a and Extended Data Fig. 2a,b.

### Pathway correlations with clinical parameters

For each sample and cell type (CD14+ monocytes, CD8+ T cells and NK cells), we computed pseudobulk gene expression profiles. Subsequently, AUC scores for the selected upregulated immune pathways, as described in Fig. 2c, were calculated in each sample. The AUC scores of these pathways were then correlated with clinical parameters using Spearman correlations. Only significant correlations were plotted.

### Cytokine data analysis

Cytokine measurements for each measured cytokine were log$_2$ transformed. Comparisons and statistical tests against COVID-naive healthy controls for each measured cytokine were performed. The Spearman correlation test was performed to assess the correlation between transformed cytokine measurements and clinical parameters.

### Subclustering analysis of CD8+ T cells, NK cells and monocytes

CD8+ T cells, NK cells and CD14+ monocytes were subsetted separately and reanalyzed. Libraries contributing less than 60 cells were removed, and integration was performed using 'RPCAIntegration' and k.weight as 60. The top 10 principal components were used for UMAP and FindNeighbors calculation. For clustering CD8+ T and NK cells, a resolution of 0.4 was used. For CD14+ monocytes, a resolution of 0.2 was used. An AUC score for Hallmark pathways enriched in pseudobulk analysis was calculated for each cell using the AUCell R package and raw counts of each cell.

### Neighborhood enrichment analysis

MiloR[56] was used for differential neighborhood analysis. The kNNGraph and neighborhoods were calculated with $k = 50$ and $d = 50$. The design matrix included the sampleID, severity_timePoint and recovered or not as covariates. Differential neighborhood tests were calculated for LC[AM] samples (T2, T3, T4 and T5) from different time points against acute COVID-19 samples (AI) or against R[LC]. The resulting differential neighborhoods were annotated based on cell clusters previously obtained for each cell subset. Neighborhoods with a spatial FDR of <0.1 were considered significant.

### Pseudotime and trajectory analysis

A Seurat RNA assay of CD14+ monocytes was used to create a singlecell experiment object using scater. Diffusion maps were calculated using the destiny R package[57]. Average dimensionality was calculated using the find_sigmas function with logCounts of single-cell data. DiffusionMap was calculated using 40 principal components and sigmas calculated in the previous step. The top diffusion components (DCs) were inspected, and DC1 and DC3 were used because DC2 showed sample-dependant bias. Slingshot[58] was used for trajectory calculations. Clustering was calculated using the top 15 DCs and the Mclust package. Clusters with >90% of cells from COVID-naive healthy controls were chosen as the starting clusters for trajectory calculation, resulting in three lineages. The expression of genes involved in key pathways (from pathway enrichment analysis) and selected upregulated genes from the MC4 cluster were plotted against pseudotime values of each cell. Similarly, AUC scores of key pathways calculated per cell were plotted against pseudotime.

### Peak calling and peak-to-gene linkage

Peaks were called for each major cell subset as identified from RNA-based annotations using Macs3 and Ensembl.Db.Hsapiens.v86. Peaks were linked to RNA assay-based gene expression using the LinkPeaks command. Differential peaks within each cluster were calculated using Seurat function FindMarkers with the 'LR' test and nCount_peaks as the latent.variable.

### Transcription factor motif annotation and enrichment

The Jaspar2020 database was used as background for the motif matrix using only the human-specific motif collection. Chromvar was used to calculate transcription factor activity for each cell. Differential transcription factor motif activity for any comparison was calculated using FindMarkers with mean.fxn set to 'rowMeans'. Motif enrichment was assessed by correcting for background peaks using MatchRegionStats.

### Analysis of scRNA-seq datasets (cohort 3)

Once the donor for each cell was assigned after demultiplexing, the doublets were filtered out, and singlets were kept. Seurat version 5.0 was used for downstream analysis. The following filtration criteria were used: 'nCount_RNA < 8,000 and nFeature_RNA < 3,500 and mitochondrial percentage < 20'. RNA integration across libraries was performed using 'RPCAIntegration' and the top 30 dimensions for both clustering and UMAP generation. Multiple resolutions varying from 0.2 to 0.5 were carried out to obtain clusters, and canonical markers were used

for identifying cell subsets (using a combination of known markers and those used by Azimuth celltype.l2) to identify CD14[+] monocytes. Further, these cells were subsetted and reintegrated with 15 principal components, and clustering was performed with a resolution of 0.2. LC-Mo/MC4 AUC scores were calculated for each cell using raw counts.

## Reporting summary

Further information on research design is available in the Nature Portfolio Reporting Summary linked to this article.

## Data availability

Single-cell multiome data were submitted to European Genome–Phenome Archive and are accessible through the following IDs: EGAS50000000142, EGAS50000000143, EGAS0000001215 and EGAS0000001216. Source data are provided with this paper.

## Code availability

Scripts and code are available at github.com/CiiM-Bioinformatics-group/LongCOVID.

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

## Acknowledgements

This project was supported by an ERC Starting Grant 948207 (ModVaccine) to Y.L., the COVID-19-Research Network of the state of Lower Saxony (COFONI) through funding from the Ministry of Science and Culture of Lower Saxony in Germany (14-76403-184) to Y.L. and T.I., the Lower Saxony Center for AI and Causal Methods in Medicine (CAIMed) grant (ZN4257) and German Federal Ministry of Education and Research grants (01EQ2302A, FEDCOV, 031L0318A, AID-PAIS) to Y.L. and US National Institutes of Health grants AG069264, AI147394, HL170961, AI176171 and AG090337 to J.S. S.A.K. was supported by the PRACTIS Clinician Scientist Program, funded by MHH and DFG (DFG ME 3696/3), and by funding from Julitta und Richard Müller Stiftung (COVIDCODE). The COVID-19 biobank of MHH was funded by the Lower Saxony Ministry of Science and Culture.

## Author contributions

Y.L., T.I. and S.K. conceived and designed the study. I.P., S.V., D.v.W., H.L., S.A.K., I.K., A.R.K. and M.M.H. acquired clinical samples and collected clinical data. L.Z., A.A. and A.Z. generated data. S.K. and Q.Z. performed data analysis and investigation. S.K., Y.L. and S.V. coordinated project administration. B.C., A.R.M.K., H.S., D.v.W., H.L., L.P. and C.L. performed experiments. Y.L., T.I., J.S., S.H., A.R.M.K., M.W., M.C. and C.-J.X. provided resources. Y.L., M.G.N and J.S. supervised the study. Y.L. and T.I. acquired funding. S.K. and Y.L. wrote the original paper. All authors reviewed and approved the final paper.

## Funding

## Competing interests

M.G.N. is the scientific founder of Biotrip, Salvina, TTxD and Lemba. MHH has received fees for consultations or lectures from 35Pharma, Acceleron, Actelion, Aerovate, AOP Health, Bayer, Ferrer, Gossamer, Inhibikase, Janssen, Keros, MSD and Novartis. The other authors declare no competing interests.

## Additional information

**Extended data** is available for this paper at https://doi.org/10.1038/s41590-025-02387-1.

**Correspondence and requests for materials** should be addressed to Yang Li.

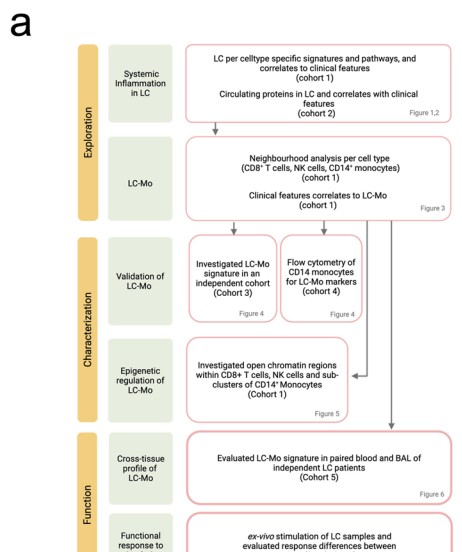

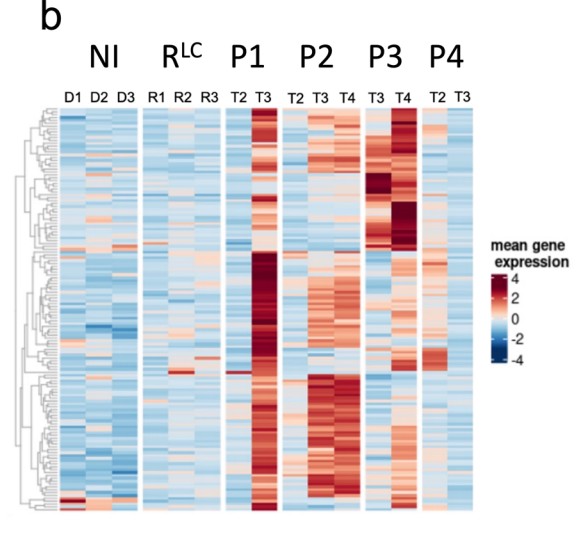

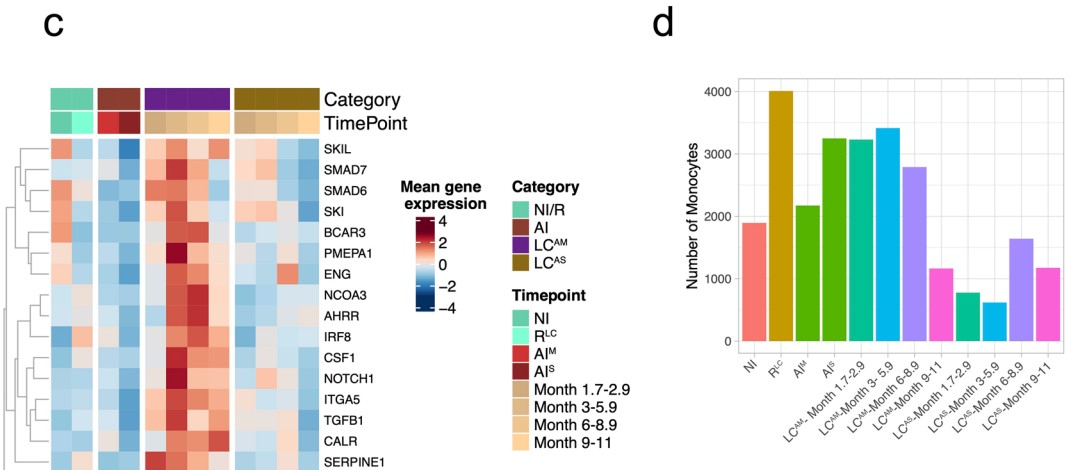

**Extended Data Fig. 1 | LC patients show heterogenous expression of LC signature. a**, Schematic view of analysis flow for the study. **b**, Mean expression of LC signature genes in LC^AM (n = 4) participants with longitudinal time points and NI (n = 3) and R^LC (n = 3) (cohort 1). **c**, Mean expression of LC signature genes showing consistent expression at all timepoints **d**, Number of CD14^+ monocytes in each category at all time points.

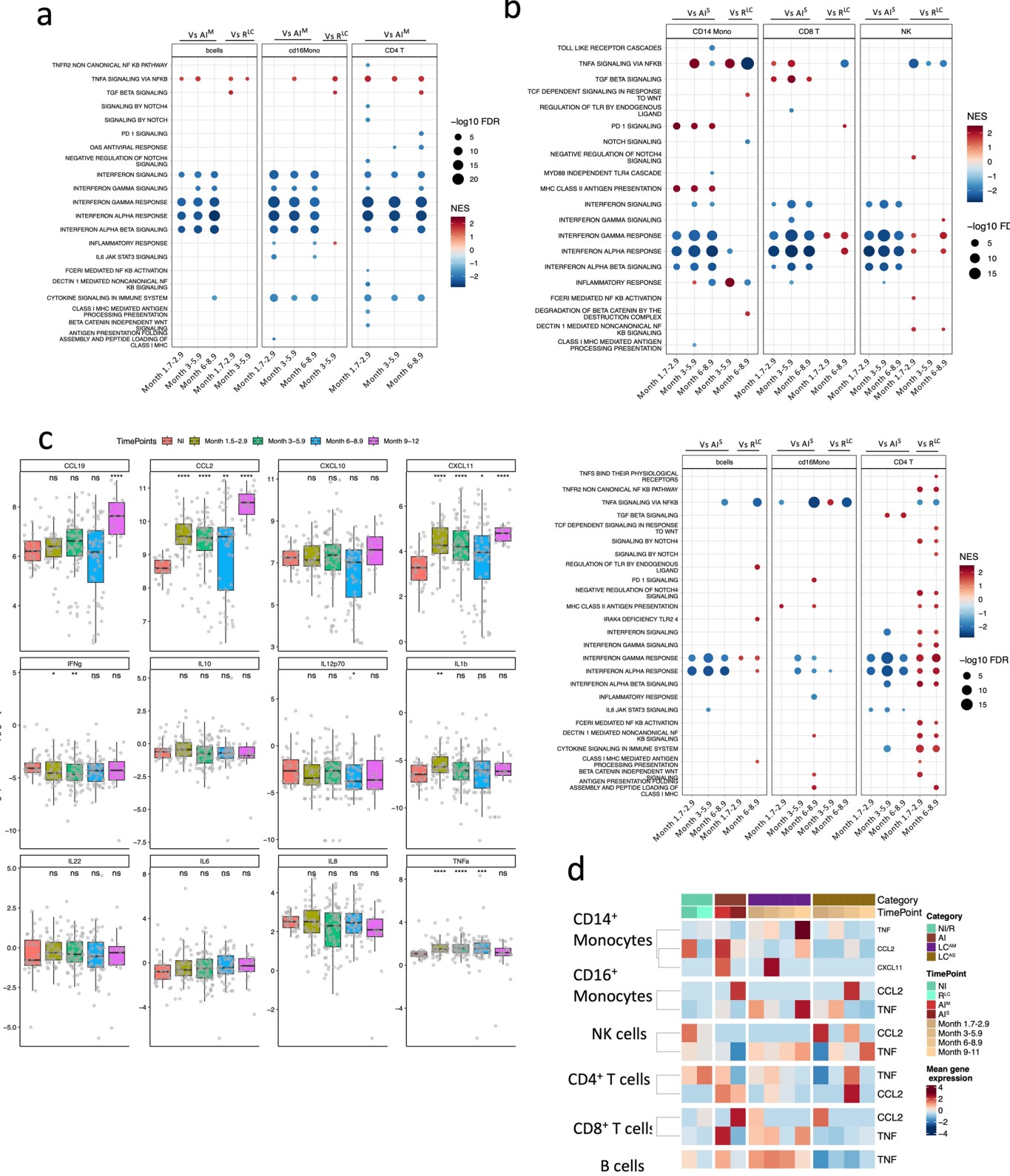

**Extended Data Fig. 2 | LC^AM and LC^AS show different pathways active in LC.**
**a,** Significant GSEA pathways in LC^AM at different time points compared to AI^M or R^LC, in B cells, CD4+ T and CD16+ monocytes **b,** GSEA pathways comparing LC^AS at different time points from for all major cell subsets (cohort 1). Pathways plotted with adj Pval < 0.1 (Kolmogorov-Smirnov-based test with permutation-derived p-values, adjusted using the Benjamini-Hochberg method). **c,** comparing measured cytokine levels (CCL19, IFNg, IL10, IL12p70, IL1b, IL22,

IL6, IL8, IP10) in LC patients at different time points with NI (n = 33), LC: month 1.5-2.9 (n = 51), month 3-5.9 (n = 75) month 6-8.9 (n = 60) month 9-12 (n = 19) (cohort 2). Two-sided Wilcox Rank Sum Test ****: p value < 0.00001, ***: p value < 0.001, **: p value < 0.01, *: p value < 0.05, ns = not significant. Boxplot shows the median (centre), first and third quartiles(bounds) and 1.5 times the interquartile range (whiskers) **d,** Mean mRNA expression of TNF, CCL2 and CXCL11 across all categories (cohort 1).

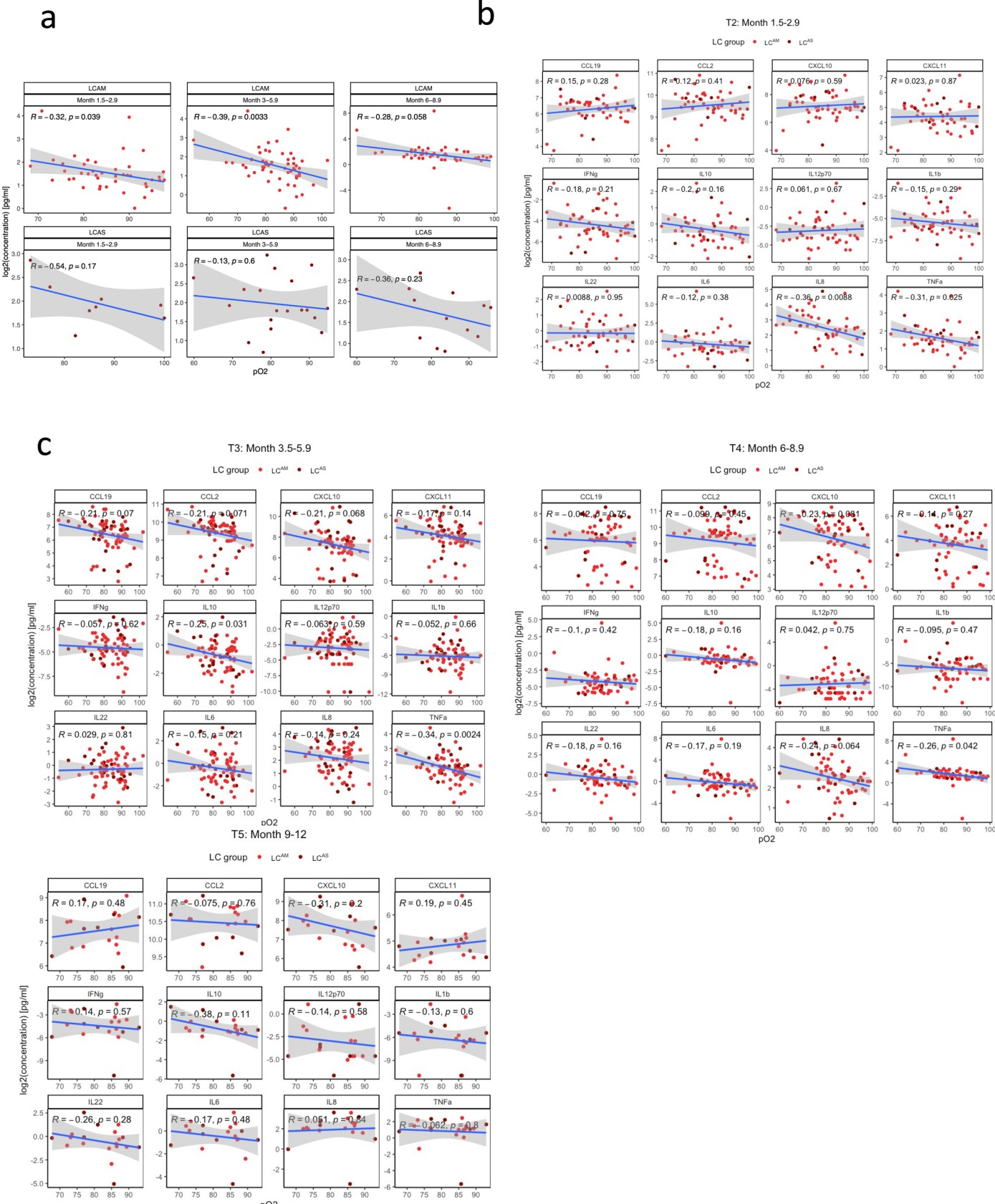

**Extended Data Fig. 3 | TNF significantly negatively correlates with pO2 in LC^AM. a**, TNF correlation with partial pressure O2 calculated separately for LC^AM and LC^AS (cohort 2) at month 1.5-2.9 (LC^AM n = 43, LC^AS n = 8), month 3-5.9 (LC^AM n = 56, LC^AS n = 19) and month 6-8.9 (LC^AM n = 47, LC^AS n = 13) **b** and **c** Correlation of all measured cytokines as for each time point against pO2. LC: month 1.5-2.9

(n = 51), month 3-5.9 (n = 75) month 6-8.9 (n = 60) month 9-12 (n = 19). Spearman correlation P values for Spearman correlation were computed using the exact/permutation-based test. The gray shaded area indicates the 95% confidence interval.

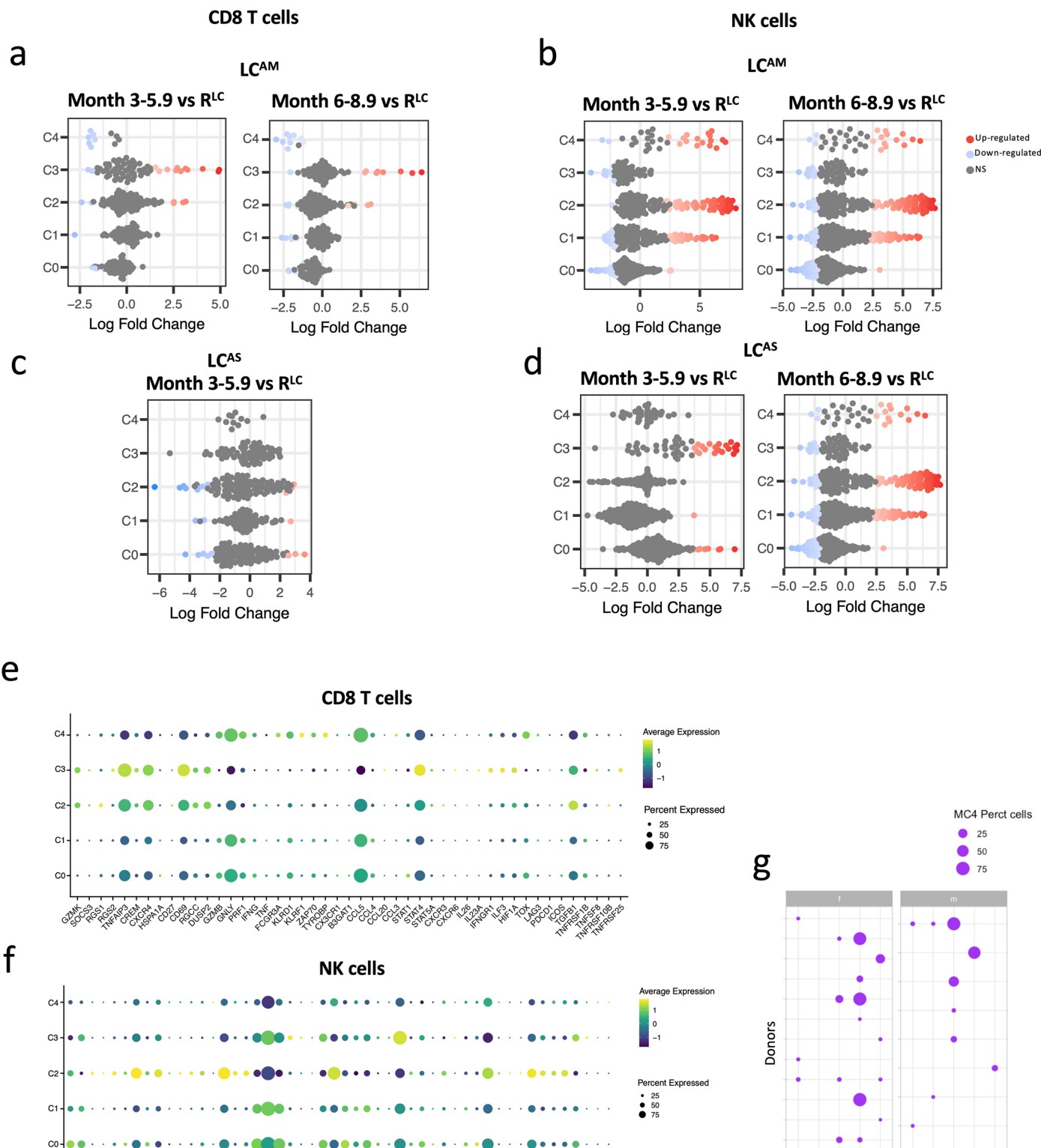

**Extended Data Fig. 4 | LC$^{AM}$ CD8$^+$ T and NK cells show increased abundance of GZMK$^+$ cells. a,b**, Neighbourhood enrichment analysis in LC$^{AM}$ (cohort 1) compared to R$^{LC}$ in CD8$^+$ T and **b**, in NK cells. **c,d**, Neighbourhood abundance enrichment analysis in LC$^{AS}$ compared to R$^{LC}$ in CD8$^+$ T and **d**, in NK cells. Red dots represent increased significant enrichment of neighbourhoods; blue dots represent significantly decreased enrichment. Transcriptional states showing significant enrichment with spatial FDR < 0.1 in red and blue (F-test statistic from the quali-likelihood F-test, graph weighted FDR). **e**, GZMK$^+$ cells signature derived from *Jonnson* et.al[28] shown in gene expression profile within CD8$^+$ T sub-clusters, **f**, within NK sub-clusters. **g**, Proportion of MC4 cells in CD14$^+$ monocytes of LC$^{AM}$ samples from multiple time points stratified on sex.

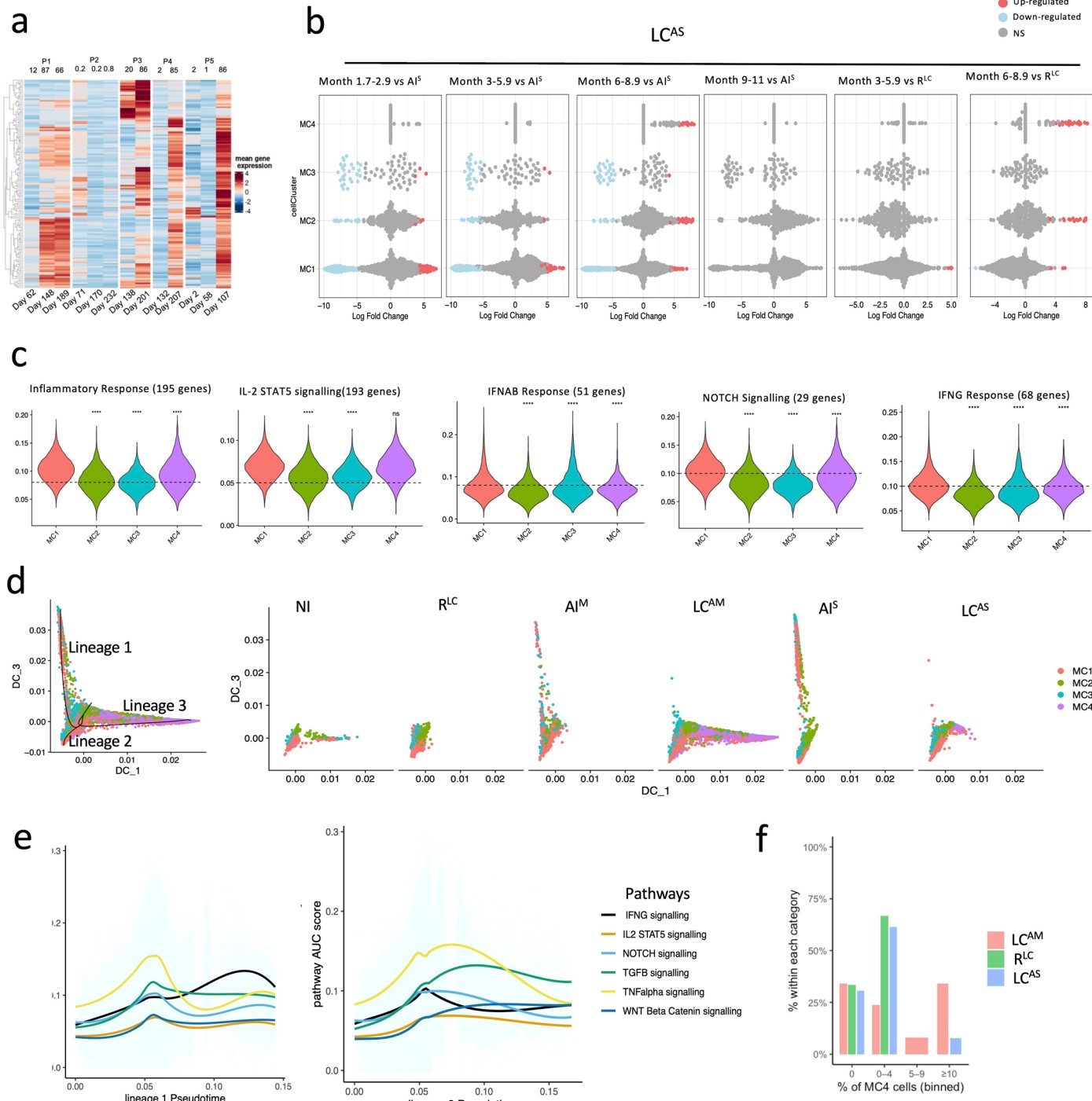

**Extended Data Fig. 5 | LC-Mo show a distinct transcriptional program.**
**a**, Mean gene expression of MC4 signature in CD14⁺ monocytes of LC^AM (n = 5) patients with longitudinal samples in cohort 1, with sampling day of each individual and percentage of MC4-like cells in each sample **b**, Differential neighbourhood enrichment in CD14⁺ monocyte of LC^AS compared to AI^S and R^LC. Red dots represent increased significant enrichment of neighbourhoods; blue dots represent significantly decreased enrichment. Transcriptional states showing significant enrichment with spatial FDR < 0.1 (F-test statistic from

the quali-likelihood F-test, graph weighted FDR) **c**, Pathway AUC scores within monocyte clusters. Two-sided Wilcox Rank Sum Test ****: pvalue < 0.00001, ***: pvalue < 0.001, **: pvalue < 0.01, *: pvalue < 0.05, NS = Non-significant **d-e**, Distinct trajectories of CD14⁺ monocytes derived from pseudotime **d**, Diffusion map of all CD14⁺ monocytes with predicted lineages (left) cells split on categories of samples (right) **e**, AUC scores of pathways arranged across lineage 1 and lineage 3. **f**, Percentage of MC4 cells within CD14⁺ monocytes, categorised and binned.

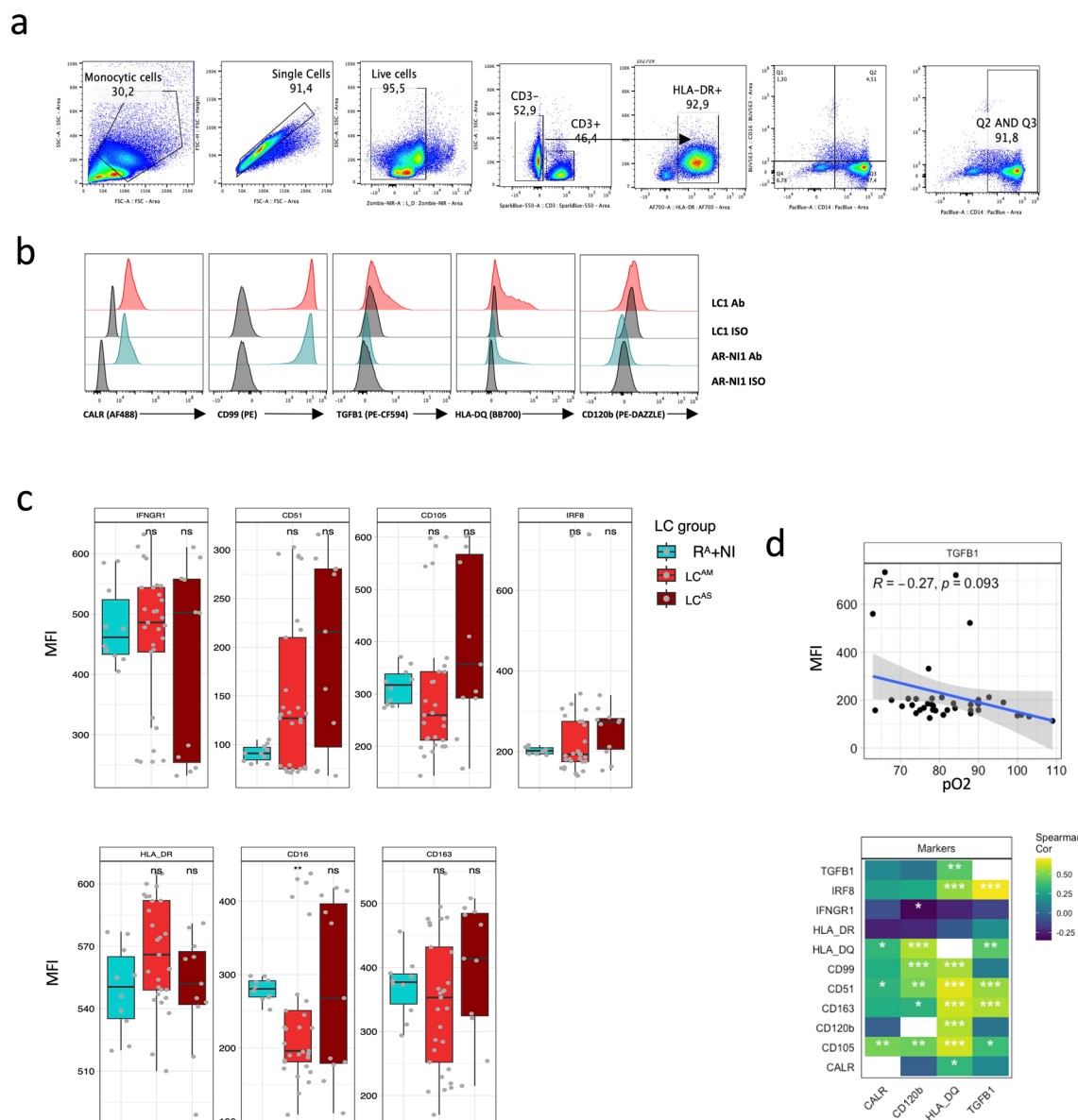

**Extended Data Fig. 6 | Flow cytometry of CD14⁺ monocytes shows higher expression of LC-Mo in LC patients. a**, Gating strategy implemented to identify Monocyte subsets in cohort 4. **b**, Histograms for the flow cytometry analysis of CALR, CD99, TGFB1, HLA-DQ, and CD120b surface expression enriched for all CD14⁺ monocytes from R^A+NI compared to LC. R^A+NI, LC antibody, and isotype-stained cells are shown in blue, red, and black, respectively. Shown is a representative donor of biological replicates with similar results **c**, Comparison of Median Fluorescence Intensity (MFI) of measured markers (IFNGR1, CD51, CD105, IRF8, HLA-DR, CD16 and CD163) in CD14⁺ monocytes. R^A+NI (n = 10),

LC^AM (n = 29), LC^AS (n = 11). P-values calculated using two-sided Wilcox Rank Sum Test * **: p value < 0.00001, ***: p value < 0.001, **: p value < 0.01, *: p value < 0.05. Boxplot shows the median (centre), first and third quartiles(bounds) and 1.5 times the interquartile range (whiskers) **d**, TGFB1 MFI correlation with blood pO2 (top), the gray shaded area indicates the 95% confidence interval. correlation of significant markers in LC patients among each other (bottom). Spearman correlation P values were computed using a t-distribution approximation applied to rank-transformed data. ***: p value < 0.001, **: p value < 0.01, *: p value < 0.05.

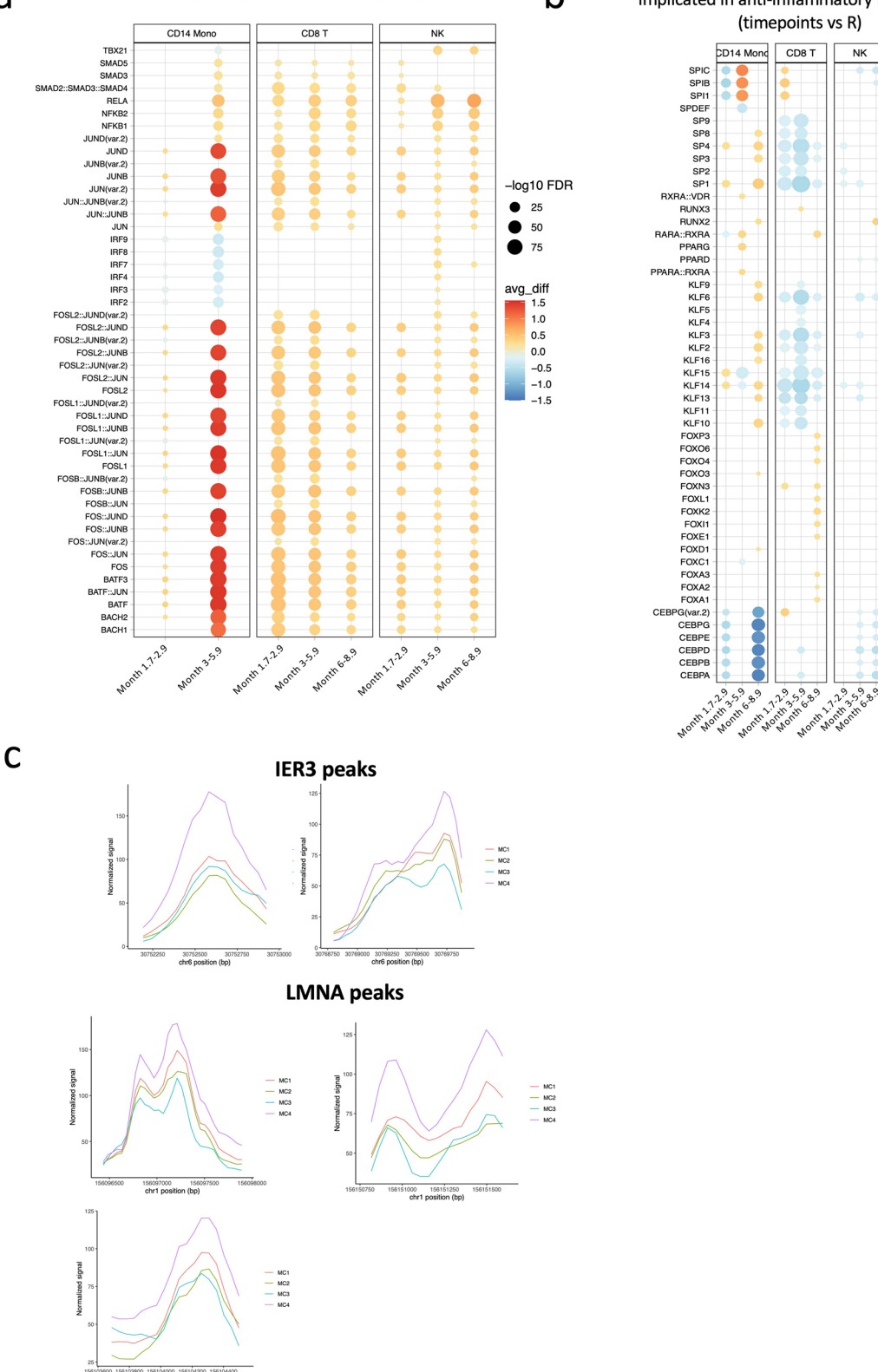

**Extended Data Fig. 7 | CD8⁺T and NK cells show persistent increased AP-1 accessibility in LCᴬᴹ. a,b,** ChromVar motif accessibility enrichments (cohort 1) of AP-1 family in CD14⁺ monocytes of LCᴬᴹ, CD8⁺ T cells and NK cells in comparison to Rᴸᶜ. Plotted TF motifs are significant with an adjusted P-value < 0.05 (two-sided Wilcox Rank Sum Test, adjusted using the Benjamini-Hochberg method) **c,** Coverage plot of highlighted peaks in Fig. 5 f and g, showing normalized signal in each of MC1-4 clusters.

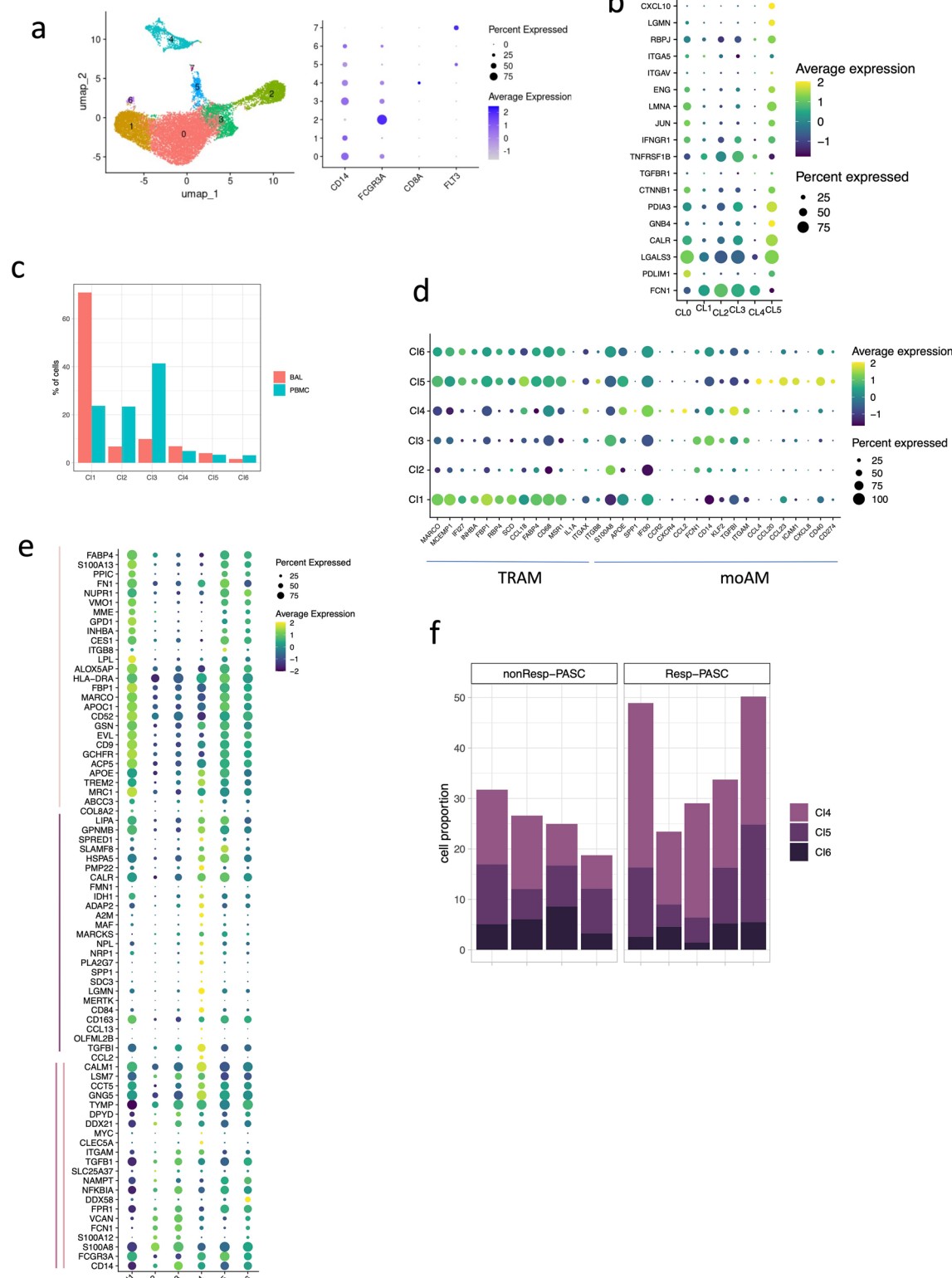

**Extended Data Fig. 8 | LC-Mo like macrophages show pro-fibrotic signature. a**, Blood (cohort 5) CD14⁺, CD16⁺ Monocytes and monocyte derived DCs from independent cohort (GSE263817)[23] **b**, Gene expression profile of clusters showing LC-Mo/MC4 like genes in cluster5. (**c**) Proportion of blood and BAL Monocyte/ Macrophage cells in each of the cluster as described in Fig. 6d. **d**, Gene expression as described in (GSE263817)[23] for Tissue resident Alveolar Macrophages (TRAM) and monocyte derived Macrophages(moAM) **e**, Gene expression profile derived from *Wendisch* et al. [33] of CI1-CI6 clsuters as shown in Fig. 6d **f**, Proportion of cells in either CI4, CI5 or CI6 clusters from each donor classified into PASC categories.

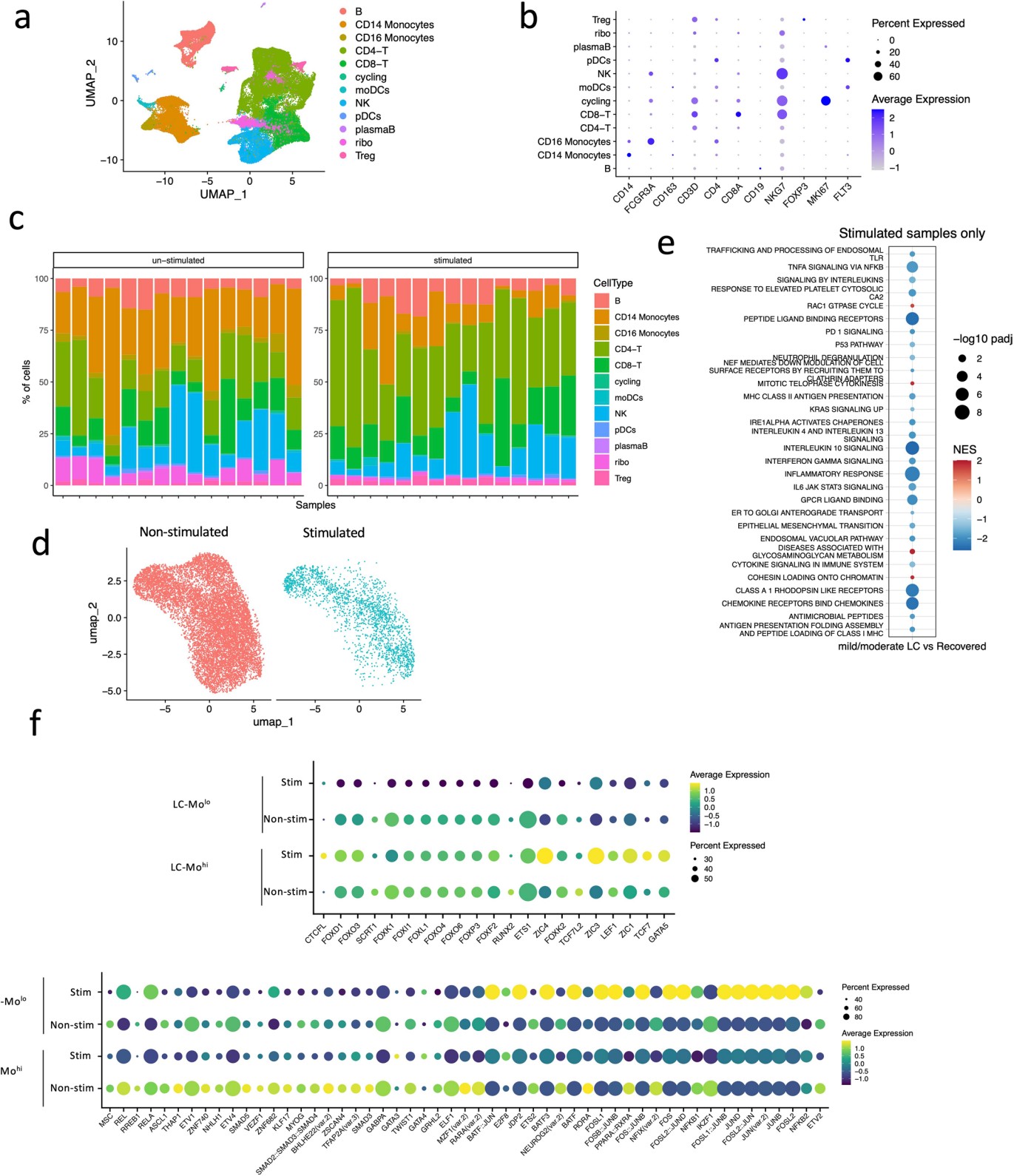

**Extended Data Fig. 9 | LC-Mo$^{hi}$ patients show increased FOXO family accessibility after stimulation. a,** UMAP of all cells from non-stimulated and stimulated samples (cohort 1 subset) **b,** Canonical markers identifying each of the cell subsets as in a **c,** Percentage of cell subsets per sample from both stimulated and -non-stimulated samples. **d** UMAP of stimulated and non-stimulated cells (Myeloid only) **e,** GSEA pathway enrichment in comparison of stimulated LC$^{AM}$ vs stimulated R$^{LC}$ **f,** Chromvar TF accessibility in LC-Mo$^{hi}$ and LC-Mo$^{lo}$ before and after stimulation. TF motifs are significant with adj Pval < 0.05 (two-sided Wilcox Rank Sum Test, using the Benjamini-Hochberg method).

# Reporting Summary

## Statistics

For all statistical analyses, confirm that the following items are present in the figure legend, table legend, main text, or Methods section.

| n/a | Confirmed | |
|---|---|---|
| ☐ | ☒ | The exact sample size (*n*) for each experimental group/condition, given as a discrete number and unit of measurement |
| ☐ | ☒ | A statement on whether measurements were taken from distinct samples or whether the same sample was measured repeatedly |
| ☐ | ☒ | The statistical test(s) used AND whether they are one- or two-sided<br>*Only common tests should be described solely by name; describe more complex techniques in the Methods section.* |
| ☐ | ☒ | A description of all covariates tested |
| ☐ | ☒ | A description of any assumptions or corrections, such as tests of normality and adjustment for multiple comparisons |
| ☐ | ☒ | A full description of the statistical parameters including central tendency (e.g. means) or other basic estimates (e.g. regression coefficient) AND variation (e.g. standard deviation) or associated estimates of uncertainty (e.g. confidence intervals) |
| ☐ | ☒ | For null hypothesis testing, the test statistic (e.g. *F*, *t*, *r*) with confidence intervals, effect sizes, degrees of freedom and *P* value noted<br>*Give P values as exact values whenever suitable.* |
| ☒ | ☐ | For Bayesian analysis, information on the choice of priors and Markov chain Monte Carlo settings |
| ☒ | ☐ | For hierarchical and complex designs, identification of the appropriate level for tests and full reporting of outcomes |
| ☐ | ☒ | Estimates of effect sizes (e.g. Cohen's *d*, Pearson's *r*), indicating how they were calculated |

*Our web collection on statistics for biologists contains articles on many of the points above.*

## Software and code

Policy information about availability of computer code

| Data collection | A detailed description of the data and sample collection can be found in the methods section of the manuscript.<br>EDTA blood was collected from patients which were admitted to the hospital due to COVID-19, as well as from ambulatory patients after SARS-COV-2-infection showing Post-COVID19 symptoms (at least 4 weeks after acute COVID-19) at the Hannover Medical school (MHH) or the Siloah hospital. PBMCs were isolated from whole blood using Ficoll gradient centrifugation. 10x Genomics Chromium Next GEM Single Cell Multiome ATAC + Gene Expression protocol and single cell GEM-X 3` Reagent Kits v4 were used to generate single cell multiome data. Sequencing was performed using the Illumina NovaSeq 6000 platform.<br>The Quanterix HD SP-X Imaging and Analysis System™ was used to measure the plasma samples. Human Corplex cytokine panel 1 10-Plex array was used. |
|---|---|
| Data analysis | For data analysis, we employed the statistical language R(version > 4).A detailed description of the analysis can be found in the methods section of the manuscript. All code for the analysis is made freely available on github.com/CiiM-Bioinformatics-group/LongCOVID. |

For manuscripts utilizing custom algorithms or software that are central to the research but not yet described in published literature, software must be made available to editors and reviewers. We strongly encourage code deposition in a community repository (e.g. GitHub). See the Nature Portfolio guidelines for submitting code & software for further information.

## Data

Policy information about availability of data

All manuscripts must include a data availability statement. This statement should provide the following information, where applicable:

- Accession codes, unique identifiers, or web links for publicly available datasets
- A description of any restrictions on data availability
- For clinical datasets or third party data, please ensure that the statement adheres to our policy

Single cell multi-ome data and  scRNA-seq data was submitted to EGA and is accessible through following IDs:
EGAS50000000142, EGAS50000000143, EGAS0000001215 and EGAS0000001216

## Research involving human participants, their data, or biological material

Policy information about studies with human participants or human data. See also policy information about sex, gender (identity/presentation), and sexual orientation and race, ethnicity and racism.

| | |
|---|---|
| Reporting on sex and gender | Our study included patients of both sexes and therefore applies to both sexes. Sexes were self-reported and later confirmed by DNA genotyping. Single cell dataset from both sexes were equally represented in the datasets. |
| Reporting on race, ethnicity, or other socially relevant groupings | *Please specify the socially constructed or socially relevant categorization variable(s) used in your manuscript and explain why they were used. Please note that such variables should not be used as proxies for other socially constructed/relevant variables (for example, race or ethnicity should not be used as a proxy for socioeconomic status).*<br>*Provide clear definitions of the relevant terms used, how they were provided (by the participants/respondents, the researchers, or third parties), and the method(s) used to classify people into the different categories (e.g. self-report, census or administrative data, social media data, etc.)*<br>*Please provide details about how you controlled for confounding variables in your analyses.* |
| Population characteristics | The only relevant population characteristic is the past history of COVID -19 infection. All patients were tested PCR negative at the time of visit and sample collection. |
| Recruitment | Patients reporting to the pneumological outpatient clinic at Hannover Medical School (MHH, Hannover, Germany) with symptoms such as headaches, dyspnoea and fatigue were recruited based on established LC criteria, 1) Symptoms that persist after acute COVID-19 or its treatment. 2) New symptoms that appear after the end of the acute COVID-19 phase, and can be a consequence of the SARS-CoV-2 infection and 3) Worsening of a pre-existing illness as a result of a SAR-CoV-2 infection. The diagnosis was made in accordance with the German S1 guidelines and the Delphi Consensus Criteria of post COVID-19. The cohort included individuals with all severity of acute COVID-19 (WHO score 1-9). |
| Ethics oversight | The ethics committee of the Hannover Medical School (MHH) approved the sample collection and analyses (ethics vote 9001_BO_K).Informed consent was obtained from all individual participants included in the study. |

Note that full information on the approval of the study protocol must also be provided in the manuscript.

# Field-specific reporting

Please select the one below that is the best fit for your research. If you are not sure, read the appropriate sections before making your selection.

☒ Life sciences ☐ Behavioural & social sciences ☐ Ecological, evolutionary & environmental sciences

For a reference copy of the document with all sections, see nature.com/documents/nr-reporting-summary-flat.pdf

# Life sciences study design

All studies must disclose on these points even when the disclosure is negative.

| | |
|---|---|
| Sample size | Samples were chosen to balance the age, gender and acute COVID-19 severity for samples collected for different time points. Where possible, longitudinal samples from same patients were used. |
| Data exclusions | Single cells with low quality, high mitochondrial gene expression and doublets were excluded from the data analysis of single cell multiome analysis of discovery cohort. |
| Replication | Findings from single cell data of cohort 1 were vaidated with findings from another single cell study of independent cohort 3 and with  publicly available datasets. |
| Randomization | Samples were randomized to include random 4 different donors for each pool during 10X experiments. Additionally, sample timepoint was randomised along with different donors before 10X experiments. |
| Blinding | The investigators performed the stratification of the long covid samples to identify differences in long covid disease. Therefore, blinding to group allocations was not possible for the further analysis. |

# Reporting for specific materials, systems and methods

We require information from authors about some types of materials, experimental systems and methods used in many studies. Here, indicate whether each material, system or method listed is relevant to your study. If you are not sure if a list item applies to your research, read the appropriate section before selecting a response.

## Materials & experimental systems

| n/a | Involved in the study |
|-----|----------------------|
| ☐ | ☒ Antibodies |
| ☒ | ☐ Eukaryotic cell lines |
| ☒ | ☐ Palaeontology and archaeology |
| ☒ | ☐ Animals and other organisms |
| ☐ | ☒ Clinical data |
| ☒ | ☐ Dual use research of concern |
| ☒ | ☐ Plants |

## Methods

| n/a | Involved in the study |
|-----|----------------------|
| ☒ | ☐ ChIP-seq |
| ☐ | ☒ Flow cytometry |
| ☒ | ☐ MRI-based neuroimaging |

## Antibodies

| | |
|---|---|
| Antibodies used | For Human:<br>CD3 SparkBlue SK7 BioLegend 344852<br>CD14  PacBlue 63D3 BioLegend 367122<br>CD16 BUV563 3G8 BD 568289<br>CD51 APC NKI-M9 BioLegend 327913<br>CD99 PE hec2 BioLegend 398205<br>CD105 BUV421 43A3 BioLegend 323219<br>CD120b PE-DAZZLE 3G702 BioLegend 358413<br>CD163 FITC/PE-CY7 GHI/61 BioLegend 333618/2268070<br>CD206 APC-CY7/PE-CY7 15-2 BioLegend 321120/321124<br>CALR purified  abcam ab2907<br> AF488  invitrogen<br>HLA-DQ BB700 Tu169 BD 745976<br>HLA-DR AF700 L243 BioLegend 307626<br>IFNGR1 purified  ab154400 ab154400<br> AF568  invitrogen<br>IRF8 PE REA516 Miltenyi 130-122-927<br>TGFB1 PE-CF594 TW4-9E7 BD 562422 |
| Validation | All used antibodies are validated/ quality control tested for the analysis of human cells by flow cytometry according to the manufacturer's information. |

## Clinical data

Policy information about clinical studies

All manuscripts should comply with the ICMJE guidelines for publication of clinical research and a completed CONSORT checklist must be included with all submissions.

| | |
|---|---|
| Clinical trial registration | *Provide the trial registration number from ClinicalTrials.gov or an equivalent agency.* |
| Study protocol | *Note where the full trial protocol can be accessed OR if not available, explain why.* |
| Data collection | *Describe the settings and locales of data collection, noting the time periods of recruitment and data collection.* |
| Outcomes | *Describe how you pre-defined primary and secondary outcome measures and how you assessed these measures.* |

## Plants

| | |
|---|---|
| Seed stocks | *Report on the source of all seed stocks or other plant material used. If applicable, state the seed stock centre and catalogue number. If plant specimens were collected from the field, describe the collection location, date and sampling procedures.* |
| Novel plant genotypes | *Describe the methods by which all novel plant genotypes were produced. This includes those generated by transgenic approaches, gene editing, chemical/radiation-based mutagenesis and hybridization. For transgenic lines, describe the transformation method, the number of independent lines analyzed and the generation upon which experiments were performed. For gene-edited lines, describe the editor used, the endogenous sequence targeted for editing, the targeting guide RNA sequence (if applicable) and how the editor was applied.* |
| Authentication | *Describe any authentication procedures for each seed stock used or novel genotype generated. Describe any experiments used to assess the effect of a mutation and, where applicable, how potential secondary effects (e.g. second site T-DNA insertions, mosiacism, off-target gene editing) were examined.* |

## Flow Cytometry

### Plots

Confirm that:

☒ The axis labels state the marker and fluorochrome used (e.g. CD4-FITC).

☒ The axis scales are clearly visible. Include numbers along axes only for bottom left plot of group (a 'group' is an analysis of identical markers).

☒ All plots are contour plots with outliers or pseudocolor plots.

☒ A numerical value for number of cells or percentage (with statistics) is provided.

### Methodology

| | |
|---|---|
| Sample preparation | Cryopeserved PBMCs were thawed, stained with the Zombie NIR™ Fixable Viability Kit (Biolegend) at room temperature (RT) in PBS for 15min. Unspecific immunolabeling conferred by Fc receptor binding was blocked by the addition of 10% Gamunex solution (Grifols Deutschland GmbH, Frankfurt am Main, Germany). Surface marker immunolabeling was performed in cell staining buffer (PBS, BSA, EDTA) and Brilliant Stain Buffer (BD), overnight at 4 ∘C with anti-human CD3, CD14, CD16, HLA-DR, CD163 and CD206, CD51, CD99, CD105, CD120b and HLA-DQ antibodies. For intracellular staining, cells were fixed and permeabilized (BD) for 30min at RT, immunolabeling of intracellular markers was performed for 30min in Permwash buffer (BD) at 4°C with anti-human primary CALR, IFNGR1, TGFB1 and IRF8 antibodies. Next cells were immunolabeled with the secondary antibody AF488 and AF568 for 30min in Permwash buffer (BD) at 4°C to label the unconjugated antibodies CALR and IFNGR1, respectively. All donors were also immunolabeled with the correspondent isotype controls for the used antibodies. Cells were washed with PBS. |
| Instrument | The samples were acquired on a Sony spectral analyzer (ID7000, Sony). |
| Software | The samples were analyzed with the FlowJo software (10.10.0 Tree Star). |
| Cell population abundance | Monocytes correspond to roughly 5-10% of the PBMC samples, with a loss of 30% during freeze-thawing process. Purity of monocytes was determined by the expression of CD14+, CD16+ and HLA-DR surface markers. |
| Gating strategy | SSC-A/FSC-A gate to select monocytic cells -> FSC-H/FSC-A gate to select single cells -> Live/Dead gate to select viable cells -> CD3- gate to exclude T cells -> HLA-DR gate to exclude NK cells -> CD14+/CD16+ gate to select classical (CD14+), non-classical (CD16+), and intermediate (CD14+/CD16+) (see also Supplementary Fig. 5c). |

☒ Tick this box to confirm that a figure exemplifying the gating strategy is provided in the Supplementary Information.

