## [Peer Review File · Nature Immunology]

A distinct monocyte transcriptional state links systemic immune dysregulation to pulmonary impairment in long COVID

Corresponding Author: Professor Yang Li

Decision Letter:

24th Apr 2025

Dear Dr. Li,

Your Article, "A distinct monocyte cellular state links systemic immune dysregulation to pulmonary impairment in long COVID" has now been seen by 2 referees. While we find your work of considerable potential interest, the reviewers have raised substantial concerns that must be addressed. As such, we cannot accept the current version of the manuscript for publication, but would be happy to consider a revised version that addresses these concerns, as long as novelty is not compromised in the interim.

Please revise the manuscript to address all issues raised by the referees. At resubmission, please include a point-by-point "Response to referees" detailing how you have addressed each referee comment (please specify the page and figure number where the new data can be found in the revised manuscript and please highlight the changes in the manuscript as well). This response will be sent back to the referees along with the revised manuscript.

In addition, please include a revised version of any required reporting checklist. It will be available to referees (and, potentially, statisticians) to aid in their evaluation if the manuscript goes back for peer review. A revised checklist is essential for re-review of the paper. The Reporting Summary can be found here:
<https://www.nature.com/documents/nr-reporting-summary.pdf>

- that unprocessed scans are clearly labelled and match the gels and western blots presented in figures.
- that control panels for gels and western blots are appropriately described as loading or sample processing controls
- all images in the paper are checked for duplication of panels and for splicing of gel lanes.

Extended Data figures and tables are online-only (appearing in the online PDF and full-text HTML version of the paper), peer-reviewed display items that provide essential background to the Article but are not included in the printed version of the paper due to space constraints or being of interest only to a few specialists. A maximum of ten Extended Data display items (figures and tables) is typically permitted. When re-submitting your manuscript, please ensure that any supplementary figures and tables that are more critical to the manuscript's conclusions are converted to Extended data to increase these data's visibility.

Link Redacted

Note: This URL links to your confidential home page and associated information about manuscripts you may have

submitted, or that you are reviewing for us. If you wish to forward this email to co-authors, please delete the link to your homepage.

We hope to receive a suitably revised manuscript within 6 months. If you cannot send it within this time, please let us know. We will be happy to consider your revision so long as nothing similar has been accepted for publication at Nature Immunology or published elsewhere.

Nature Immunology is committed to improving transparency in authorship. As part of our efforts in this direction, we are now requesting that all authors identified as 'corresponding author' on published papers create and link their Open Researcher and Contributor Identifier (ORCID) with their account on the Manuscript Tracking System (MTS), prior to acceptance. ORCID helps the scientific community achieve unambiguous attribution of all scholarly contributions. You can create and link your ORCID from the home page of the MTS by clicking on 'Modify my Springer Nature account'. For more information please visit please visit www.springernature.com/orcid.

Thank you for the opportunity to review your work.

Sincerely,

Ioana Staicu, Ph.D.
Senior Editor
Nature Immunology

Tel: 212-726-9207
Fax: 212-696-9752
www.nature.com/ni

Reviewers' Comments:

Reviewer #1 (Remarks to the Author):

In the paper entitled ""A distinct monocyte cellular state links systemic immune dysregulation to pulmonary impairment in long COVID" (ref. num. NI-A39964-T), Li and colleagues perform a large number of transcriptomal and functional studies on peripheral blood monocytes from a large number of patients with long Covid. The paper uses complementary techniques to derive a signature for cells that are enriched in patients with long Covid, and suggests that signatures for patients who had mild/moderate acute disease may be different than ones from patients that had severe acute disease.

Over all the findings of the paper are interesting and should add to our knowledge regarding the pathogenesis of long Covid. The paper has a number of strong elements and has potential for impact. However, there are a number of limitations that need to be addressed.

Major comments.

1. The paper is difficult to follow. In part, this arises from the overabundance of data and lack of focus. For example, the first part of the paper delves into differences observed over time in long COVID patients and further stratifies these differences based on initial severity of COVID infection. This seems like it could be a major focus / message of the paper. However, as the paper progresses this theme is lost entirely, and a new focus on respiratory vs non respiratory long COVID phenotypes emerges. The focus of the paper is further diluted by large paragraphs in the results section that refer to supplementary figures that don't always relate to the core theme. As a result, the main message of the story becomes convoluted. While it is important to acknowledge the abundance of data and analysis that are included in the paper, much of it detracts from the story rather than adding.

2. A potential strength of the paper is the richness of the clinical cohort. However, details about the cohort are not described in sufficient detail to enable the reader to follow key elements in the manuscript. This weighs down the paper. Items to consider include the following.

- a. A Table describing the cohort should appear in the main body (rather than the supplement)
- b. Criteria used to diagnose long Covid should be clearly stated in the text. (Of note, the only reference that speaks to long Covid diagnosis is #21. I could only find the text in German. This limits a large portion of the general readership)
- c. Of the patients diagnosed with long-COVID what domains are the ones most impacted? I realize this information can be difficult to gather. However, if this can be achieved it would help the reader contextualize which organs / physical states are affected in the long-Covid cohort. This is relevant since later in the paper, data from an independent cohort with respiratory vs non-respiratory long-Covid are analyzed.
- d. Mean values for PaO₂ and pulmonary physiology parameters fall in the normal range for all groups. While MMRC scores appear to be elevated – these are subjective and can be elevated in patients without lung disease (such as those with chronic fatigue syndrome). Is there any objective evidence that the long covid cohort has persistent lung disease?

e. How are recovered patients defined? Did these patients never develop long Covid? Or did they have LC and then recover?

f. Were PaO₂, PFTs, FAS and MMRC measured at the same time PBMCs were drawn?

3. Patients with long COVID are grouped based on the initial severity of disease and timing from COVID onset. While this could be considered a strength of the paper, it also raises major concerns.

a. Separating the long covid patients into multiple groups (N=8) reduces statistical power and creates problems with multiple comparisons. Accordingly, it is absolutely essential to provide a strong rationale for breaking patients into groups based on:

a) initial severity and b) time from infection.

b. A major finding of the paper is an abnormal gene signature in long Covid patients that had mild/moderate acute disease. Importantly, this abnormal signature is primarily evident in two time points (Day 101-170, Day 191-251) as illustrated in Fig2b. The signature is not present at the earlier time interval or the final time interval. It is also not present in patients that had severe acute disease. This raises the question as to whether this is a true phenomenon. This is a major point that must be addressed.

c. Comparisons are made between long-COVID patients and recovered patients are made for the different time points (i.e. Fig 2c, multiple supplements). Are these comparisons made for recovered patients as a whole, or are the comparisons made within timepoints (i.e. Long Covid for T3 vs Recovered for T3). The number of recovered patients at each timepoint are quite small.

d. How were intervals chosen for T2-T5? These seem arbitrary and based on convenience rather than a biologic rationale.

e. Early and late time intervals (T2, T5) are missing for comparisons of moderate LC vs recovered in Fig 2c and Fig 3E. This gives the appearance that data are “cherry picked” since these early and late time points have less impressive gene signatures (see comment 2b above).

f. Levels of plasma cytokines (Fig2D) do not appear to exhibit the same kinetic as suggested by pseudobulk analysis. Since plasma is from a larger cohort, this reinforces concerns raised in point 2b above. Although it is not implicitly stated, it is implied that peripheral blood leukocytes are the source of the cytokines. Accordingly, if analysis needs to be broken down by time it would be useful to show gene expression for TNF, CCL2, CXCL11 from PBMCs used to generate Fig 2b, c.

4. It seems inappropriate to include recovered patients in statistical analysis for figure 2f. By definition, these patients should have low FAS scores. If gene expression has “normalized” in these patients as part of recovery they will skew the data.

5. Pseudotime trajectory analysis adds little to the paper. I recommend removing it. The method “forces” a trajectory whether one is present or not. Moreover, it is not true reflection of lineage tracing since monocyte survival in the circulation is limited to a few days. In comparison, timecourses used for the analysis are on the order of many months.

6. Experiments that reanalyze published data from LC patients are both confusing and confounded by lack of healthy controls as a reference to show whether cells are “pro-inflammatory” or “pro-fibrotic.” The strength of the published data arises from patients with pulmonary symptoms versus non-pulmonary symptoms. If these data are to be leveraged, then the author’s primary dataset should also incorporate respiratory versus non-pulmonary phenotyping. The ideal analysis would be to derive a LC-CS signature from the author’s dataset and to demonstrate whether this gene signature maps on to PBMCs from the published dataset for respiratory vs non-respiratory phenotypes. This would help to confirm the author’s results and provide a firm link to the first 5 figures of the paper. In it’s current form, the link is tenuous.

Minor

Line 116. Text is missing from the sentence that ends with “characterized by elevated.”

Line 204 states “ CD14+ monocytes in these patients showed elevated ‘PD1 signalling’, suggestive of a low phagocytotic state...”. This is an over interpretation. PD1 is involved in a number of functions in monocytes (including cell death). The role of PD1 in phagocytosis has primarily been suggested in tumor associated macrophages.

The Y axis on Figure 1D is confusing. Have data values been log-transformed? Would it be clearer to display the data using a log scale?

Flow cytometry provides a nice companion to monocyte transcriptional phenotyping. In the supplemental figures gating strategies should be shown. In addition, it will be important to show data for some of the trickier markers (i.e. TGFb1) along with their isotypes. Lastly, it will be important to confirm that isotypes were used as fluorescence minus one (FMO) fashion in which all other markers received true Abs.

Reviewer #2 (Remarks to the Author):

In this manuscript, Kumar et al. present the results of a multiomics profiling study performed on blood samples from a prospective cohort of patients with acute COVID-19 or sequelae (“Long-COVID” or PASC), fully recovered patients, or healthy controls, in an effort to uncover features and potential mechanisms of immunopathology in PASC. They performed single cell multiomics profiling using single nuclei RNA sequencing, scATAC-Seq, multi color flow cytometry on PBMC and plasma cytokine profiling. They additionally analyzed scRNA-Seq data from an external cohort of patients with PASC. Here, they integrated scRNA-Seq data from PBMC and bronchoalveolar lavage (BAL) cells.

The main finding is a distinct transcriptional profile in PBMC, particularly in CD14+ monocytes, in patients with PASC after mild to moderate disease during the acute stage of the infection, compared to acute disease or recovered patients. PBMC from patients with PASC after mild to moderate acute COVID-19 displayed an enrichment of genes associated with TGFbeta- and WNT-Catenin signaling. The authors describe a cellular substate in CD14 monocytes that was positively correlated with participants' fatigue score and negatively correlated with O2 saturation. Moreover, they describe a similarity of this monocyte state (MC4) with BAL macrophages (from PASC patients) that express profibrotic signatures, indicating that this monocyte state may be mechanistically linked to fibrotic responses in patients with respiratory symptoms and PASC. Overall, this is a complex study that offers interesting insights into transcriptional states in PASC patients. Specifically, distinguishing between PASC patients with acute-to-moderate initial disease and those with initially severe disease is important, since the clinical phenotype and the underlying pathophysiology are likely distinct. However, the study contains a number of inconsistencies and shortcomings, which compromise the overall interpretation and the potential impact of the study. Details are listed below.

Specific comments:

1. The authors measured samples from a complex cohort, with both longitudinal and cross-sectional sampling, and participants with distinct outcomes, and controls. Figure 1 attempts to depict the samples from each time point, but it fails to provide a sufficient overview of the composition of the cohort. Suppl. Table 1 adds some information, but the overall description of the cohort is insufficient, and additional information on which samples were taken from the same individuals etc. should be added.
2. Some notes on the cohort. Some groups differ in size (e.g. n=28 PASC_mild/moderate vs n=8 recovered) and in baseline characteristics (e.g. avg. age 39 yrs in recovered vs. 52 yrs in recovered, and male:female ratio of 2 in recovered and 0.58 in PASC_mild/moderate), making it difficult to clearly attribute transcriptomic differences to the clinical outcome, as opposed to baseline differences. It is also unclear, which participants contributed several samples and how the cohorts were composed at the different time points with regards to demographic baseline parameters. Overall, I feel that it would be more powerful to compare matched cohorts.
3. In lines 161/162 the authors state: "In all cell subsets, genes upregulated during acute infection were markedly downregulated in LC samples from individuals with mild to moderate acute infection." While there is a general pattern, there is clearly also a cluster of genes in the top of the 'Day 292-311' column (Figure 2B) that appear to be upregulated with some overlap to the acute infection.
4. In lines 166-171 the authors state: "Specifically, differential gene expression analysis in CD14+ monocytes revealed distinct upregulation of 1737 genes in LC samples, particularly in individuals with mild-to-moderate acute SARS-CoV2 infection, compared to samples from acute infection, recovered patients and HCs. Notably, the upregulation of these genes intensified over time, indicating persistent and progressive alterations in the gene expression profile of CD14+ monocytes in LC. The persistent upregulated genes are associated with both pro-inflammatory (e.g., CSF1, IRF8, RELA, NOTCH1) and anti-inflammatory (e.g., TGFB1, SMADs, ENG and SERPINE1/PAI-1) functions." Contrary to what is stated here, looking at the data in Figure 2B, it seems that differential gene expression is rather transient and the transcriptional changes observed at Day 101-170 and Day 191-251 are largely reversed by Day 292.
5. Along these lines, which time point was analyzed in Figure 3F?
6. Also, the authors state that TGF-beta signaling and WNT-beta catenin signaling pathways in CD14+ monocytes were positively correlated with FAS scores (lines 243-244). They should acknowledge that the correlations were rather weak (R=0.28).
7. Figure 3C shows GSEA for CD14 monocytes and NK cells for 3 different time points in the comparison with acute disease ("Vs Acute") and 2 time points in the comparison with Recovered ("Vs Recov), missing "Day 53-78". Yet, for CD8 T cells there are 3 time points in both comparisons. What is the reason? Also, it would be important to also compare this with Healthy controls.
8. Figure 3 and some of the interpretations are problematic and inconsistent. The authors describe four monocyte clusters MC1-MC4 within the monocyte space and found MC4 to display "higher activity in "TGF-beta signalling" and "WNT-beta catenin signalling". They state that LC (PASC) samples from severe initial disease did not enrich for MC4, which is also evident in Suppl. Figure 4. In contrast, in Figure 3G, where the authors correlated MC cluster abundance per donor with FAS scores, it is clear that the majority (if not all!) donors with high percentages of MC4 of the CD14 monocytes were in fact "LC-severe" (=dark red dots). This does not square with some of the previous statements that showed a specific transcriptional program in LC-mild/moderate compared to LC-severe. It is also not in line with the results shown in Figure 4C.
9. Similar to the previous comment, Figure 4H (left most panel, "TGFB1") suggests that TGFB1 was indeed increased in the 'LC-severe' samples, which contradicts prior statements that this was a specific feature of 'LC-mild/moderate'.
10. It is also not clear when LC samples were compared to samples from patients with acute disease or those who had recovered, like in Figure 2C, and when they were compared to healthy controls, like in Figure 4G-I. This should be largely consistent throughout the study or specifically stated.
11. In Figure 6J,K the authors analyzed the response of PBMC from patients and recovered donors. They categorized the donors based on the content in CD14 monocytes with predefined features that the authors had termed LC-CS (Long COVID

cellular state) into those containing >10% or <10% of LC-CS cells. They then found differences with regards to TGF-beta signaling, WNT-beta signaling and other pathways that were previously identified as defining features of LC-CS. When categorizing samples into those with a high and low content of cells with predetermined features and then finding differences in these exact features it seems like "self-fulfilling". It would make much more sense to compare functional responses of PBMC from recovered patients to LC / PASC patients and healthy controls.

12. Finally, the fact that the authors find similarities between blood monocytes from PASC patients and BAL macrophages from PASC patients is interesting. However, I wonder whether these are actually physiologically linked. The authors have previously argued that LC-CS (or MC4 etc.), with elevated TGF-beta and WNT-signaling are a specific feature of 'LC-mild/moderate', hence, LC patients with a mild disease course. These are more likely to experience fatigue rather than dyspnoe and reduced oxygenation, which is more frequently observed after severe COVID-19. This is also evident in their data (Suppl. Table 1). BAL samples from LC/PASC patients were likely taken from patients with respiratory symptoms ('LC-severe'). So, if the 'LC-CS' / MC4 is found primarily in 'LC-mild/moderate' how does this connect to profibrotic macrophages and respiratory PASC in 'LC-severe' patients?

13. Overall, the data appears a bit contradictory regarding the described dichotomy between transcriptional features in 'LC-mild/moderate' vs 'LC-severe' and the authors may want to clarify and solidify this a bit more.

Version 1:

Decision Letter:

10th Jul 2025

Dear Dr. Li,

Thank you for the resubmission of your revised Article, "A distinct monocyte cellular state links systemic immune dysregulation to pulmonary impairment in long COVID". Based on the referees' comments requesting a better description of the patient cohorts, we would suggest including graphs like those found in Liew et al. (Fig. 1a,b, <https://www.nature.com/articles/s41590-024-01778-0>) and Bailey et al (Fig. 1a,b, <https://www.nature.com/articles/s41590-024-01975-x>), to describe the distribution of long covid symptoms; range of acute COVID severity; timepoints of testing, sample collection and type of clinical tests or analysis in each patient. Please include this information (graphs) for all 5 cohorts. In addition, please use similarly-formatted tables for the patient parameters for all cohorts (we suggest excel tables submitted as Supplementary Tables, one Table for each cohort. Word tables would work too, but please format them clearly and similarly).

Link Redacted

We hope to receive your revised manuscript within one-two weeks. If you would need more time, please let us know. Please do not hesitate to contact me if you have any questions or would like to discuss these revisions further.

Sincerely,

Ioana Staicu, Ph.D.
Senior Editor
Nature Immunology

Tel: 212-726-9207
Fax: 212-696-9752
www.nature.com/ni

Decision Letter:

6th Sep 2025

Dear Dr. Li,

Thank you for your response to the referees' comments on your Article, "A distinct monocyte cellular state links systemic immune dysregulation to pulmonary impairment in long COVID". While we are interested in the possibility of publishing your study in Nature Immunology, the issues raised by the referees need to be addressed.

Please revise along the lines specified in your letter. At resubmission, please include a "Response to referees" detailing, point-by-point, how you addressed each referee comment (please specify the page number and the figures where the new data is found). If no action was taken to address a point, you must provide a compelling argument. This response will be sent back to the referees along with the revised manuscript.

Please include a revised version of any required reporting checklist. It will be available to referees to aid in their evaluation. Reporting summary: <https://www.nature.com/documents/nr-reporting-summary.pdf>

When submitting the revised version of your manuscript, please pay close attention to our [href="https://www.nature.com/nature-portfolio/editorial-policies/image-integrity">Digital Image Integrity Guidelines. and to the following points below:](https://www.nature.com/nature-portfolio/editorial-policies/image-integrity)

Please note, Extended Data figures and tables are online-only (appearing in the online PDF and full-text HTML version of the paper), peer-reviewed display items that provide essential background to the Article but are not included in the printed version of the paper due to space constraints or being of interest only to a few specialists. A maximum of ten Extended Data display items (figures and tables) is typically permitted. When re-submitting your manuscript, please ensure that any supplementary figures and tables that are more critical to the manuscript's conclusions are converted to Extended data to increase these data's visibility.

Link Redacted

We hope to receive your revised manuscript within two-four weeks. If you cannot send it within this time, please let us know. We will be happy to consider your revision so long as nothing similar has been accepted for publication at Nature Immunology or published elsewhere.

Nature Immunology is committed to improving transparency in authorship. As part of our efforts in this direction, we are now requesting that all authors identified as 'corresponding author' on published papers create and link their Open Researcher and Contributor Identifier (ORCID) with their account on the Manuscript Tracking System (MTS), prior to acceptance. ORCID helps the scientific community achieve unambiguous attribution of all scholarly contributions. You can create and link your ORCID from the home page of the MTS by clicking on 'Modify my Springer Nature account'. For more information please visit <http://www.springernature.com/orcid>.

Sincerely,

Ioana Staicu, Ph.D.
Senior Editor
Nature Immunology

Tel: 212-726-9207
Fax: 212-696-9752
www.nature.com/ni

Reviewers' Comments:

Reviewer #2 (Remarks to the Author):

Kumar et al. have significantly improved their manuscript and answered most of my questions and responded adequately to most previous comments. It still remains a complex study and some inherent weaknesses such as imperfect matching of cases and controls in the

cohorts, uneven distribution of samples and cell yields over different conditions and time points remain an important limitation. The authors have made significant efforts to address some of these issues and to mitigate potential effects of confounding. They have also added new information and figures to improve the clarity and transparency of their results and their methods.

However, there are still some issues that require clarification.

1. The study falls short of showing evidence of causality (which I acknowledge is challenging in observational settings) or directionality. I feel that this aspect still requires some more critical discussion.
2. In this regard, given the longitudinal data, do the authors have any evidence regarding the time point of MC-4 appearance and abundance in relation to the onset or resolution of LC symptoms?
3. As mentioned, the correlations between MC4 (LC-CS) monocytes and clinical parameters such as Fatigue score or pO₂ were weak. In fact, the majority of patients in the cohort had very low (or undetectable?) numbers of MC4 cells, as shown in Figure 3G. In contrast, these data points (samples with 0% MC4) seem to be missing in Figure 3H, in which the y-axis is cut just below 10%. These data points should be included in the graph and in the statistical analysis. Even in the current state of Figure 3H / 3G, there seems to be a lot of overlap between the different categories, and many patients have low or no MC4 monocytes, which poses the question whether MC4 (LC-CS) are indeed causally linked to PASC / Long-COVID or whether this is an epi- or side phenomenon.

Overall, the study is impressive in size and complexity and it adds a wealth of interesting data and insights into potential links between altered monocyte states and post-acute sequelae of COVID-19, which is definitely an area of high interest. But it may require more critical discussion of the limitations specifically with regards to causality and directionality of their findings.

Reviewer #3 (Remarks to the Author):

I have been asked to consider the authors rebuttals and revisions in response to comments from Reviewers #1 and #2. Broadly, I believe that the authors have done a thorough and well considered job of responding to both reviewers, including the incorporation of a new cohort and a series of new analyses, all of which have strengthened the main conclusions. There are still gaps, but I believe they are inherent limitations in the samples/data available that do not impact the overall conclusions. I have a few specific comments, following the rebuttal:

Reviewer #1, comment 4, regarding figure 2f - I agree that the figure display statistics should not include the Recovered group. The authors demonstrate in the rebuttal that the interpretation of this data does not change if this group are excluded, so I suggest you change the statistics in 2f to exclude the Recovered group, but leave the blue dots on the plot as a comparator. Amend the legend to reflect this. You could also colour the R and P value text in the figure to red, making it apparent that the statistics refer to only one of the groups on the plot.

Reviewer #1, comment 3b and Rev#2 comment 4 are the same, warranting particular attention. On balance I think that the issue is likely to have arisen from a relative underpowering of the T5 group (being both smaller in patient number and with fewer monocytes sequenced). This isn't ideal, but is I believe acceptable when viewing the study pragmatically.

Additional comments:

While not raised by reviewers 1 and 2, I believe there is one important element that requires attention, specifically, the definition used for PASC/long COVID - I am concerned that the participants in this study may not meet many of the common definitions for 'long COVID': 1) The study includes participants with 'long COVID' at less than 3 months post acute disease, but most definitions require 3 months to have elapsed to be certain the 'acute' phase is over. 2) Do your cohorts meet the NASEM criteria for symptoms lasting >3 months? Using appropriate definitions is essential, and I fear that the current data does not indicate whether participants in this study meet these current definitions. This needs to be clarified. Reviewers 1&2 may not have identified this as the new Figure 1 graphics make this most apparent. I believe that including samples from earlier than 3 months post acute disease is still informative, if those participants are known to have maintained disease for at least 3 months (meeting the NASEM criteria). I sympathise with the authors that these definitions are likely to have changed during the course of conducting this analysis, but I view it as essential to the ability to interpret this data that it is clear if/how these cohorts meet these definitions.

In summary, I believe that the article makes a series of important discoveries that will be both valuable additions to the literature on long COVID, and potentially be of interest to the wider immunology field as an example of the immunopathology evident in a sequelae of respiratory viral infection. I find the concept of underpinning monocyte dysregulation that manifests in different tissues, owing to trafficking to different sites (i.e. fatigue versus cardiorespiratory symptoms) particularly interesting and fitting with other literature. This gives me confidence that the manuscript would be a welcome and valuable addition to the field.

Reviewer #4 (Remarks to the Author):

Kumar and colleagues applied snRNA, snATAC-seq, and targeted proteomics to PBMC and plasma samples of two discovery cohorts and scRNA-seq to PBMC and bronchoalveolar lavage fluid samples of three separate validation cohorts to assess different factors and cellular states correlating with active long COVID (LC) in patients suffering from fatigue and

respiratory symptoms following COVID-19. A subset of the discovery cohort samples was stimulated using *P. aeruginosa* followed by single-cell analysis. They found a LC cellular state (LC-CS) in circulating CD14⁺ monocytes and increased TNF, CCL2 and CXCL11 concentrations in patients with active LC who had previously experienced mild-to-moderate acute COVID-19. In two independent LC cohorts of patients with respiratory symptoms, this LC-CS was enriched in blood CD14⁺ monocytes and also found in myeloid cells isolated from bronchoalveolar lavage fluid. Monocytes of individuals with high percentages of LC-CS CD14⁺ monocytes showed an abnormal response to in vitro stimulation with *P. aeruginosa*. Overall, this is an interesting study characterizing monocytes in LC patients. The authors are experts in multiome analysis of immune cells. The epigenetic insight into monocytes and immune cells of LC patients is particularly noteworthy.

Major points

(1) The authors characterized different LC-CS in monocytes. The LC-CS in MC4 was very different between months 3 to 9 to that of the other monocyte subsets (Fig. 3E). Are these inter-monocyte differences driven mainly by the pathways shown in Fig. 3F, i.e. TNF, WNT-beta-catenin, and TGF-beta signaling or by other factors?

(2) Cytokines and chemokines are very tightly regulated. By using the Quanterix HD SP-X Imaging and Analysis System together with a cytokine and a separate chemokine panel, the authors describe increased concentrations of TNF, CCL2 and CXCL11 in patients with active LC who had previously experienced mild-to-moderate acute COVID-19. Did the authors verify their measurements using another assay?

(3) The curves in Fig. 3G for MC4 seem to correlate with some outliers rather than with the majority of data points. How would the analyses look like if the most extreme outliers were excluded? How did the authors determine high vs. low MC4 samples? Would the percentage of high vs. low MC4 samples still correlate with the FAS score, as shown in Fig. 3H, if the authors left out the apparent outliers?

(4) The epigenetic analysis of the identified different monocyte subsets in Fig. 5 is interesting. Based on the landscapes shown of IER3 and LMNA there seem to be rather minimal differences between MC4 and the other monocyte subsets. Yet, the signal counts appear to be significantly different. Could the authors make this connection between landscapes and calculated signal counts more intuitive to appreciate?

(5) I understand the monocyte clusters identified in BAL myeloid cells and shown in Fig. 6 are different from the ones described in the previous figures. The BAL monocyte cluster C1 appears to correlate most with the PBMC monocyte cluster MC4, and C1 appears to show a profibrotic gene signature. Is this profibrotic signature mainly based on TGF-beta? If yes, this might not be a solid signature molecule indicating a profibrotic state, as depending on its cytokine microenvironment, TGF-beta can promote type 3 inflammation or immune regulation, both without causing fibrosis. Does MC4 also show a profibrotic gene signature?

Minor points

(a) In the legend to figure 5, Asterix should be asterisk.

(b) In Fig. 6I, it should read *P. aeruginosa* instead of *P. auerginosa*.

Version 3:

Decision Letter:

Dear Dr. Li,

Your Research Article "A distinct monocyte cellular state links systemic immune dysregulation to pulmonary impairment in long COVID" has now been seen by 2 referees. We are happy to inform you that if you revise your manuscript appropriately in response to the referees' comments and our editorial requirements your manuscript should be publishable in Nature Immunology.

Please revise your manuscript according with the reviewers' comments. At resubmission, please include a point-by-point response to the referees' comments, noting the pages and lines where the changes can be found in the revision. Please highlight the changes in the revised manuscript as well.

We are trying to improve the quality and transparency of methods and statistics reporting in our papers (please see our editorial in the May 2013 issue). Please update the Life Sciences Reporting Summary, and supplements if applicable, with any information relevant to any new experiments and upload it (as a Related Manuscript File) along with the files for your revision. If nothing in the checklist has changed, please upload the current version again.

TRANSPARENT PEER REVIEW

Nature Immunology offers a transparent peer review option for new original research manuscripts submitted from 1st December 2019. We encourage increased transparency in peer review by publishing the reviewer comments, author rebuttal letters and editorial decision letters if the authors agree. Such peer review material is made available as a supplementary peer review file. **Please state in the cover letter 'I wish to participate in transparent peer review' if you**

want to opt in, or 'I do not wish to participate in transparent peer review' if you don't. Failure to state your preference will result in delays in accepting your manuscript for publication.

ORCID

Nature Immunology is committed to improving transparency in authorship. As part of our efforts in this direction, we are now requesting that all authors identified as 'corresponding author' on published papers create and link their Open Researcher and Contributor Identifier (ORCID) with their account on the Manuscript Tracking System (MTS), prior to acceptance. ORCID helps the scientific community achieve unambiguous attribution of all scholarly contributions. For more information please visit www.springernature.com/orcid.

Before resubmitting the final version of the manuscript, if you are listed as a corresponding author on the manuscript, please follow the steps below to link your account on our MTS with your ORCID. If you don't have an ORCID yet, you will be able to create one in minutes. If you are not listed as a corresponding author, please ensure that the corresponding author(s) comply.

1. From the home page of the [MTS](https://mts-ni.nature.com/cgi-bin/main.plex) click on '**Modify my Springer Nature account**' under '**General tasks**'.
2. In the '**Personal profile**' tab, click on '**ORCID Create/link an Open Researcher Contributor ID(ORCID)**'. This will redirect you to the ORCID website.
- 3a. If you already have an ORCID account, enter your ORCID email and password and click on '**Authorize**' to link your ORCID with your account on the MTS.
- 3b. If you don't yet have an ORCID, you can easily create one by providing the required information and then click on '**Authorize**'. This will link your newly created ORCID with your account on the MTS.

IMPORTANT: All authors identified as 'corresponding authors' on the manuscript must follow these instructions. Non-corresponding authors do not have to link their ORCIDs, but please note that it will not be possible to add/modify ORCIDs at proof. Thus, if they wish to have their ORCID added to the paper, they must also follow the above procedure prior to acceptance. To support ORCID's aims, we only allow a single ORCID identifier to be attached to one account. If you have any issues attaching an ORCID identifier to your Manuscript Tracking System account, please contact the [Platform Support Helpdesk](http://platformsupport.nature.com/). We hope that you will support this initiative and supply the required information. Should you have any query or comments, please do not hesitate to contact me.

Author names using non-Roman characters

Nature Portfolio journals can support presentation of author names using non-Roman characters in the HTML version of the article. If you wish to, please include author names in parentheses after the Roman-character spelling; [see example online here](https://www.nature.com/articles/s44222-024-00258-2). Currently supported scripts are: Arabic, Chinese, Cyrillic, Devanagari, Greek, Hebrew, Hangul, Japanese and Persian. You will be asked to verify the rendering is correct at proof stage.

Nature Immunology has now transitioned to a unified Rights Collection system which will allow our Author Services team to quickly and easily collect the rights and permissions required to publish your work. Once your paper is accepted, you will receive an email in approximately 10 business days providing you with a link to complete the grant of rights. If you choose to publish Open Access, our Author Services team will also be in touch at that time regarding any additional information that may be required to arrange payment for your article.

In recognition of the time and expertise our reviewers provide to Nature Immunology's editorial process, we would like to formally acknowledge their contribution to the external peer review of your manuscript entitled "A distinct monocyte cellular state links systemic immune dysregulation to pulmonary impairment in long COVID". For those reviewers who give their assent, we will be publishing their names alongside the published article.

When you are ready to submit your revised manuscript, please use the URL below to submit the revised version:

Link Redacted

We hope to receive your revised manuscript in 10 days, by 30th Oct 2025. Please let us know if circumstances will delay submission beyond this time. If you have any questions please do not hesitate to contact me.

Sincerely,

Ioana Staicu, Ph.D.
Senior Editor
Nature Immunology

Tel: 212-726-9207
Fax: 212-696-9752
www.nature.com/ni

Reviewer #2 (Remarks to the Author):

The authors have addressed my previous comments through additional analyses and textual revisions, with one exception. I believe it remains important to acknowledge, both in the Results and Discussion sections, that the observed associations between LC-CS (MC4) and O₂-saturation or fatigue scores are relatively weak, and that MC4 was undetectable in a substantial proportion of samples. This clearly suggests that additional or alternative mechanisms contribute to PASC. While the authors acknowledge this point in their response to my comments, it is not clearly reflected in the manuscript itself. Explicitly stating this limitation, would not detract from the careful and comprehensive characterization of monocyte states in PASC presented in this well-conducted study. Rather, it would strengthen the critical discussion of the results. Overall, the manuscript and the accompanying data represent a valuable contribution to the growing body of work on potential immune mechanisms underlying PASC.

Reviewer #4 (Remarks to the Author):

The authors have convincingly provided data and rationale to address my previous comments.

Version 4:

Decision Letter:

Our ref: NI-A39964D

29th Oct 2025

Dear Dr. Li,

Thank you for submitting your revised manuscript "A distinct monocyte cellular state links systemic immune dysregulation to pulmonary impairment in long COVID" (NI-A39964D). We are happy to inform you that if you revise your manuscript appropriately according to our editorial requirements, your manuscript should be publishable in Nature Immunology.

I will now pre-edit the current version of your paper. We will also perform detailed checks on your paper and will send you a checklist detailing our editorial and formatting requirements in one-two weeks. Please do not upload the final materials and make any revisions until you receive this additional information from us.

While waiting for the pre-edit check, please deposit all omic and code data into public repositories so that the accession codes are readily available to be added in the revised manuscript. We cannot accept the paper without the codes.

In addition, all corresponding authors need to update and link their ORCID to their Nature account. We cannot accept the paper without this information. We suggest that you look into this while waiting for the pre-edited manuscript. Should you have any query or comments about ORCID, please do not hesitate to contact our editorial assistant at immunology@us.nature.com.

If you had not uploaded a Word file for the current version of the manuscript, we will need one before beginning the editing process; please email that to immunology@us.nature.com at your earliest convenience.

Thank you again for your interest in Nature Immunology. Please do not hesitate to contact me if you have any questions.

Sincerely,

Ioana Staicu, Ph.D.
Senior Editor
Nature Immunology

Tel: 212-726-9207
Fax: 212-696-9752
www.nature.com/ni

We sincerely thank the reviewers for their thorough and thoughtful evaluation of our manuscript. We are encouraged by their positive assessments, including comments such as:

- *“Over all the findings of the paper are interesting and should add to our knowledge regarding the pathogenesis of long Covid. The paper has a number of strong elements and has potential for impact. (Reviewer. #1)”*
- *“this is a complex study that offers interesting insights into transcriptional states in PASC patients. Specifically, distinguishing between PASC patients with acute-to-moderate initial disease and those with initially severe disease is important (Reviewer. #2)”*

We are particularly grateful for the constructive suggestions that helped us further improve and refine the manuscript.

Below, we provided a detailed point-by-point response to each comment.

- **Reviewers' comments** are presented in **black**,
- **Our responses** are in **blue**
- **Revisions made to the manuscript** are highlighted in **purple**

REVIEWERS' COMMENTS:

Reviewer #1 (Remarks to the Author):

In the paper entitled “A distinct monocyte cellular state links systemic immune dysregulation to pulmonary impairment in long COVID” (ref. num. NI-A39964-T), Li and colleagues perform a large number of transcriptomal and functional studies on peripheral blood monocytes from a large number of patients with long Covid. The paper uses complementary techniques to derive a signature for cells that are enriched in patients with long Covid, and suggests that signatures for patients who had mild/moderate acute disease may be different than ones from patients that had severe acute disease.

Over all the findings of the paper are interesting and should add to our knowledge regarding the pathogenesis of long Covid. The paper has a number of strong elements and has potential for impact. However, there are a number of limitations that need to be addressed.

We thank the reviewer for the positive assessment of our study.

Major comments.

1. The paper is difficult to follow. In part, this arises from the overabundance of data and lack of focus. For example, the first part of the paper delves into differences observed over time in long COVID patients and further stratifies these differences based on initial severity of COVID infection. This seems like it could be a major focus / message of the paper. However, as the paper progresses this theme is lost entirely, and a new focus on respiratory vs non respiratory long COVID phenotypes emerges. The focus of the paper is further diluted by large paragraphs in the results section that refer to Supplementary Figs that don't always relate to the core theme. As a result, the main message of the story becomes convoluted. While it is

important to acknowledge the abundance of data and analysis that are included in the paper, much of it detracts from the story rather than adding.

We thank the reviewer for this thoughtful and constructive feedback. We sincerely apologize for the lack of clarity and narrative cohesion in the previous version of the manuscript. In response, we have made significant revisions to improve focus, readability and alignment with the core theme of the study. Specifically:

1. We have updated the cohort figure (**Figure 1A/R1** in this document) to more accurately depict the different cohorts that are used in this study and different data modalities evaluated from each cohort. **We have also introduced a new cohort (Cohort 3), to provide a clearer view of how our findings (LC-CS), identified using patient stratification link with the respiratory PASC phenotype.** Cohort 3 was carefully selected to recruit patients with mild COVID-19 infection history. A few of these patients also showed Bronchial Hyperresponsiveness (BHR), thereby representing respiratory PASC with lung function abnormality (more clinical information on page 5 of this document). In summary, we found LC-CS within all LC patients of this cohort (Figures R9 on page 16 of this document). Additionally, patients with BHR also showed higher expression of LC-CS signature compared to other LC patients (R12 on page 20 of this document).

Figure R1: Revised cohort description, now presented as Figure 1A in the manuscript.

2. We also added a new **overview figure (Figure 1B/Figure R2** in this document), which outlines the structure of the manuscript and clearly maps each figure to the corresponding research objectives and analyses. We have re-structured the manuscript to outline the flow of the study with rationale for each section in bold:
 - a. **Exploration-** which included exploring the single-cell multiome datasets (Cohort 1) and cytokine dataset (Cohort 2) to identify the **correlates of systemic inflammation in LC** (Figure 2). The correlates from single cell data led us to **identify the LC-CS** in circulating monocytes (Figure 3).

- b. **Characterization-** We next **validated** this cellular state in an independent cohort of LC patients (Cohort 3), showing higher expression of LC-CS signature in Respiratory PASC patients with BHR and performed flow cytometry analysis of key markers of LC-CS (Cohort 4). Findings from both of these analyses are included in Figure 4. We also characterized the **epigenetic regulation** of the LC-CS (Cohort 1) in Figure 5.
- c. **Function-** We leveraged the published data scRNA-seq data of paired PBMC and BAL samples to examine the **role of LC-CS in the tissue** (Cohort 5) and found that the macrophages with LC-CS signature showed a pro-fibrotic signature in BAL. We also examined the **functional immune response** of patients with a higher proportion of LC-CS within their PBMC upon an immune insult (Cohort 1 subset). All findings for this part are summarized in Figure 6.

Figure R2: Schematic of the analysis flow (now included as Figure 1B in the revised manuscript).

- 3 We also substantially revised the content to streamline the narrative, improve logical flow and relocated extended analysis and descriptions to Extended Data. We clarified

the relevance of all remaining Extended Data Figs, ensuring each supports a key conclusion. Below we highlight some of the major changes of each section

Introduction:

- a. By including the rationale for patient stratification, leading to our hypothesis.
- b. Improving the overall flow of introduction.

Results:

- a. Substantially removed content that deviated from main message by removing cell-cell communication analysis and moving pseudotime trajectory analysis to Extended Data, reducing the whole section to only a short paragraph.
- b. Added cohort 3 analysis, showing how the findings (LC-CS) from patient stratification link to LC and particularly implied in Respiratory PASC phenotype.

Discussion:

- a. Rearranged text to conclude results and implications of each finding in the context of LC disease.

Methods:

- a. Added new methods section on patient-sample stratification performed for the analysis.

We hope that with these changes, the focus of the paper is improved and allows a better follow-up of all the analysis, rationale and interpretations of the study. For this general comment, we have provided a comprehensive response without referencing specific line numbers, as the improvements are distributed across multiple sections of the text. Nevertheless, the changes described above are specified individually with line numbers in manuscript with replies to the comments below.

2. A potential strength of the paper is the richness of the clinical cohort. However, details about the cohort are not described in sufficient detail to enable the reader to follow key elements in the manuscript. This weighs down the paper. Items to consider include the following.

We thank the reviewer for their thoughtful comments. We have addressed each point in the revised manuscript, as detailed below.

a. A Table describing the cohort should appear in the main body (rather than the supplement)

In response, we have included detailed descriptions of the primary cohorts (Cohorts 1 and 2) in the main manuscript: Table 1 presents the information for samples with sequencing data and Table 2 summarizes the donors/samples used for cytokine data generation. Additionally, we have added Extended data table 1, which provides demographic and clinical metadata for each sequenced sample from cohort 1. Information on the validation cohorts (Cohort 3 and 4) is now included in Extended Data Fig 6A and 6B. Since cohort 5 is published data, its information is available with the publication.

The manuscript text referencing these tables/figure has been updated accordingly at line 170

“The cohort included individuals with ongoing SARS-CoV-2 infection, patients experiencing post-acute sequelae of SARS-CoV-2 infection (PASC/LC) at varying time points (53-311 days

post-infection), samples from patients recovered from their LC symptoms (Recovered) and COVID-naïve healthy controls (HC) (Table 1, Extended Data Fig.1A and Extended data table 1).”

at line 176

“These plasma samples represented 142 patients, 39 patients with longitudinal data and 103 patients at single time points. Clinical parameters, including blood gases, FAS scores and dyspnoea scores, were recorded (Table 2).”

and at line 334

To validate the LC-CS signature, we analysed scRNA-seq data from an independent cohort (Cohort 3) of eight LC patients who had experienced mild acute COVID-19 disease but presented with severe LC symptoms 8-42 months post-infection (Extended Data Fig 6A).

Patient ID	Category	Gender	Age	acute COVID-group	Months post infection until sampling	Fatigue	Pneumological Manifestation	Pulmonary Function Test result	Cardiological Manifestation	mMRC
1_SK	nonResp-PASC	M	44	mild	16	yes	no	no abnormalities	no	2
2_SK	nonResp-PASC	F	21	mild	18	yes	yes	no abnormalities	no	3
3_SK	nonResp-PASC	F	41	mild	19	yes	yes	no abnormalities	no	2
4_SK	Resp-PASC	F	63	mild	12	yes	yes	bronchial hyperresponsiveness	no	2
5_SK	Resp-PASC	F	39	mild	8	yes	yes	bronchial hyperresponsiveness	yes	3
6_SK	nonResp-PASC	M	53	mild	8	yes	no	no abnormalities	no	2
7_SK	nonResp-PASC	F	46	mild	11	yes	yes	no abnormalities	no	2
8_SK	Resp-PASC	M	48	mild	42	yes	yes	bronchial hyperresponsiveness, reduced FEV1 and restrictive ventilation dysfunction	yes	4

Figure R3: Demographic and clinical information of samples in the new added cohort (cohort 3).

And at line 347

The LC patient samples were collected between 100 days to 433 days post-acute COVID-19 along with their clinical measurements (Extended Data Fig 6B, See Methods).

b. Criteria used to diagnose long Covid should be clearly stated in the text. (Of note, the only reference that speaks to long Covid diagnosis is #21. I could only find the text in German. This limits a large portion of the general readership)

We thank the reviewer for pointing this point and apologize for the lack of clarity. Below, we provide a summary of the diagnostic criteria for Long COVID used in our study.

Patients diagnosed with severe cases of the disease were offered appointments at the outpatient clinic, provided they were still experiencing symptoms. Patients with a mild-to-moderate course of the disease also reported at the outpatient clinic if they had symptoms after recovering from a previous infection with the virus. All reporting patients were polymerase chain reaction (PCR) tested for SARS-CoV-2 infection. Upon a negative PCR result, a diagnosis of exclusion was performed, in which other diseases that could explain the symptoms were ruled out. Therefore, the diagnosis was made in accordance with the S1 guidelines and the Delphi Consensus Criteria.

From the S1 guidelines criteria:

Based on the Cochrane Rehabilitation Review and an international Delphi conference with the participation of the WHO, one of the following three categories can be used to diagnose PCS syndrome:

- 1) Symptoms that persist after acute COVID-19 or its treatment,*
- 2) new symptoms that appear after the end of the acute COVID-19 phase, but can be understood as a consequence of the SARS-CoV-2 infection,*
- 3) Worsening of a pre-existing illness as a result of a SARS-CoV-2 infection.*

Three different patient populations were covered. The overall group of all patients with Post-Covid Syndrome (PCS), who can be significantly impaired in their participation in school, education, work and other social life:

- 1. Patients who have received intensive care treatment for COVID-19 and suffer from postintensive care syndrome (PICS).*
- 2. Patients who develop secondary diseases such as cardiovascular complications, cognitive impairment or a mental disorder as a result of a SARS-CoV-2 infection with a time latency,*
- 3. Patients with symptoms of fatigue and exercise intolerance with/without dyspnea and neurocognitive disorders ("brainfog").*

We have included the information pertaining the LC diagnosis criteria at line 147:

"Inclusion was based on the following criteria: (1) symptoms persisting beyond the acute phase of SARS-COV-2 infection or its treatment; (2) new symptoms emerging after resolution of the acute illness and are attributed to the prior infection; and (3) exacerbation of pre-existing conditions as a consequence of SAR-CoV-2 infection"

c. Of the patients diagnosed with long-COVID what domains are the ones most impacted? I realize this information can be difficult to gather. However, if this can be achieved it would help the reader contextualize which organs / physical states are affected in the long-Covid cohort. This is relevant since later in the paper, data from an independent cohort with respiratory vs non-respiratory long-Covid are analyzed.

We thank the reviewer for this insightful comment. We fully agree that identifying the affected domains in LC patients is crucial for contextualizing disease heterogeneity, particularly in light of our comparative analyses involving respiratory and non-respiratory subgroups.

For the primary cohorts (**Cohort 1 and 2**) and a validation cohort (**Cohort 4**), detailed domain-specific data were limited, as these patients primarily presented to the pneumological outpatient clinic at Hannover Medical School (MHH, Hannover, Germany) with a range of symptoms including dyspnoea, fatigue, and headaches. Functional assessments such as the post-COVID-19 Functional Status scale (PCFS) scores, FAS scores, blood gases analysis and lung function tests were performed; however, a systematic organ-level domain classification was not feasible due to variability and retrospective nature of data collection.

We have included following text regarding this section in the Results at 144

Patients reporting to the pneumology outpatient clinic at Hannover Medical School (MHH, Hannover, Germany) with symptoms such as headache, dyspnoea and fatigue were recruited based on established diagnostic criteria in accordance with the German S1 guidelines and the Delphi Consensus Criteria of post COVID-19 condition ^{21,22}.

To address this limitation and improve clinical contextualization, we incorporated an additional, newly added cohort (**Cohort 3**) specifically assessing affected domains more granularly. In this cohort, beyond fatigue and dyspnoea, two patients showed cardiological manifestations, three showed bronchial hyperresponsiveness (BHR), and one had reduced FEV1 and restrictive ventilatory patterns. Based on these features, three patients were classified as having **respiratory PASC** with BHR, and the remaining as **non-respiratory PASC** (even though they had fatigue and dyspnoea). All patients had a history of mild acute COVID-19 disease. Details for this new cohort are provided in Extended Data Fig 6A and in Figure R3 (page 5 of this document).

Regarding **Cohort 5** (external, US-based study), this group comprised patients with severe **respiratory** complaints and a pro-fibrotic phenotype in their lungs. As circulating monocytes have been implicated in pro-fibrotic phenotype, we had explored whether similar monocyte cellular state found in peripheral blood were also present in BAL fluid, potentially contributing to the respiratory-dominated LC phenotype. We have clarified this rationale in the manuscript at line 405

Previous studies have implicated a role of circulating monocytes in the pathogenesis of PASCs, particularly in the context of pulmonary fibrosis^{42,43}. We hypothesized that the LC-CS monocyte subset, identified in circulation, may contribute to fibrotic remodelling upon recruitment to the lungs. To investigate this, we analysed an independent cohort comprising paired PBMCs and BAL fluid samples from patients clinically classified as having respiratory or non-respiratory PASC (Resp-PASC and nonResp-PASC, respectively). This classification was based on differences in lung parameters within these two categories.

d. Mean values for PaO₂ and pulmonary physiology parameters fall in the normal range for all groups. While MMRC scores appear to be elevated – these are subjective and can be elevated in patients without lung disease (such as those with chronic fatigue syndrome). Is there any objective evidence that the long covid cohort has persistent lung disease?

Thank you for this insightful comment. We agree that summarized PaO₂ and standard pulmonary physiology parameters of the primary cohorts fall within the normal range i.e. (<

80%/target) for all the groups, and that mMRC scores, while elevated, are indeed subjective and can be influenced by non-pulmonary factors, such as chronic fatigue.

While we lack radiographic data (e.g. chest CT) or advanced cardiopulmonary testing (e.g. CPET) for these cohorts, we interpret the elevated mMRC dyspnoea scores within the **broader context of LC symptomatology**. We acknowledge that this does not constitute objective evidence of persistent lung disease in the primary cohorts.

To enhance transparency, we now have provided individual-level pulmonary function data for each sequenced patient and time point in Extended data table 1, highlighting the variability within the cohort. We have also revised the manuscript to clarify this limitation and emphasize that mMRC elevations are interpreted as part of the multisystem presentation of LC rather than definitive markers of pulmonary pathology.

Notably, in the newly added cohort (**Cohort 3**) collected from patients with a mild COVID-19 disease history, three patients were diagnosed with BHR, thereby reflecting abnormalities of the **airway**. Other five LC patients showed fatigue and dyspnoea. We therefore classified these eight patients as either Resp-PASC (patients with BHR) and nonresp-PASC (patients with no abnormalities in pulmonary function test) and generated scRNA-seq data from their PBMCs. We found LC-CS signature in all eight patients. Notably, we observed higher enrichment of LC-CS signature within Resp-PASC patients in comparison to nonresp-PASC patients (Figure R9 and R12 of this document). These results additionally support the link of LC-CS to Respiratory PASC, as we established with the published dataset

We have included Extended data table 1 for Cohort 1 and Extended Data Fig 6A as mentioned at page 4-5 of this document.

And included text summarizing this in the Discussion section at line 479.

Even though blood O₂ of the cohort is considered within the normal ranges, the negative correlation may suggest minor gas-exchange abnormalities in these patients and do not constitute direct evidence of persistent lung disease.

e. How are recovered patients defined? Did these patients never develop long Covid? Or did they have LC and then recover?

Recovered patients here represents, reporting patients who developed long COVID and were recovered. All patients reported for a follow up visit > 50 days since their acute COVID-19 disease. Recovered status was assigned if all the measured parameters including FAS, MMRC and Quality of life scores were below the threshold along with clinical parameters of blood gases and lung function were within the normal ranges, as verified by the physicians. The detailed information is also available in Extended Data table 1.

We have reworded it in the manuscript at line 168

“The cohort included individuals with ongoing SARS-CoV-2 infection, patients experiencing post-acute sequelae of SARS-CoV-2 infection (PASC/LC) at varying time points (53-311 days post-infection), **samples from patients recovered from their LC symptoms (Recovered)**”

f. Were PaO₂, PFTs, FAS and MMRC measured at the same time PBMCs were drawn?

Yes, all the clinical and patient questionnaire-based information was measured at each visit along with the PBMCs collection. The Extended data table 1 includes this information for each patient sample at each time point.

We have added a sentence from line 158 clarifying this point as following:

All clinical parameters, including blood gas measurement, FAS scores, dyspnoea scores, and quality-of-life metrics, were systematically collected and documented at each patient visit.

3. Patients with long COVID are grouped based on the initial severity of disease and timing from COVID onset. While this could be considered a strength of the paper, it also raises major concerns.

a. Separating the long covid patients into multiple groups (N=8) reduces statistical power and creates problems with multiple comparisons. Accordingly, it is absolutely essential to provide a strong rationale for breaking patients into groups based on: a) initial severity and b) time from infection.

We thank the reviewer for this comment. We fully agree with the reviewer and therefore have expanded text on the rationale for patient stratification in introduction at line 114

Previous studies investigating immune alterations in LC, often overlooked the inter-patient variability, even among patients reporting with LC symptoms e.g., associated with respiratory issues. Notably, while the majority of LC cases emerge in patients who experienced mild-to-moderate acute illness¹³, most molecular studies fail to stratify patients by the severity of their acute infection⁶⁻⁸. This oversight is critical, as individuals who experienced severe acute illness—particularly those requiring ICU care—often underwent a range of interventions- likely resulting in distinct molecular imprints on the circulating immune cells, compared to those from mild-to-moderate LC patients. These imprints may confound efforts to link immune molecular signatures to LC symptoms, underscoring the need for more refined patient stratification.

And we have added the following text in Results at line 184

Heterogeneity in LC molecular imprints may be influenced by differences in acute disease severity and treatment, particularly ICU interventions. To account for this, we stratified LC patients into two groups (LC-mild-to-moderate and LC-severe), based on their acute COVID-19 disease WHO scores (See Methods). To examine the molecular signatures of disease progression, we further stratified samples by time since acute infection diagnosis (See Methods).

Additionally, the methods section at line 718 now includes information on stratification, grouping of samples and the comparisons made.

Patient and sample stratification and sample category classifications (Cohort 1, 2 and 3):

We first stratified LC patients into two groups, based on their acute COVID-19 disease WHO scores. Where LC samples from patients with acute COVID-19 WHO scores 1-5 were classified as LC-mild-to-moderate patients and LC samples from patients with acute COVID-19 WHO scores 6-9 were classified as LC-severe patients (6-9). Further samples were also classified based on time-points of collection post-acute-infection. These time points are as following: T1- Acute infection, T2- Day 53-78: ~ 2.5 months, T3 - Day 101-170: ~ 5.5 months, T4 - Day 191-251 ~ 8 months and T5 - Day 292-311 ~ 10 months.

To ensure our findings were not convoluted with COVID-19 infection imprint on immune cells, and were unique to LC, we performed transcriptome comparisons either to acute infection samples i.e., T1 or to recovered samples i.e., samples from patients that were recovered from LC. All recovered patient samples were considered as 1 category. Consequently, within each group of patients (LC-mild-to-moderate/LC-severe), comparisons were performed as T2 vs T1, T3 vs T1, T4 vs T1, or T2 vs Recovered, T3 vs Recovered etc. In heatmaps, the transcriptome signatures were plotted for all categories, including COVID-19 naïve HCs.

For cohort 3, LC-CS signature was checked in CD14+ monocytes of each LC patient. Patients were further grouped into Respiratory-PASC or non-Respiratory-PASC as described in Extended Data Fig 6A.

We also included an Extended Data Fig 1A that highlights the number of samples that were compared for each analysis.

Figure R4: Figure describing distribution of samples within different groups, where rows of each group represent patients.

With text at line 170

To comprehensively characterize the cellular, molecular and regulatory landscape of LC, we performed single-cell multiome analysis (snRNA-seq and snATAC-seq) from 78 peripheral blood mononuclear cell (PBMC) samples, incorporating both longitudinal and cross-sectional data from 45 patients and 6 healthy controls, yielding approximately 118,000 high-quality cells. (Fig. 1A and Extended Data Fig 1A).

b. A major finding of the paper is an abnormal gene signature in long Covid patients that had mild/moderate acute disease. Importantly, this abnormal signature is primarily evident in two time points (Day 101-170, Day 191-251) as illustrated in Fig2b. The signature is not present at the earlier time interval or the final time interval. It is also not present in patients that had severe acute disease. This raises the question as to whether this is a true phenomenon. This is a major point that must be addressed.

We thank the reviewer for raising this important point. Indeed, the broadest signature is **most evident** at Day 101-170, Day 191-251, corresponding approximately to 5-8 months post-acute COVID-19 infection.

However, we provide multiple lines of evidence to support the robustness and biological relevance of this signature:

- **Persistent components across time:** While the broader signature peaks at the middle time points, key genes including *TGFB1* and *CALR*, as shown in Figure 2B and Figure R5 below—exhibiting **consistent upregulation across all time points** in the mild/moderate group. Importantly, these genes were also validated at **protein level** via flow cytometry **in an independent LC cohort**, supporting their persistent activity.

Figure R5. Subset of genes persistently upregulated at all time points in LC-mild-to-moderate samples.

- **Early and late signature presence in individuals:** Elements of the LC-CS signature are detectable as early as **Days 53-78** and persist in some individuals up to **Day 292-311** suggesting inter-individual variability in the temporal dynamics of the response. The fluctuating presence of the signature likely reflects the heterogeneous and dynamic trajectories of LC, rather than the absence of a true biological phenomenon. This aligns

with known variability in symptom persistence and resolution across patients. For example, as shown in the Figure R6 below:

- Patients P2 and P3 exhibit a **gradual increase** in LC-CS signature expression over time.
- P1 shows a **marked increase at T3**, with minimal upregulation at T2.
- P4 shows the signature at T2 and was subsequently diagnosed as recovered at T4, with no further follow-up.

Figure R6. Monocyte LC signature dynamics in different patients with longitudinal samples.

- **Data recovery limitations at T5:** We note that the cell yields for LC-mild/moderate at T5 (Day 292-311) was the lowest within this group (Figure R7). Although all samples were randomized before sequencing to minimize batch effect, factors such as variable cell viability and sample quality can influence data recovery. Since the LC-CS cellular state represents **a relatively smaller fraction of the CD14+ monocyte population**, the reduced number of cells may also have limited our ability to fully capture this signature in pseudobulk analyses. In summary, technical limitations at T5 may have contributed to the weaker detection of the LC-CS signature, rather than its true biological absence.

Figure R7. Monocyte numbers within each stratified category and time point.

- **Independent cohort replication.** Despite inter-sample variability described above, we observe consistent **neighbourhood enrichment of LC-CS associated cellular-state** in T5 samples compared to both acute and Recovered samples. This signature was also replicated in monocytes of an independent cohort of LC patients (newly added Cohort 3), as detailed in our response to “Comment 3e”. Furthermore, enrichment of the LC-CS signature was observed in PBMCs from a previously published cohort (Cohort 5, from a US study, Figure 6B) providing additional validation across independent datasets.

Together, these findings support the presence of a dynamic but biologically meaningful LC-CS signature in mild/moderate long COVID patients, with variability across time points likely reflecting both individual disease trajectories and technical factors affecting detection sensitivity.

We have updated the manuscript to better contextualize the temporal dynamics of the signature, and we now explicitly discuss these points in the revised manuscript, at line 198 and added the Figures R5-R7 in Extended Data Figs 1B, 1C and 1D.

Notably, the upregulation of these genes intensified over time (Figure 2B and Extended Data Fig 1B), indicating persistent and progressive alterations in the gene expression profile of CD14⁺ monocytes in LC up to Day 191-251, while the signature appears subdued in samples from Day 292-311, possible due to lower number of cells (Extended Data Fig 1C). This late time point also showed upregulation of a subset of genes overlapping acute infection profile. The persistent upregulated genes across all time points were associated with both pro-inflammatory (e.g., *CSF1*, *IRF8*, *RELA*, *NOTCH1*) and anti-inflammatory (e.g., *TGFB1*, *SMADs*, *ENG* and *SERPINE1/PAI-1*) functions (Extended Data Fig 1D).

And at line 477 highlighting patient heterogeneity as evaluated longitudinally.

Longitudinal analysis showed the progression of LC-CS signature, further describing heterogeneity in LC disease dynamics among these patients.

c. Comparisons are made between long-COVID patients and recovered patients are made for the different time points (i.e. Fig 2c, multiple supplements). Are these comparisons made for recovered patients as a whole, or are the comparisons made within timepoints (i.e. Long Covid for T3 vs Recovered for T3). The number of recovered patients at each timepoint are quite small.

We thank the reviewer for this thoughtful question. We agree that the number of recovered patients at individual time points is limited. For this reason, we grouped all recovered samples together as one group for comparison, rather than stratifying by time point.

This decision was based on two key considerations:

1) **Lack of availability of samples.** Once patients are clinically recovered, it becomes logistically difficult to obtain PBMC samples and clinical metadata at multiple longitudinal time points.

2) **Resolution of disease-associated signals:** Recovery likely reflects a return to immunological baseline, with resolution of transcriptomic alterations associated with acute infection or

persistent symptoms. Therefore, we reasoned that the precision timing of sample collection in recovered individuals would have a minimal effect on the comparisons, particularly relative to the dynamic changes observed in LC.

We now clarify this approach with an updated Extended Data Fig 1A (shown in Figure R4 above), added information in the Methods section at line 729

All recovered patient samples were considered as one category.

and address its implications in the Discussion at line 537

Our study compared molecular alterations in LCs to recovered samples (unstratified) due to limited sample size. Stratified recovered groups may additionally provide deeper insights for a matched comparison in future studies.

d. How were intervals chosen for T2-T5? These seem arbitrary and based on convenience rather than a biologic rationale.

We thank the reviewer for this valuable observation and apologize for not explaining the rationale more clearly in the manuscript.

Due to variability in the timing of patient follow-up visits (e.g. P12 was likely sampled at Day 53 post-acute infection, while P23 was probably sampled at Day 65), we grouped the samples into broader intervals (T2-T5) to enable meaningful comparisons across individuals.,

These intervals were defined as follows:

- T2: up to ~2.5 months post-acute infection
- T3: up to ~5.5 months
- T4: up to ~8 months
- T5: up to ~10 months

This grouping was designed to **balance biological relevance** (examine disease progression imprints, as shown in Figure R6 on page 12 of this document) with the **practical constraints** of patient availability, while still capturing different phases of post-acute immune dynamics. We have now clarified this approach in the revised manuscript in the methods at line 722.

Further samples were also classified based on time-points of collection post-acute-infection.

These time points are as following: T1- Acute infection, T2- Day 53-78: ~ 2.5 months, T3 - Day 101-170: ~ 5.5 months, T4 - Day 191-251 ~ 8 months and T5 - Day 292-311 ~ 10 months.

e. Early and late time intervals (T2, T5) are missing for comparisons of moderate LC vs recovered in Fig 2c and Fig 3E. This gives the appearance that data are “cherry picked” since these early and late time points have less impressive gene signatures (see comment 2b above).

We thank the reviewer for this important observation and apologize for not making the rationale behind the figure sections clearer in the manuscript.

Our primary aim in Fig. 2C was to investigate the dynamics of immune-related pathways in LC samples in comparison to acute COVID-19 samples. Since immune pathways are typically downregulated following acute infection, we sought to identify the key immune pathways

that remain elevated —or are reactivated—in LC. To expand this analysis, we also include comparisons to recovered individuals to determine whether any immune pathways showed continued or selective upregulation in LC.

The most prominent immune pathway dynamics were observed at T3 and T4, which is why these are the focal points in the main figures. As this analysis was done using both HALLMARK and REACTOME gene sets, it generated a comprehensive list of significant pathway enrichment results. To improve clarity and focus in the main text, we limited the display to immune-related pathways that showed statistically significant changes, and we plotted only these comparisons where significance was reached. Comparisons that did not reach significance for the pathways shown —such as at T2 and T5—were therefore not displayed in Figure 2C.

Specifically regarding T5, as noted in our response to “Comment 3b”, cell number per sample were substantially lower at this late time point, reducing statistical power and limiting the detection of significant pathway changes. As such, from the listed pathways in Figure 2C, no pathways reached statistical significance for LC vs. recovered at T5, and this comparison was excluded from the main figure.

To ensure transparency and provide a complete overview, we have included all pathway enrichment results—including those for T2 and T5 comparisons, across all major subsets and stratifications—in Extended data table 2. We have now clarified this rationale in Methods section of the revised manuscript at line 748.

Pseudobulk of each donor per time point was calculated, followed by similar comparisons as described in previous section using DESeq2 were done. Gene Set Enrichment Analysis (GSEA) using HALLMARK and REACTOME pathways as background was calculated using fgsea R package. Further, immune related pathways showing statistically significance for any comparisons were plotted. The whole list of statistically significant pathways resulting from all comparisons is added in Extended data table 2.

Similar to the approach in Figure 2C, the initial comparisons in Figure 3E were performed against the acute time point (T1) to track the evolution of cellular states over time. In response to the reviewer’s suggestion, we have now included the results for T2 and T5 comparisons as well. These time points show broadly similar patterns as shown in comparison to T1 in terms of the MC4 cellular state. At T2, fewer cellular neighbourhoods reached statistical significance— likely due to not fully developed cellular state in most donors at this stage (as shown above in figure R6 on page 12 of this document). However, in the T5 vs Recovered comparison, we observed neighbourhoods reaching statistical significance in the enrichment of the MC4 cellular state, supporting the persistence of this state at later time points. (See figures below)

Figure R8.1: Analysis for comparison between Mild-to-moderate T2 vs Recovered. Red indicates significant upregulated neighbourhoods, blue indicates significantly downregulated neighbourhoods, grey represent not significant. MC1-MC4 represent monocyte clusters. MC4 represents the LC-CS.

Figure R8.2: Analysis for comparison between Mild-to-moderate T5 vs Recovered. Red indicates significant upregulated neighbourhoods, blue indicates significantly downregulated neighbourhoods, grey represent not significant. MC1-MC4 represent monocyte clusters. MC4 represents the LC-CS.

We have included these two new figures in the main figure 3E with the text in the Results section at line 296.

Given the well-defined and significant role of monocytes in acute SARS-CoV-2 infection^{31,32}, we next performed differential neighbourhood analysis and compared LC samples to acute and recovered samples at all time points (Fig. 3E and Extended Data Fig. 5A-5B).

Additionally, the new cohort (cohort 3) comprising of LC patients with a mild COVID-19 disease history showed enrichment of LC-CS signature (Clust1). These samples were collected from > 8 months (T4) to up to 42 months (beyond our T5 time point) since acute infection. Importantly, the cellular state was found in each of the LC patients. Please see below:

Figure R9: (A) CD14⁺ Monocytes from LC patients of Cohort 3. **(B)** LC-CS signature AUC scores in the 3 subclusters, showing highest activity in Clust1. **(C)** Distribution of Clust1 in each of the LC patient's monocytes.

We have updated the text at line 335 in manuscript with new figures:

All patients reported fatigue and dyspnoea; three also showed bronchial hyperresponsiveness³⁵ and were classified as Respiratory-PASC (Resp-PASC). CD14⁺ monocyte analysis revealed three main clusters, with Clust1 showing significantly elevated LC-CS signature AUC scores based on the top 141 marker genes of LC-CS identified above (Figure 4A, 4B).

f. Levels of plasma cytokines (Fig2D) do not appear to exhibit the same kinetic as suggested by pseudobulk analysis. Since plasma is from a larger cohort, this reinforces concerns raised in point 2b above. Although it is not implicitly stated, it is implied that peripheral blood leukocytes are the source of the cytokines. Accordingly, if analysis needs to be broken down by time it would be useful to show gene expression for *TNF*, *CCL2*, *CXCL11* from PBMCs used to generate Fig 2b, c.

We thank the reviewer for this insightful comment. We agree that plasma cytokines levels do not fully mirror the kinetics observed in the pseudobulk transcriptomic analysis of monocytes. This discrepancy likely reflects the **systemic nature of plasma cytokines**, which represent a cumulative signal from multiple cellular sources beyond peripheral blood leukocytes. As such their levels may not directly correlate with transcriptional activity within PBMCs alone.

In response to the reviewer's suggestion, we have now **examined the gene expression of *TNF*, *CCL2* and *CXCL11* across all immune cell subsets** at all available time points. As shown in the new figure below, *TNF* expression is consistently upregulated in LC-mild/moderate patients across most immune subsets.

In contrast, *CCL2* and *CXCL11* expression is not markedly elevated in PBMCs and was not detected in all immune subsets, supporting the possibility that these cytokines are predominantly produced by non-hematopoietic sources such as endothelial, epithelial, stromal cells.

Figure R10: Gene expression of TNF, CCL2, CXCL11 in major immune cell subsets from PBMC.

We have included these additional analyses and clarified this interpretation in the revised Results Sections line 240 and have added this figure in the Extended Data Fig 2D.

TNF α gene expression was also persistently upregulated within LC mild-to-moderate samples in most immune subsets except for CD14⁺ Monocytes and CD4⁺ T cells (Extended Data Fig 2D).

4. It seems inappropriate to include recovered patients in statistical analysis for figure 2f. By definition, these patients should have low FAS scores. If gene expression has “normalized” in these patients as part of recovery they will skew the data.

We thank the reviewer for this important observation. We would like to clarify that gene expression normalization was performed at the **genome-wide level using the well-established normalization methods such as those implemented in Seurat** ensuring that statistical comparisons between conditions were not biased toward any specific group, including recovered patients. We agree, that if the FAS scores are really correlated with the pathway activity (and not with the patient status), then the recovered samples with low FAS scores should also have lower expression of genes of these pathways.

Therefore, our primary aim in Figure 2f was to assess the **association between pathway activity scores** (calculated using AUC scores of ~ 40-80 genes per pathway from pseudobulk expression per cell subset) and **FAS scores**, independent of disease status. Including recovered individuals, who by definition have low FAS scores and correspondingly lower pathway activity, strengthens this analysis by illustrating **a gradient of association across the clinical spectrum**, rather than comparing only symptomatic patients.

Indeed, the fact that recovered individuals cluster at the lower end of both fatigue and pathway activity supports the biological relevance of the relationship between these pathways and fatigue severity. Importantly, patients with higher FAS scores exhibit the highest pathway activity, consistent with this trend.

That said, we acknowledge the reviewer's concern and would be happy to provide an additional sensitivity analysis excluding recovered patients. As shown below, the direction of the correlation stays the same without the recovered samples included for all comparisons.

Figure R11: Correlations of pathways with FAS scores without recovered samples

We have added text to clarify the point at line 264

These correlations performed without patient stratification, suggest a gradient of association of these pathways across the clinical spectrum, rather than patients alone.

5. Pseudotime trajectory analysis adds little to the paper. I recommend removing it. The method “forces” a trajectory whether one is present or not. Moreover, it is not true reflection of lineage tracing since monocyte survival in the circulation is limited to a few days. In comparison, timecourses used for the analysis are on the order of many months.

We thank the reviewer for this thoughtful comment. We fully agree that **pseudotime does not reflect true lineage tracing**, particularly given the short lifespan of circulating monocytes and the extended clinical time course in our study.

Our intention, however, was not to infer cellular lineage or fate, but rather to **explore the continuum of transcriptional programs** within monocytes at a single-cell level. This approach allows us to capture the **diversity of immune states** across patients and time points in an **unsupervised, non-stratified manner**, thereby complementing our patient/timepoint-based analyses.

Importantly, this analysis provided **orthogonal validation of transcriptional modules** identified in stratified datasets, suggesting that these programs emerge consistently, independent of sample metadata. To address the reviewer's concern, we have now moved this analysis to **Extended Data Fig 5** and **substantially condensed the corresponding text**, highlighting only the core rationale and key findings at line 313 in the revised manuscript .

We additionally evaluated the transcriptional program of MC4-like cells among all CD14⁺ monocytes through trajectory analysis (See Methods) in a stratification-independent manner (Extended Data Fig 5D). The resulting three lineages showed the segregation of single cells solely based on transcriptome, with lineage 3 overlapping MC4 immune program, without prior sample stratification (i.e., acute, recovered and LC samples category) and allowed to recapitulate the pathway dynamics from stratified analysis (Extended Data Fig 5E).

We hope this clarification addresses the concern and demonstrates that this analysis was used cautiously and interpretively, without overstating biological trajectory or cellular ontogeny

6. Experiments that reanalyze published data from LC patients are both confusing and confounded by lack of healthy controls as a reference to show whether cells are “pro-inflammatory” or “pro-fibrotic.” The strength of the published data arises from patients with pulmonary symptoms versus non-pulmonary symptoms. If these data are to be leveraged, then the author's primary dataset should also incorporate respiratory versus non-pulmonary phenotyping. The ideal analysis would be to derive a LC-CS signature from the author's dataset and to demonstrate whether this gene signature maps on to PBMCs from the published dataset for respiratory vs non-respiratory phenotypes. This would help to confirm the author's results and provide a firm link to the first 5 figures of the paper. In its current form, the link is tenuous.

We thank the reviewer for this important and constructive comment, and we apologize for any confusion caused. We agree that the absence of pulmonary vs non-pulmonary symptom stratification in our primary cohort (Cohort 1) limits the interpretation of LC-CS implications in respiratory PASC. We addressed this point using two approaches; using a new cohort and examining published cohort based on reviewer's suggestion. Details of both are below.

- We incorporated a **new independent cohort (cohort 3)** from a pulmonary rehabilitation clinic in Germany. All patients in this cohort reported LC symptoms such as fatigue and dyspnoea, with three patients additionally diagnosed with bronchial hyperresponsiveness (BHR). All had experienced mild acute COVID-19 disease. Based on these clinical features, we classified the patients into respiratory-PASC and nonrespiratory-PASC subgroups. Notably, LC-CS signature cells was found in all LC patients (Figure R8 in this document), and their expression was **significantly higher** in respiratory PASC patients, compared to non-respiratory PASC patients. Please see below:

Figure R12: LC-CS signature AUC score in LC patient’s Clust1 cells, with higher AUC scores in respiratory PASC patients.

We have incorporated this analysis at line 339 of manuscript:

Notably, Respiratory-PASC patients showed significantly higher LC-CS enrichment compared to non-Respiratory-PASC (nonResp-PASC) individuals (Figure 4C). These results independently validate the presence of the LC-CS signature in mild-to-moderate LC patients and suggest a potential association with progression to Respiratory-PASC.

- We also went back and examined the published dataset again (cohort 5) and found additional immune cells within the PBMCs from 2 HCs. Therefore, we included cells from HCs also in the analysis and show results within the PBMCs for a better comparison. Following the reviewer’s helpful suggestion, we **re-evaluated the enrichment of the LC-CS signature within PBMC of 2 HCs and PASC patients**, where monocyte cluster 5 showed the highest enrichment (Figure 6A-6B, also shown below).

We then stratified the cells in the published PBMC dataset by **clinical phenotype (respiratory-PASC vs nonrespiratory-PASC vs healthy controls)**. This allowed us to test whether our LC-CS signature derived from our dataset could distinguish relevant patient subtypes in the external cohort, as suggested by the reviewer.

Indeed, we observed that **cells from respiratory PASC patients exhibited significantly higher enrichment of the LC-CS/MC4 signature**, particularly within monocyte clusters 5. In contrast, **non-respiratory PASC patients and healthy controls showed no such enrichment**. These findings support the relevance of the LC-CS signature to pulmonary manifestations of Long COVID and **establish a clearer link between our primary data and the published study**, thereby strengthening the translational relevance of our findings.

Figure R13: LC-CS signature enrichment within cluster 5 monocytes, cells stratified based on HCs and PASC categories.

We have now incorporated this analysis in line number 416 and added the figure panel in 6C.

Moreover, cluster 5 cells of Resp-PASC patients showed significantly higher enrichment of LC-CS signature in comparison to cells of both nonResp-PASC and HCs (Fig. 6C).

And in discussion at line 481:

However, the LC-CS signature showed highest expression in patients with clinically severe respiratory LC/PASC symptoms in two independent cohorts, irrespective of their acute COVID-19 disease history. Within the BAL samples, LC-CS-like macrophages showed a pro-fibrotic phenotype and were in higher proportions in patients with Resp-PASC.

And line 520:

It is intriguing, that while the LC-CS signature was predominantly identified in the circulating monocytes of LC-mild-to-moderate patients, it is also present in the myeloid cell populations in the BAL of LC with severe respiratory PASC. In our dataset, we did not see LC-CS in circulation of LC-severe category, except for one patient. It is therefore tempting to speculate that monocytes with the LC-CS signature remain in the circulation and are not actively recruited in the lungs of LC-mild/moderate patients, leading to fatigue and subtle impairment in pulmonary function. While they migrate in the lungs of LC-severe patients, where they contribute to the severity of the lung pathophysiology, upon receiving signals from the tissue micro-environment.

We hope that these two independent analyses, from different cohorts strengthens the link between the LC-CS signature identified in our primary dataset and its link to pro-inflammatory/pro-fibrotic profiles in patients with respiratory difficulties, in alignment with the findings from the published dataset.

Minor

Line 116. Text is missing from the sentence that ends with “characterized by elevated.”

We apologize for this issue. We have re-written this section at line 130 as:

Our findings revealed a distinct long COVID cellular state (LC-CS) within the circulating CD14⁺ monocytes, particularly enriched in patients who experienced mild-to-moderate acute infection.

Line 204 states “ CD14⁺ monocytes in these patients showed elevated ‘PD1 signalling’, suggestive of a low phagocytotic state...”. This is an over interpretation. PD1 is involved in a number of functions in monocytes (including cell death). The role of PD1 in phagocytosis has primarily been suggested in tumor associated macrophages.

We agree with the reviewer’s suggestion and have removed the over-interpretation from the manuscript to ensure a more accurate representation of the data.

The Y axis on Figure 1D is confusing. Have data values been log-transformed? Would it be clearer to display the data using a log scale?

The data were indeed log₂-transformed for each cytokine. We have now added this information to the figure 2 legend

Plot of log₂ (pg/ml) of cytokines showing significant differences compared to healthy controls.

Flow cytometry provides a nice companion to monocyte transcriptional phenotyping. In the supplemental figures gating strategies should be shown. In addition, it will be important to show data for some of the trickier markers (i.e. TGFb1) along with their isotypes. Lastly, it will be important to confirm that isotypes were used as fluorescence minus one (FMO) fashion in which all other markers received true Abs.

We thank the reviewer for these helpful suggestions. In response, we have added histogram plots in **Extended Data Fig 6** (please see below) in the revised manuscript to enhance transparency and reproducibility.

For monocyte gating, we used CD3⁺ cells as a control population. Since CD3⁺ cells do not express monocyte-specific markers, their inclusion allowed us to define negative staining boundaries and confidently gate monocyte subsets in the presence of the same antibody panel.

Regarding the use of isotype and FMO controls, we acknowledge their distinct roles. FMO controls are typically used during panel optimization to evaluate fluorochrome spillover and background fluorescence, while isotype controls are important for determining nonspecific antibody binding, particularly in cells with high Fc receptor expression such as monocytes.

In our study, we focused on intracellular markers including TGFb and IRF8, which are known to be challenging due to potential background staining.

To minimize non-specific binding and maximize marker resolution:

- We included an Fc receptor blocking step prior to antibody staining.

- We designed two independent flow cytometry panels using our spectral analyser equipped with 5 lasers to minimize the fluorochrome overlap
- For key markers identified in transcriptomic analyses (e.g. TGFb1), we incorporated isotype controls to guide accurate gating and confirm the specificity of antibody binding.

Figure R14: Histograms for the flow cytometry analysis of CALR, CD99, TGFb1, HLA-DQ, and CD120b surface expression enriched for all CD14⁺ monocytes from healthy controls (HC) compared to long-COVID patients (LC). HC, LC antibody and isotype stained are shown in blue, red and black, respectively. Shown is a representative donor of biological replicates with similar results.

These plots are added in Extended Data Fig 6 with amendment to main text at line 360

Compared to HCs, LC patients showed a significantly increased expression of HLA-DQ, CD120b, CALR, CD99 and TGF- β 1 markers on CD14 monocytes (Fig. 4H and Extended Data Fig. 6C and 6D).

Reviewer #2 (Remarks to the Author):

In this manuscript, Kumar et al. present the results of a multiomics profiling study performed on blood samples from a prospective cohort of patients with acute COVID-19 or sequelae (“Long-COVID” or PASC), fully recovered patients, or healthy controls, in an effort to uncover features and potential mechanisms of immunopathology in PASC. They performed single cell multiomics profiling using single nuclei RNA sequencing, scATAC-Seq, multi color flow cytometry on PBMC and plasma cytokine profiling. They additionally analyzed scRNA-Seq data from an external cohort of patients with PASC. Here, they integrated scRNA-Seq data from PBMC and bronchoalveolar lavage (BAL) cells.

The main finding is a distinct transcriptional profile in PBMC, particularly in CD14+ monocytes, in patients with PASC after mild to moderate disease during the acute stage of the infection, compared to acute disease or recovered patients. PBMC from patients with PASC after mild to moderate acute COVID-19 displayed an enrichment of genes associated with TGFbeta- and WNT-Catenin signaling. The authors describe a cellular substate in CD14 monocytes that was positively correlated with participants’ fatigue score and negatively correlated with O2 saturation. Moreover, they describe a similarity of this monocyte state (MC4) with BAL macrophages (from PASC patients) that express profibrotic signatures, indicating that this monocyte state may be mechanistically linked to fibrotic responses in patients with respiratory symptoms and PASC.

Overall, this is a complex study that offers interesting insights into transcriptional states in PASC patients. Specifically, distinguishing between PASC patients with acute-to-moderate initial disease and those with initially severe disease is important, since the clinical phenotype and the underlying pathophysiology are likely distinct. However, the study contains a number of inconsistencies and shortcomings, which compromise the overall interpretation and the potential impact of the study. Details are listed below.

We thank the reviewer for their positive assessment of our manuscript.

Specific comments:

1. The authors measured samples from a complex cohort, with both longitudinal and cross-sectional sampling, and participants with distinct outcomes, and controls. Figure 1 attempts to depict the samples from each time point, but it fails to provide a sufficient overview of the composition of the cohort. Suppl. Table 1 adds some information, but the overall description of the cohort is insufficient, and additional information on which samples were taken from the same individuals etc. should be added.

We thank the reviewer for this important comment and apologize for the lack of clarity in the original presentation of the cohort. To address this, we have revised the text and supplemental materials to provide a more comprehensive overview.

Our study design involved both **longitudinal** and **cross-sectional** sampling of participants across multiple clinical categories, including **Long COVID (LC) patients, recovered individuals, and acute COVID-19 cases.**

To improve transparency, we have provided detailed information on the two discovery cohorts in **Table 1 and Table 2 (both in main text)** and now included an expanded **Extended data table 1**, which provides:

- Anonymised donor and sample identifiers
- Classification into clinical categories and time points.
- Availability of clinical metadata measured

We have also updated **Extended Data Fig 1A (Figure R4 on page 10 of this document)** to better reflect the structure of the cohort, sampling schema and number of samples used for each time point in each stratified comparisons for transcriptomics data analysis. Specifically, we performed pairwise comparisons such as T2 vs T1, T3 vs T1, and comparisons of LC timepoints (T2 vs Recov, T3 vs Recov and so on) vs the Recovered group (comprising 8 donor samples).

Information on longitudinal samples is added in the Results section at line 165

“To comprehensively characterize the cellular, molecular and regulatory landscape of LC, we performed single-cell multiome analysis on 78 peripheral blood mononuclear cell (PBMC) samples, incorporating both longitudinal and cross-sectional data from 45 patients and 6 healthy controls (Fig. 1A and **Extended Data Fig 1A**). **These PBMC samples represented 13 patients with longitudinal samples covering at least two time points from the same patients.** The cohort included individuals with ongoing SARS-CoV-2 infection, patients experiencing post-acute sequelae of SARS-CoV-2 infection (PASC/LC) at varying time points (53-311 days post-infection), samples from patients recovered from their LC symptoms (Recovered) and COVID-naïve healthy controls (HC) (Table 1, **Extended Data Fig. 1A and Extended data table 1**).

2. Some notes on the cohort. Some groups differ in size (e.g. n=28 PASC_mild/moderate vs n=8 recovered) and in baseline characteristics (e.g. avg. age 39 yrs in recovered vs. 52 yrs in recovered, and male:female ratio of 2 in recovered and 0.58 in PASC_mild/moderate), making it difficult to clearly attribute transcriptomic differences to the clinical outcome, as opposed to baseline differences. It is also unclear, which participants contributed several samples and how the cohorts were composed at the different time points with regards to demographic baseline parameters. Overall, I feel that it would be more powerful to compare matched cohorts.

We thank the reviewer for this thoughtful and important comment. We acknowledge the inherent limitations in cohort composition, particularly regarding group sizes and baseline demographic differences such as age and sex ratios. However, to address these potential confounding factors, we structured our analyses to mitigate their influence wherever possible.

Specifically, **Long COVID (LC) samples were not only stratified by clinical severity but also across distinct timepoints (T2–T5)**. This approach allowed for more **balanced group sizes**, particularly at timepoints T3, T4, and T5, where each group within LC-mild/moderate includes approximately **10 samples** as also reflected in **Figure R4** (page 10 of this document), compared to **8 samples in the Recovered group**.

While it was not possible to fully control the **sex ratio** in the Recovered group due to limited sample availability, we did perform additional comparisons to acute-phase samples (T1) for

each LC timepoint, where the **male-to-female ratio is more balanced** (4 males and 3 females). Moreover, the **LC-associated cellular state** was consistently enriched **in both male and female** patients with the LC-mild/moderate group (see the figure below), suggesting that sex is unlikely to be the primary driver of the observed transcriptional differences.

Figure R15: Proportions of MC4/LC-CS cells within each donor from LC-mild-to-moderate, stratified based on sex. Each dot corresponds to a sample, the size of dot represents percentage of MC4/LC-CS cells within CD14+ monocytes of the sample.

To improve transparency, we have now:

- **Updated the Extended Data Fig 1A (Figure R4)** to clearly indicate the distribution of samples from unique versus repeated donors at each time point
- **Expanded Extended data table 1** to include donor-specific demographic data (age, sex, timepoint and measured clinical parameters)
- Added a clarifying description of the **Methods section** line 718 and referenced the new Extended Data materials accordingly as also described on pages 9-10 of this document.

And we have added Figure R15 in Extended Data Fig 5A with mention in text at line 299

“In both comparisons, LC samples particularly from mild-to-moderate acute infection showed a significantly increased abundance of neighbourhoods in the MC4 cluster, **in both males and females** (Fig. 3E and **Extended Data Fig 5A**).”

While we agree that matched cohort analyses offer stronger control over confounders, such matching is often impractical in real-world longitudinal studies involving post-acute infection syndromes. Nonetheless, we believe our stratified approach and additional validation steps help ensure the robustness of our findings.

3. In lines 161/162 the authors state: “In all cell subsets, genes upregulated during acute infection were markedly downregulated in LC samples from individuals with mild to moderate

acute infection.” While there is a general pattern, there is clearly also a cluster of genes in the top of the ‘Day 292-311’ column (Figure 2B) that appear to be upregulated with some overlap to the acute infection.

We thank the reviewer for this valuable observation. We agree that while the overall pattern shows downregulation of acute-phase genes in LC samples with mild/moderate initial disease, there is indeed a subset of genes — particularly in the Day 292-311 samples — that show overlapping upregulation with those observed during acute infection.

This partial overlap may reflect heterogeneity within the LC cohort at later timepoints. Notably, the majority of Day 292-311 samples are from **cross-sectional collections**, and it is possible that some individual exhibit **persistent or reactivated inflammatory programs** resembling those seen during acute infection.

We have revised the manuscript accordingly at line 192 and at line 201 to clarify this point and acknowledge this nuance in the data.

“In all cell subsets, genes upregulated during acute infection were markedly downregulated in LC samples from individuals with mild-to-moderate acute infection, **until Day 191-251**.”

This late time point also showed upregulation of a subset of genes overlapping acute infection profile.

4. In lines 166-171 the authors state: “Specifically, differential gene expression analysis in, CD14+ monocytes revealed distinct upregulation of 1737 genes in LC samples, particularly in individuals with mild-to-moderate acute SARS-CoV2 infection, compared to samples from acute infection, recovered patients and HCs. Notably, the upregulation of these genes intensified over time, indicating persistent and progressive alterations in the gene expression profile of CD14+ monocytes in LC. The persistent upregulated genes are associated with both pro-inflammatory (e.g., CSF1, IRF8, RELA, NOTCH1) and anti-inflammatory (e.g., TGFB1, SMADs, ENG and SERPINE1/PAI-1) functions.” Contrary to what is stated here, looking at the data in Figure 2B, it seems that differential gene expression is rather transient and the transcriptional changes observed at Day 101-170 and Day 191-251 are largely reversed by Day 292.

We thank the reviewer for this important observation. A similar concern was raised by Reviewer 1 (Comment 3b), and we refer to our detailed response **on page 11-13**.

In brief, while the LC-CS transcriptional signature is most prominent at Days 101-170 and Day 191-251, several lines of evidence support its biological relevance and persistence:

- Key genes such as *TGFB1* and *CALR* show consistent upregulation across all time points and were validated at the protein level in an independent LC cohort.
- The signature is also detectable in some individuals as early as Day 53–78 and as late as Day 292–311, suggesting temporal heterogeneity.
- We independently replicated the LC-CS monocyte signature in a new cohort (Cohort 3) and observed enrichment of these cell states in T5 samples in primary cohort (Cohort 1).
- The observed fluctuations likely reflect variability in long COVID disease trajectories. Additionally, low cell yields at T5 may have limited detection of rare LC-CS cells in pseudobulk analyses.

Together, these results support the LC-CS signature as a dynamic yet biologically meaningful feature of mild/moderate long COVID, with its variability shaped by both patient-specific progression and technical constraints.

We have clarified these points in the revised manuscript, at line 200 and added the revision figures in Extended Data Figs 1D,1E and Figure 3E

“Notably, the upregulation of these genes intensified over time (Figure 2B and Extended Data Fig 1B), indicating persistent and progressive alterations in the gene expression profile of CD14⁺ monocytes in LC up to Day 191-251, while the signature appears subdued in samples from Day 292-311, possible due to lower number of cells (Extended Data Fig 1C), the persistent upregulated genes across all time points are associated with both pro-inflammatory (e.g., *CSF1*, *IRF8*, *RELA*, *NOTCH1*) and anti-inflammatory (e.g., *TGFB1*, *SMADs*, *ENG* and *SERPINE1/PAI-1*) functions (Extended Data Fig 1D).”

We have also included the T5 vs Recovered neighbourhood enrichment in the main figure 3E with the text in the Results section at line 296.

Given the well-defined and significant role of monocytes in acute SARS-CoV-2 infection^{31,32}, we next performed differential neighbourhood analysis and compared LC samples to acute and recovered samples at all time points (Fig. 3E and Extended Data Fig. 5A-5B).

5. Along these lines, which time point was analyzed in Figure 3F?

We thank the reviewer for this question and apologize for the lack of clarity. Below, we clarify the analyses presented in Figure 2F and 3F:

Figure 2F:

This figure includes **all samples** from **all time points** for both LC and recovered individuals. For each sample and cell type, we computed a pseudobulk gene expression profile, and subsequently calculated AUC scores for the selected pathways, each composed of approximately 40-80 genes. We then assessed the Spearman correlation between these pathway scores and the corresponding FAS score across all samples. This analysis was designed to evaluate the association between transcriptional pathway activity and fatigue severity, independent of disease stage or time point.

Figure 3F:

Here, we calculated **AUC scores at the single-cell level**, regardless of time point, to assess pathway activity across the cell cluster defined in Figure 3C. This approach enabled us to examine transcriptional heterogeneity within monocyte subpopulations, independently of longitudinal sampling.

We have updated the manuscript Methods section to include these clarifications at line 752.

Pathway correlations with clinical parameters.

For each sample and cell type (CD14 monocytes, CD8 T and NK cells), we computed pseudobulk gene expression profiles. Subsequently, AUC scores for the selected upregulated immune pathways as described in Figure 2C were calculated in each sample. The AUC scores of these pathways were then correlated with clinical parameters using spearman correlations. Only significant correlations were plotted.

And at line 264 in Results section.

These correlations performed without patient stratification, suggest a gradient of association of these pathways across the clinical spectrum, rather than patients alone.

6. Also, the authors state that TGF-beta signaling and WNT-beta catenin signaling pathways in CD14+ monocytes were positively correlated with FAS scores (lines 243-244). They should acknowledge that the correlations were rather weak (R=0.28).

We thank the reviewer for this important observation. We agree that the correlation values reported in Figure 2F (e.g. R=0.28) are modest in strength, though statistically significant. Notably, these pathways (e.g., TGF- β and WNT/ β -catenin signalling) were among the few that showed any meaningful correlation with FAS scores, while other pathways exhibited weaker or non-significant correlations (data not shown).

It is important to note that the correlation in Figure 2F were derived from pseudobulk gene expression profiles of entire cell subsets, such as CD14⁺ monocytes. However, as we demonstrated in later figures, the increased pathways activity is largely confined to a specific monocyte subpopulation (MC4 cluster/LC-CS). Since pseudobulk analysis averages the signals across all monocyte states (MC1-MC4), this likely attenuates the correlation signal observed at the bulk level. We used this approach as a first-pass, cell type- level analysis to explore potential links between pathway activity and FAS score.

We have now acknowledged the limited strength of the correlation and clarified this point in the revised text at line 255

Interestingly, FAS scores showed modest positive correlations with both 'TGF- β signalling' and 'WNT- β catenin signalling' pathways in CD14⁺ monocytes (Spearman R = 0.28, 0.31; p value = 3.6×10^{-2} , 1.8×10^{-2} respectively) (Fig. 2F).

7. Figure 3C shows GSEA for CD14 monocytes and NK cells for 3 different time points in the comparison with acute disease ("Vs Acute") and 2 time points in the comparison with Recovered ("Vs Recov), missing "Day 53-78". Yet, for CD8 T cells there are 3 time points in both comparisons. What is the reason? Also, it would be important to also compare this with Healthy controls.

We thank the reviewer for this helpful comment and apologize for the lack of clarity in the manuscript.

In Figure 2C, our initial aim was to examine the dynamics of immune-related pathways in LC samples in comparison to acute infection. Under the hypothesis that most immune pathways

in LC samples would diminish post-acute infection. This approach allowed us to highlight pathways persistently or selectively upregulated in LC.

We then included comparisons with recovered individuals to determine whether these same immune pathways, or additional ones, remained elevated specifically in LC samples.

Because the transcriptomic signature was most prominent at the T3 and T4 time points, these comparisons yielded the most robust enrichment of immune-related pathway. In contrast, T5 samples—particularly in CD14⁺ monocytes and NK cells—contained fewer cells per sample (as previously discussed in response to Comment 4), which reduced statistical power and limited pathway detection. For CD8⁺ T cells, T2 was included in both comparisons (vs. Acute and vs Recovered), as sufficient cell numbers, resulting in significant enrichments were retained across time points. We calculated pathway enrichments against the entire HALLMARK and REACTOME pathway datasets, which yielded an extensive list of significant pathways. For clarity and focus in the main figure, we presented only a curated subset of significantly enriched immune-related pathways from the whole list. Therefore, comparisons for time points without statistically significant enrichment for these immune pathways were not plotted.

However, we provide a comprehensive list of all significant pathway enrichments (including those from T2 and T5 vs Recovered, all major subsets, and additional comparisons stratified by patient group, including comparisons against healthy controls— in the **Extended data table 2**. This resource allows the reader to fully explore the breadth of pathway dynamics observed across the dataset.

We have clarified this rationale and added a reference to Extended data table 2 in the Methods section at line 748.

Pathway analysis per cell type

Pseudobulk of each donor per time point was calculated, followed by similar comparisons as described in previous section using DESeq2 were done. Gene Set Enrichment Analysis (GSEA) using HALLMARK and REACTOME pathways as background was calculated using fgsea R package. Further, immune related pathways showing statistically significance for any comparisons were plotted. The whole list of statistically significant pathways resulting from all comparisons is added in Extended Data table 2.

8. Figure 3 and some of the interpretations are problematic and inconsistent. The authors describe four monocyte clusters MC1-MC4 within the monocyte space and found MC4 to display “higher activity in “TGF-beta signalling” and “WNT-beta catenin signalling”. They state that LC (PASC) samples from severe initial disease did not enrich for MC4, which is also evident in Suppl. Figure 4. In contrast, in Figure 3G, where the authors correlated MC cluster abundance per donor with FAS scores, it is clear that the majority (if not all!) donors with high percentages of MC4 of the CD14 monocytes were in fact “LC-severe” (=dark red dots). This does not square with some of the previous statements that showed a specific transcriptional program in LC-mild/moderate compared to LC-severe. It is also not in line with the results shown in Figure 4C.

We thank the reviewer for carefully pointing out this important inconsistency, and we sincerely apologize for the confusion it caused. Upon thorough review of our data and code, we discovered that the group colours in Figure 3G were mistakenly assigned due to a bug in

the plotting script—specifically, the group labels were randomly applied rather than accurately mapped to their respective patient categories for this figure alone, as defined in the legend.

We have now corrected the issue and the updated version of figure 3G accurately displays the correct group colours (see the figure below). As expected, donors with the highest proportion of the MC4 cellular state are predominantly from the LC-mild/moderate group (now correctly shown in bright red). This correction restores consistency with our earlier findings, including those shown in updated Extended Data Fig 5D, and supports our interpretation that the MC4 transcriptional program is “predominantly” observed in the LC-mild/moderate subgroup.

We would also like to clarify that this was solely a visualization error. We have also uploaded the corrected code on github. The correlation coefficients shown in the figure were calculated using all samples, regardless of group assignment, and are therefore unaffected by the coloring issue. As such, the conclusions drawn from these analyses remain valid.

We appreciate the reviewer’s close attention to detail, which enabled us to identify and correct this issue.

Figure R16: Updated Main Figure 3G with correct color labels

9. Similar to the previous comment, Figure 4H (left most panel, “TGFB1”) suggests that TGFB1 was indeed increased in the ‘LC-severe’ samples, which contradicts prior statements that this was a specific feature of ‘LC-mild/moderate’.

We thank the reviewer for this insightful comment. Figure 4H represents protein-level expression of TGFB1 measured in an independent validation cohort of both mild-to-moderate and severe patients, compared to Healthy Controls.

We agree that TGFB1 alone, at protein level, does not completely distinguish between LC-severe and LC-mild/moderate groups. Rather, it reflects one component of a broader transcriptional and functional cellular program. As noted in our main analysis, the LC-associated cellular state (LC-CS), characterized by coordinated activation of multiple genes

and pathways—including but not limited to TGFB1—shows differential enrichment based on initial disease severity.

We have now clarified this point in the Discussion section at line 514 to better reflect this distinction between individual protein markers and the composite transcriptional signature.

While the flow cytometry results showed that all LC patients expressed higher levels of TGF- β 1, we showed that it is the LC-CS that segregates the transcriptomic imprint of LC patients.

10. It is also not clear when LC samples were compared to samples from patients with acute disease or those who had recovered, like in Figure 2C, and when they were compared to healthy controls, like in Figure 4G-I. This should be largely consistent throughout the study or specifically stated.

We thank the reviewer for pointing out this important clarification. We apologize for the confusion regarding the comparison groups used across figures.

Figure 4F-I refer to an independent validation cohort, distinct from the single-cell and cytokine profiling cohorts. In this validation cohort, majority of healthy controls were individuals with a history of SARS-CoV-2 infection but without Long COVID symptoms. Therefore, for these analyses, the comparison allowed us to focus on features specific to persistent symptoms in LC samples rather than general post-infection effects.

In contrast, in **Figure 2C** and related analyses, particularly involving all transcriptome datasets, comparisons were made to either acute or recovered patients and also examined against COVID-naïve healthy control group. This approach was chosen to better delineate the trajectory of immune recovery or persistence of transcriptional alterations following infection.

We have revised the manuscript to clearly state the rationale behind the selection of comparison groups in each section and to ensure consistency in terminology and context. This clarification has been added exclusively to every Results section mentioning comparisons at line 239

Among the cytokines measured, CXCL11/ITAC, CCL2/MCP1 and TNF α showed persistently elevated expression in LC samples compared to HCs (COVID-naïve) up to more than 8 months post-infection.

And also, at line 345

Having characterised the LC-CS within the CD14⁺ monocytes, we next evaluated the protein expression of marker genes associated with LC-CS. We performed flow cytometry staining from an independent set of 40 LC patients and 10 HCs (COVID-experienced).

11. In Figure 6J,K the authors analyzed the response of PBMC from patients and recovered donors. They categorized the donors based on the content in CD14 monocytes with predefined features that the authors had termed LC-CS (Long COVID cellular state) into those containing >10% or <10% of LC-CS cells. They then found differences with regards to TGF-beta signaling, WNT-beta signaling and other pathways that were previously identified as defining features of LC-CS. When categorizing samples into those with a high and low content of cells with predetermined features and then finding differences in these exact features it seems like “self-fulfilling”. It would make much more sense to compare functional responses of PBMC from recovered patients to LC / PASC patients and healthy controls.

We thank the reviewer for their keen observation and their valuable suggestion.

Our intention in this section was to explore the functional behaviour of monocytes with LC-CS (Long COVID-associated cellular state) features **in response to an external stimulus** (*Pseudomonas* stimulation), and to gain mechanistic insight into the immune response associated with this cellular phenotype. To this end, we classified donors based on the proportion of CD14+ monocytes with LC-CS characteristics (>10% vs <10%), and compared their transcriptional responses in both unstimulated and stimulated conditions. The pathway analysis shown in figure 6K with enrichments of WNT-beta signalling and TGF-beta signalling **observed in the un-stimulated samples, were plotted to merely show comparison of before and after stimulation differences**. Particularly, this analysis highlights the functional response differences post-stimulation, compared to un-stimulated condition.

We acknowledge the reviewer's concern that comparing predefined LC-CS high vs. low groups for pathways previously used to define LC-CS may introduce circular reasoning. However, our analysis aimed to examine whether the presence of LC-CS cells predicts functional immune alterations **under stimulation**, rather than merely recapitulating marker expression as done in previous analysis (Figure 3 and Extended Data Fig. 5). Of note, **all recovered donors fell into the LC-CS low category**, highlighting a biologically relevant distinction.

Notably, this analysis showed marked downregulation of pathways such as interferon signalling and upregulation of regulatory markers (e.g. *HMOX1*), specifically upon stimulation in the LC-CS high group. Our results in this section show that the dysregulated downstream interferon signalling persists **even in the stimulated monocytes** of the LC-CS high donors as shown in Figure 6K. In contrast, within LC-CS low groups, IFN signalling genes are upregulated in response to stimulation. This suggests that patients with higher proportions of LC-CS, experience high fatigue and may also have a compromised functional immune response to an infection.

As per reviewer's suggestion, we performed an additional analysis comparing PBMC responses between LC-mild/moderate patient's samples and recovered donor samples, independent of LC-CS classification. This analysis similarly revealed a downregulated inflammatory, interferon gamma signalling response and downregulation of IL10 signalling **upon stimulation** in the LC group. Additionally, downregulation of IL6 JAK STAT3 signalling, antigen presentation pathways was also found in LC group, while cell cycle-based pathways were upregulated. Please see below:

Figure R17: Pathway enrichment of stimulated samples only, in comparison of LC-mild-to-moderate vs Recovered.

These complementary approaches allowed us to evaluate both the impact of LC-CS cell abundance and the overall group-level functional differences between LC and recovered individuals. We have now included these new results in the revised manuscript as Extended Data Fig 9E, to show both comparisons at line 442.

To further investigate on the functional implications of LC-CS in response to immune insult, we stimulated PBMCs with heat-inactivated *Pseudomonas aeruginosa* for 4 hours and generated single cell multiome datasets (Fig. 1A, Extended Data Fig. 8A and 8B). Upon stimulation, we observed a reduction in the total number of monocytic myeloid cells and CD14⁺ monocytes (Extended Data Fig. 8C and 8D), consistent with findings from other studies^{52–54}. We performed joint analysis of stimulated and unstimulated samples using two approaches (Figure 6I). First, we compared the stimulated CD14⁺ monocytes from LC-mild-to-moderate and recovered samples. Pathway analysis of this comparison showed significant downregulation of “Inflammatory response” and “Interferon gamma signalling”, “IL10 signalling”, “cytokine signalling” and “IL6 JAK STAT3 signalling” (Extended Data Fig 9E).

And line 449

All recovered samples were within the low LC-CS donors’ category, along with samples from LC-severe category.

12. Finally, the fact that the authors find similarities between blood monocytes from PASC patients and BAL macrophages from PASC patients is interesting. However, I wonder whether these are actually physiologically linked. The authors have previously argued that LC-CS (or MC4 etc.), with elevated TGF-beta and WNT-signaling are a specific feature of ‘LC-

mild/moderate', hence, LC patients with a mild disease course. These are more likely to experience fatigue rather than dyspnoea and reduced oxygenation, which is more frequently observed after severe COVID-19. This is also evident in their data (Suppl. Table 1). BAL samples from LC/PASC patients were likely taken from patients with respiratory symptoms ('LC-severe'). So, if the 'LC-CS' / MC4 is found primarily in 'LC-mild/moderate' how does this connect to profibrotic macrophages and respiratory PASC in 'LC-severe' patients?

We thank the reviewer for this important comment, which underlines the complexity and heterogeneity of the Long COVID (PASC) phenotype. In our work, we identified the LC-CS/MC4 monocyte population as pre-dominantly enriched in the peripheral blood of LC patients who experienced a mild/moderate acute COVID-19 disease course. The mild/moderate classification of LC patients is solely based on their acute infection WHO score and does not indicate the LC disease severity. This cellular state was characterized by elevated TGF- β and WNT signalling, among other features. Notably, this signature was also detected in PBMCs (Figure 6B) and BAL myeloid cells (Figure 6F) from an independent, previously published cohort. As the reviewer correctly points out, it is indeed that in the published study BAL samples were collected from hospitalized patients. However, severity of LC symptoms is not exclusively dependent on acute COVID-19 disease history, as described below.

In this revision, we introduce a new cohort (cohort 3) (Figure R1 on page 2 of this document), which consists of patients from LC-mild-to-moderate COVID-19 disease history but show severe LC symptoms (more information on page 5 of this document). These patients show multiple domains affected, with three patients showing bronchial hyperresponsiveness (BHR), while all patients showed fatigue and dyspnoea. Therefore, we classified the 3 patients with BHR as respiratory-PASC patients, while others were classified as nonrespiratory-PASC patients. We next performed scRNA-seq analysis of circulating monocytes from these patients. As shown in figure R9 (page 16) and figure R12 (page 20) of this document, all these LC patients showed enrichment of LC-CS signature, while respiratory-PASC patients showed highest expression (single cells scoring high AUC scores) compared to nonrespiratory-PASC. This provides evidence that the LC-CS, although mostly enriched and identified in LC-mild/moderate category is linked to fatigue and dyspnoea. But in patients with respiratory PASC, it is also linked with the presence/ emergence of LC-CS. We cannot exclude however that this signature may also be present in BAL from 'LC-mild/moderate', but unfortunately that data is not available.

In the re-analysis of the published dataset (Cohort 5), we included HCs that were also available from the study. We found that cluster 5 of PBMC monocytes—enriched for the LC-CS signature (Figure 6A and 6B) was particularly showing higher levels in patients with respiratory PASC, patients with abnormal lung function test parameters (Figure R13 of this document). The same LC-CS signature was not as highly expressed in non-respiratory PASC or healthy controls. This observation, along with observations from the new cohort suggests a broader relevance of this cellular state in Respiratory-PASC pathophysiology, beyond initial disease severity stratification.

It is possible that this cellular state may also exist in LC-severe individuals, particularly with Respiratory PASC phenotype, in the lung, even if not detected at high frequency in peripheral blood. We speculate that in LC-severe patients, monocytes exhibiting the LC-CS/MC4

signature actively migrate to lungs, where they persist and acquire additional functional programs in response to the local microenvironment.

Consistent with this idea, we observed that the LC-CS transcriptional signature correlates with fatigue scores across all samples, regardless of initial COVID-19 severity (Figure 2F and 3G). Furthermore, mild/moderate LC patients in our cohort also experienced dyspnoea, suggesting that this immune cell phenotype is associated with both systemic symptoms (e.g., fatigue) and possibly mild respiratory impairment.

This also supports the hypothesis that LC-CS-like cells have the potential to acquire a profibrotic phenotype upon entering the lung and encountering inflammatory cues, including T cell-derived signals, as described in the original publication. Previous studies reporting respiratory PASCs have similarly reported the role of monocyte derived profibrotic macrophages in the lungs contributing to pulmonary fibrosis. Taken together, our findings suggest that the LC-CS state may serve as a systemic precursor to profibrotic monocyte/macrophage activity in the lungs, particularly in LC-severe patients. In LC-mild/moderate patients, in contrast, this state may remain confined to the circulation and contribute to fatigue and dyspnoea. Importantly, LC-CS may play a role in patients with Respiratory-PASC, irrespective of their acute COVID-19 disease history.

We have revised the Discussion in the manuscript to reflect these clarifications at line 520

Higher expression and proportion of these cells in Resp-PASC patients links these cells to chronic inflammation and fibrosis in the lungs. It is intriguing, that while the LC-CS signature was predominantly identified in the circulating monocytes of LC-mild-to-moderate patients, it is also present in the myeloid cell populations in the BAL of LC with severe respiratory PASC. In our dataset, we did not see LC-CS in circulation of LC-severe category, except for one patient. It is therefore tempting to speculate that monocytes with the LC-CS signature remain in the circulation and are not actively recruited in the lungs of LC-mild/moderate patients, leading to fatigue and subtle impairment in pulmonary function. While they migrate in the lungs of LC-severe patients, where they contribute to the severity of the lung pathophysiology, upon receiving signals from the tissue micro-environment.

13. Overall, the data appears a bit contradictory regarding the described dichotomy between transcriptional features in 'LC-mild/moderate' vs 'LC-severe' and the authors may want to clarify and solidify this a bit more.

We thank the reviewer for highlighting this important point regarding the apparent dichotomy in transcriptional features between 'LC-mild/moderate' and "LC-severe' patients. We appreciate the opportunity to clarify and have revised the manuscript to provide a more cohesive narrative that reflects the complex and evolving nature of LC pathophysiology.

While the LC-CS transcriptional signature is predominantly enriched in PBMCs of LC-mild/moderate patients, we agree that certain elements appeared inconsistent or underexplained. We now propose a refined framework in which:

1. The LC-CS state represents a **circulating monocyte phenotype** linked to fatigue and subtle respiratory dysfunction, particularly prevalent in LC-mild/moderate patients.
2. In LC-severe patients, we hypothesize that LC-CS monocytes **are actively recruited** to lungs and may transition into more pro-fibrotic states upon recruitment to sites such as the lungs—driven by local microenvironmental cues.
3. This model explains the differential detection of LC-CS in PBMCs across patient groups and integrates symptom severity with transcriptional and functional changes

To support and clarify this framework, we have added the following analyses:

- **Temporal signature consistency:** We now include a plot showing genes consistently upregulated in LC-mild/moderate CD14+ monocytes across all time points, highlighting the persistence of the LC-CS signature. We also show the dynamic nature of cellular state development in different patients. (Details can be found on pages 11 - 13 of this document).
- **Cellular Neighbourhood and Trajectory Analyses:** We performed additional comparisons of timepoints T2 and T5 versus Recovered controls (similar to Figure 3E), confirming a sustained enrichment of LC-CS-like monocyte neighbourhood at later timepoints (T5). (Details can be found on page 15 of this document); In addition, we performed a trajectory inference analysis independent of patient stratification. This identified a Long COVID-associated trajectory (Lineage 3) predominantly populated by LC-mild/moderate cells, though one LC-severe patient also showed partial enrichment, highlighting biological overlap. (Details can be found on page 19 of this document)
- **Independent validation in new cohort of LC patients:** We show LC-CS in monocytes of all new LC patients with mild COVID-19 disease history (Cohort 3). Additionally, patients with Respiratory PASC (patients with BHR) showed significantly higher enrichment of this cellular state compared to non-Respiratory PASC patients (Details can be found on page 16 and 20 of this document).
- **Respiratory PASC association in public data:** In the public dataset (Cohort 5), PBMCs from respiratory PASC patients showed significantly higher LC-CS signature enrichment than both HCs and non-respiratory PASC patients, similar to observations in Respiratory PASC patients from LC-mild/moderate group. (Details can be found on page 20,21 and 34 of this document)
- **Functional evidence:** We directly compared stimulated PBMCs from LC-mild/moderate patients with those from Recovered controls. This revealed a reduced inflammatory and interferon gamma signalling response in LC patients, consistent with a functional altered monocyte state. (Details can be found on page 34 of this document)
- **Clarified Controls & Consistency:** We have clarified and standardized the comparison groups throughout the manuscript using Recovered patients and COVID- experienced HCs where appropriate to remove confounding COVID-19 imprinting and better isolate LC-specific signatures. (Details can be found on pages 32-33 of this document); We have corrected the colour coding in Figure 3G to accurately reflect the patient group. This resolves the inconsistency the reviewer helpfully pointed out. (Details can be found on page 31 of this document)

- **Sex-independence of LC-CS:** We have shown that the LC-CS signature is not sex-dependant and is observed in both male and female patients. (Details can be found on page 26 of this document)
- **Technical Clarification:** We note that low cell yields at T5 may have limited pseudobulk detection of rare LC-CS cells. Nevertheless, we showed enrichments of LC-CS is also observed at T5 in comparison to Recovered (Details can be found on page 13 and 15 of this document).

Together, these clarifications strengthen the interpretation of LC-CS as a central, dynamic monocyte program in long COVID. This circulating state may evolve differently in tissues in severe cases. This integrated view aligns with the diverse clinical manifestations of LC and resolves the perceived dichotomy.

We have updated the manuscript text accordingly in several sections of both the Results and Discussion to incorporate the revisions, particularly on pages 8, 9, 10,11, 12 and 13 of the manuscript

We sincerely thank the reviewers for their thorough and thoughtful evaluation of our manuscript. We are encouraged by their positive assessments, including comments such as:

- *“Over all the findings of the paper are interesting and should add to our knowledge regarding the pathogenesis of long Covid. The paper has a number of strong elements and has potential for impact. (Reviewer. #1)”*
- *“this is a complex study that offers interesting insights into transcriptional states in PASC patients. Specifically, distinguishing between PASC patients with acute-to-moderate initial disease and those with initially severe disease is important (Reviewer. #2)”*

We are particularly grateful for the constructive suggestions that helped us further improve and refine the manuscript.

Below, we provided a detailed point-by-point response to each comment.

- **Reviewers' comments** are presented in **black**,
- **Our responses** are in **blue**
- **Revisions made to the manuscript** are highlighted in **purple**

REVIEWERS' COMMENTS:

Reviewer #1 (Remarks to the Author):

In the paper entitled “A distinct monocyte cellular state links systemic immune dysregulation to pulmonary impairment in long COVID” (ref. num. NI-A39964-T), Li and colleagues perform a large number of transcriptomal and functional studies on peripheral blood monocytes from a large number of patients with long Covid. The paper uses complementary techniques to derive a signature for cells that are enriched in patients with long Covid, and suggests that signatures for patients who had mild/moderate acute disease may be different than ones from patients that had severe acute disease.

Over all the findings of the paper are interesting and should add to our knowledge regarding the pathogenesis of long Covid. The paper has a number of strong elements and has potential for impact. However, there are a number of limitations that need to be addressed.

We thank the reviewer for the positive assessment of our study.

Major comments.

1. The paper is difficult to follow. In part, this arises from the overabundance of data and lack of focus. For example, the first part of the paper delves into differences observed over time in long COVID patients and further stratifies these differences based on initial severity of COVID infection. This seems like it could be a major focus / message of the paper. However, as the paper progresses this theme is lost entirely, and a new focus on respiratory vs non respiratory long COVID phenotypes emerges. The focus of the paper is further diluted by large paragraphs in the results section that refer to Supplementary Figs that don't always relate to the core theme. As a result, the main message of the story becomes convoluted. While it is

important to acknowledge the abundance of data and analysis that are included in the paper, much of it detracts from the story rather than adding.

We thank the reviewer for this thoughtful and constructive feedback. We sincerely apologize for the lack of clarity and narrative cohesion in the previous version of the manuscript. In response, we have made significant revisions to improve focus, readability and alignment with the core theme of the study. Specifically:

1. We provide detailed information on all cohorts in Figure 1 (spread throughout in Figures R2, R3 and R14 on page 6,10 and 25 of this document) in the manuscript to more accurately depict cohort characteristics, different data modalities, distribution of LC samples, symptoms, clinical tests and COVID-19 disease severity. We updated the LC sampling from Days to months, to provide an easier interpretation. **We have also introduced a new cohort (Cohort 3), to provide a clearer view of how our findings (LC-CS), identified using patient stratification link with the respiratory PASC phenotype.** Cohort 3 was carefully selected to recruit patients with mild COVID-19 infection history. A few of these patients also showed Bronchial Hyperresponsiveness (BHR), thereby representing respiratory PASC with lung function abnormality (more clinical information on page 5 of this document). In summary, we found LC-CS within all LC patients of this cohort (Figures R9 on page 16 of this document). Additionally, patients with BHR also showed higher expression of LC-CS signature compared to other LC patients (R12 on page 20 of this document).
2. We also added a new **overview figure (Figure 1B/**Figure R1 in this document), which outlines the structure of the manuscript and clearly maps each figure to the corresponding research objectives and analyses. We have re-structured the manuscript to outline the flow of the study with rationale for each section in bold:
 - a. **Exploration-** which included exploring the single-cell multiome datasets (Cohort 1) and cytokine dataset (Cohort 2) to identify the **correlates of systemic inflammation in LC** (Figure 2). The correlates from single cell data led us to **identify the LC-CS** in circulating monocytes (Figure 3).
 - b. **Characterization-** We next **validated** this cellular state in an independent cohort of LC patients (Cohort 3), showing higher expression of LC-CS signature in Respiratory PASC patients with BHR and performed flow cytometry analysis of key markers of LC-CS (Cohort 4). Findings from both of these analyse are included in Figure 4. We also characterized the **epigenetic regulation** of the LC-CS (Cohort 1) in Figure 5.
 - c. **Function-** We leveraged the published data scRNA-seq data of paired PBMC and BAL samples to examine the **role of LC-CS in the tissue** (Cohort 5) and found that the macrophages with LC-CS signature showed a pro-fibrotic signature in BAL. We also examined the **functional immune response** of patients with a higher proportion of LC-CS within their PBMC upon an immune insult (Cohort 1 subset). All findings for this part are summarized in Figure 6.

Figure R1: Schematic of the analysis flow (now included as Figure 1B in the revised manuscript).

- 3 We also substantially revised the content to streamline the narrative, improve logical flow and relocated extended analysis and descriptions to Extended Data. We clarified the relevance of all remaining Extended Data Figs, ensuring each supports a key conclusion. Below we highlight some of the major changes of each section

Introduction:

- a. By including the rationale for patient stratification, leading to our hypothesis.
- b. Improving the overall flow of introduction.

Results:

- a. Added extensive information regarding distribution of LC samples across different cohorts, along with range of symptoms and clinical tests per patient for each sample performed.
- b. Substantially removed content that deviated from main message by removing cell-cell communication analysis and moving pseudotime trajectory analysis to Extended Data, reducing the whole section to only a short paragraph.
- c. Added cohort 3 analysis, showing how the findings (LC-CS) from patient stratification link to LC and particularly implied in Respiratory PASC phenotype.

Discussion:

- a. Rearranged text to conclude results and implications of each finding in the context of LC disease.

Methods:

- a. Added new methods section on patient-sample stratification performed for the analysis.

We hope that with these changes, the focus of the paper is improved and allows a better follow-up of all the analysis, rationale and interpretations of the study. For this general comment, we have provided a comprehensive response without referencing specific line numbers, as the improvements are distributed across multiple sections of the text. Nevertheless, the changes described above are specified individually with line numbers in manuscript with replies to the comments below.

2. A potential strength of the paper is the richness of the clinical cohort. However, details about the cohort are not described in sufficient detail to enable the reader to follow key elements in the manuscript. This weighs down the paper. Items to consider include the following.

We thank the reviewer for their thoughtful comments. We have addressed each point in the revised manuscript, as detailed below.

- a. A Table describing the cohort should appear in the main body (rather than the supplement)

In response, we have included detailed descriptions of all cohorts (1-5) in main new figure 1. As per editor's suggestion, we added new figures depicting each cohort's sampling time (in months), acute COVID-19 disease severity, LC symptoms, clinical tests and distribution of LC sample per patient in Figure 1 (R2 on page 6, R3 on page 10 and R14 on page 25 of this document). We have included all the demographic and clinical tests summary for all the cohorts in Supplementary Material as Supplementary Table 1-4. Since cohort 5 is data is collected from a published study (PMID: 39018367), we have summarized its table in Supplementary Table 5.

- b. Criteria used to diagnose long Covid should be clearly stated in the text. (Of note, the only reference that speaks to long Covid diagnosis is #21. I could only find the text in German. This limits a large portion of the general readership)

We thank the reviewer for pointing this point and apologize for the lack of clarity. Below, we provide a summary of the diagnostic criteria for Long COVID used in our study.

Patients diagnosed with severe cases of the disease were offered appointments at the outpatient clinic, provided they were still experiencing symptoms. Patients with a mild-to-moderate course of the disease also reported at the outpatient clinic if they had symptoms after recovering from a previous infection with the virus. All reporting patients were polymerase chain reaction (PCR) tested for SARS-CoV-2 infection. Upon a negative PCR result, a diagnosis of exclusion was performed, in which other diseases that could explain the

symptoms were ruled out. Therefore, the diagnosis was made in accordance with the S1 guidelines and the Delphi Consensus Criteria.

From the S1 guidelines criteria:

Based on the Cochrane Rehabilitation Review and an international Delphi conference with the participation of the WHO, one of the following three categories can be used to diagnose PCS syndrome:

- 1) Symptoms that persist after acute COVID-19 or its treatment,*
- 2) new symptoms that appear after the end of the acute COVID-19 phase, but can be understood as a consequence of the SARS-CoV-2 infection,*
- 3) Worsening of a pre-existing illness as a result of a SARS-CoV-2 infection.*

Three different patient populations were covered. The overall group of all patients with Post-Covid Syndrome (PCS), who can be significantly impaired in their participation in school, education, work and other social life:

- 1. Patients who have received intensive care treatment for COVID-19 and suffer from postintensive care syndrome (PICS).*
- 2. Patients who develop secondary diseases such as cardiovascular complications, cognitive impairment or a mental disorder as a result of a SARS-CoV-2 infection with a time latency,*
- 3. Patients with symptoms of fatigue and exercise intolerance with/without dyspnea and neurocognitive disorders ("brainfog").*

We have included the information pertaining the LC diagnosis criteria at line 147:

“Inclusion was based on the following criteria: (1) symptoms persisting beyond the acute phase of SARS-COV-2 infection or its treatment; (2) new symptoms emerging after resolution of the acute illness and are attributed to the prior infection; and (3) exacerbation of pre-existing conditions as a consequence of SAR-CoV-2 infection. These above-mentioned criteria were used for recruitment of cohorts 1-4. Cohort 5 represents published dataset²³.

c. Of the patients diagnosed with long-COVID what domains are the ones most impacted? I realize this information can be difficult to gather. However, if this can be achieved it would help the reader contextualize which organs / physical states are affected in the long-Covid cohort. This is relevant since later in the paper, data from an independent cohort with respiratory vs non-respiratory long-Covid are analyzed.

We thank the reviewer for this insightful comment. We fully agree that identifying the affected domains in LC patients is crucial for contextualizing disease heterogeneity, particularly in light of our comparative analyses involving respiratory and non-respiratory subgroups. We have included a new Figure 1B showing the number of LC patients with available information on different symptoms.

Figure R2: Symptoms from each of LC cohort and clinical tests measurements collected. Empty tiles indicative of information unavailable.

For the primary cohorts (**Cohort 1 and 2**) and a validation cohort (**Cohort 4**), detailed domain-specific data were limited, as these patients primarily presented to the pneumological outpatient clinic at Hannover Medical School (MHH, Hannover, Germany) with a range of symptoms including dyspnoea, fatigue, and headaches. Functional assessments such as the post-COVID-19 Functional Status scale (PCFS) scores, FAS scores, blood gases analysis and lung function tests were performed; however, a systematic organ-level domain classification was not feasible due to variability and retrospective nature of data collection. Summarized clinical information is available in Supplementary table 1,2 and 4.

We have included following text regarding this section in the Results at 144

Patients reporting to the pneumology outpatient clinic at Hannover Medical School (MHH, Hannover, Germany) with symptoms such as headache, dyspnoea and fatigue were recruited based on established diagnostic criteria in accordance with the German S1 guidelines and the Delphi Consensus Criteria of post COVID-19 condition ^{21,22}.

The newly added cohort (**Cohort 3**) provided affected domains more granularly. In this cohort, beyond fatigue and dyspnoea, two patients showed cardiological manifestations, three showed bronchial hyperresponsiveness (BHR), and one had reduced FEV1 and restrictive ventilatory patterns. Based on these features, three patients were classified as having respiratory PASC with BHR (resp-PASC-BHR), and the remaining as respiratory PASC (resp-PASC) (with fatigue and dyspnoea). Detailed clinical information is available in Supplementary table 3

Regarding **Cohort 5** (external, US-based study), this group comprised patients with severe **respiratory** complaints and a pro-fibrotic phenotype in their lungs. Detailed clinical information is available in Supplementary table 5. As circulating monocytes have been implicated in pro-fibrotic phenotype, we had explored whether similar monocyte cellular state found in peripheral blood were also present in BAL fluid, potentially contributing to the respiratory-dominated LC phenotype. We have clarified this rationale in the manuscript at line 411

Previous studies have implicated a role of circulating monocytes in the pathogenesis of PASCs, particularly in the context of pulmonary fibrosis^{42,43}. We hypothesized that the LC-CS monocyte subset, identified in circulation, may contribute to fibrotic remodelling upon recruitment to the lungs. To investigate this, we analysed an independent cohort comprising

paired PBMCs and BAL fluid samples from patients clinically classified as having respiratory or non-respiratory PASC (Resp-PASC and nonResp-PASC, respectively). This classification was based on differences in lung parameters within these two categories (Supplementary Table 5).

d. Mean values for PaO₂ and pulmonary physiology parameters fall in the normal range for all groups. While MMRC scores appear to be elevated – these are subjective and can be elevated in patients without lung disease (such as those with chronic fatigue syndrome). Is there any objective evidence that the long covid cohort has persistent lung disease?

Thank you for this insightful comment. We agree that summarized PaO₂ and standard pulmonary physiology parameters of the primary cohorts fall within the normal range i.e. (< 80%/target) for all the groups, and that mMRC scores, while elevated, are indeed subjective and can be influenced by non-pulmonary factors, such as chronic fatigue.

While we lack radiographic data (e.g. chest CT) or advanced cardiopulmonary testing (e.g. CPET) for these cohorts, we interpret the elevated mMRC dyspnoea scores within the **broader context of LC symptomatology**. We acknowledge that this does not constitute objective evidence of persistent lung disease in the primary cohorts.

To enhance transparency, we provide individual-level pulmonary function data for cohort 1, highlighting variability of PaO₂ and pulmonary physiology parameters within this cohort. We have also revised the manuscript to clarify this limitation and emphasize that mMRC elevations are interpreted as part of the multisystem presentation of LC rather than definitive markers of pulmonary pathology at line 485:

Even though blood O₂ of the primary cohort is within the normal ranges, the negative correlation may suggest minor gas-exchange abnormalities in these patients and may not provide direct evidence of persistent lung disease.

Notably, in the newly added cohort (**Cohort 3**) collected from patients with a mild COVID-19 disease history, three patients were diagnosed with BHR, thereby reflecting abnormalities of the **airway**. Other five LC patients showed fatigue and dyspnoea. We therefore classified these eight patients as either Resp-PASC-BHR (patients with BHR) and Resp-PASC (patients with dyspnoea and fatigue). Summarized clinical information for this cohort is included in Supplementary table 3.

e. How are recovered patients defined? Did these patients never develop long Covid? Or did they have LC and then recover?

In cohort 1, recovered patients represent, reporting patients who developed long COVID and were recovered. All patients reported for a follow up visit > 50 days since their acute COVID-19 disease. Recovered status was assigned if all the measured parameters including FAS, MMRC and Quality of life scores were below the threshold along with clinical parameters of blood gases and lung function were within the normal ranges, as verified by the physician.

We have reworded it in the manuscript at line 173

“cohort 1 included samples from individuals with ongoing SARS-CoV-2 infection, patients with LC symptoms at varying time points (1.7- 10.2 months post-infection), samples from patients recovered from their LC symptoms (Recovered)”

f. Were PaO₂, PFTs, FAS and MMRC measured at the same time PBMCs were drawn?

Yes, all the clinical and patient questionnaire-based information was measured at each visit along with the PBMCs collection. This information is also available in the per sample table attached with this document.

We have added a sentence from line 161 clarifying this point as following:

All clinical parameters, including blood gas measurement, FAS scores, dyspnoea scores, and quality-of-life metrics, were systematically collected and documented at each patient visit (Figure 1B and Supplementary Table 1-5).

3. Patients with long COVID are grouped based on the initial severity of disease and timing from COVID onset. While this could be considered a strength of the paper, it also raises major concerns.

a. Separating the long covid patients into multiple groups (N=8) reduces statistical power and creates problems with multiple comparisons. Accordingly, it is absolutely essential to provide a strong rationale for breaking patients into groups based on: a) initial severity and b) time from infection.

We thank the reviewer for this comment. We fully agree with the reviewer and therefore have expanded text on the rationale for patient stratification in introduction at line 114

Previous studies investigating immune alterations in LC, often overlooked the inter-patient variability, even among patients reporting with LC symptoms e.g., associated with respiratory issues. Notably, while the majority of LC cases emerge in patients who experienced mild-to-moderate acute illness¹³, most molecular studies fail to stratify patients by the severity of their acute infection⁶⁻⁸. This oversight is critical, as individuals who experienced severe acute illness—particularly those requiring ICU care—often underwent a range of interventions- likely resulting in distinct molecular imprints on the circulating immune cells, compared to those from mild-to-moderate LC patients. These imprints may confound efforts to link immune molecular signatures to LC symptoms, underscoring the need for more refined patient stratification.

And we have added the following text in Results at line 191

Heterogeneity in LC molecular imprints may be influenced by differences in acute disease severity and treatment, particularly ICU interventions. To account for this, we stratified LC patients into two groups (LC-mild-to-moderate and LC-severe), based on their acute COVID-19 disease WHO scores (See Methods). To examine the molecular signatures of disease

progression, we further stratified samples by time since acute infection diagnosis (See Methods).

Additionally, the methods section at line 727 now includes information on stratification, grouping of samples and the comparisons made.

Patient and sample stratification and sample category classifications (Cohort 1, 2 and 3):

We first stratified LC patients into two groups, based on their acute COVID-19 disease WHO scores. Where LC samples from patients with acute COVID-19 WHO scores 1-5 were classified as LC-mild-to-moderate patients and LC samples from patients with acute COVID-19 WHO scores 6-9 were classified as LC-severe patients (6-9). Further samples were also classified based on time-points of collection post-acute-infection. These time points are as following: T1- Acute infection, T2: 1.7-2.9 months, T3: 3-5.9 months, T4: 6-8.9 months and T5: 9-11 months.

To ensure our findings were not convoluted with COVID-19 infection imprint on immune cells, and were unique to LC, we performed transcriptome comparisons either to acute infection samples i.e., T1 or to recovered samples i.e., samples from patients that were recovered from LC. All recovered patient samples were considered as 1 category. Consequently, within each group of patients (LC-mild-to-moderate/LC-severe), comparisons were performed as T2 vs T1, T3 vs T1, T4 vs T1, or T2 vs Recovered, T3 vs Recovered etc. In heatmaps, the transcriptome signatures were plotted for all categories, including COVID-19 naïve HCs.

For cohort 3, LC-CS signature was checked in CD14+ monocytes of each LC patient. Patients were further grouped into Resp-PASC-BHR or Resp-PASC as described in Extended Data Fig 6A.

We also included Fig 1C that highlights the number of samples that were compared for each analysis in Cohort 1 and 2.

Figure R3: Figure describing distribution of samples within different groups, where rows of each group represent patients.

With text at line 174:

In summary, cohort 1 included samples from individuals with ongoing SARS-CoV-2 infection, patients with LC symptoms at varying time points (1.7- 10.2 months post-infection), samples from patients recovered from their LC symptoms (Recovered) and COVID-naïve healthy controls (HC) (Fig. 1A and 1C (top)).

And at line 180:

These plasma samples represented 142 patients, 36 patients with longitudinal data and 106 patients at single time points (Figure 1C (bottom)).

b. A major finding of the paper is an abnormal gene signature in long Covid patients that had mild/moderate acute disease. Importantly, this abnormal signature is primarily evident in two time points (Day 101-170, Day 191-251) as illustrated in Fig2b. The signature is not present at the earlier time interval or the final time interval. It is also not present in patients that had severe acute disease. This raises the question as to whether this is a true phenomenon. This is a major point that must be addressed.

We thank the reviewer for raising this important point. Indeed, the broadest signature is **most evident** from. 3-9 months post-acute COVID-19 infection.

However, we provide multiple lines of evidence to support the robustness and biological relevance of this signature:

- **Persistent components across time:** While the broader signature peaks at the middle time points, key genes including *TGFB1* and *CALR*, as shown in Figure 2B and Figure R4 below—exhibiting **consistent upregulation across all time points** in the mild/moderate group. Importantly, these genes were also validated at **protein level** via flow cytometry in an **independent LC cohort**, supporting their persistent activity.

Figure R4. Subset of genes persistently upregulated at all time points in LC-mild-to-moderate samples.

- **Early and late signature presence in individuals:** Elements of the LC-CS signature are detectable as early as **Month 1.7-2.9** and persist in some individuals up to **Month 9-11** suggesting inter-individual variability in the temporal dynamics of the response. The fluctuating presence of the signature likely reflects the heterogeneous and dynamic trajectories of LC, rather than the absence of a true biological phenomenon.

This aligns with known variability in symptom persistence and resolution across patients. For example, as shown in the Figure R5 below:

- Patients P2 and P3 exhibit a **gradual increase** in LC-CS signature expression over time.
- P1 shows a **marked increase at T3**, with minimal upregulation at T2.
- P4 shows the signature at T2 and was subsequently diagnosed as recovered at T4, with no further follow-up.

Figure R5. Monocyte LC signature dynamics in different patients with longitudinal samples.

- **Data recovery limitations at T5:** We note that the cell yields for LC-mild/moderate at T5 (Month 9-11) was the lowest within this group (Figure R6). Although all samples were randomized before sequencing to minimize batch effect, factors such as variable cell viability and sample quality can influence data recovery. Since the LC-CS cellular state represents **a relatively smaller fraction of the CD14+ monocyte population**, the reduced number of cells may also have limited our ability to fully capture this signature in pseudobulk analyses. In summary, technical limitations at T5 may have contributed to the weaker detection of the LC-CS signature, rather than its true biological absence.

Figure R6. Monocyte numbers within each stratified category and time point.

- **Independent cohort replication.** Despite inter-sample variability described above, we observe consistent **neighbourhood enrichment of LC-CS associated cellular-state** in T5 samples compared to both acute and Recovered samples. This signature was also replicated in monocytes of an independent cohort of LC patients (newly added Cohort 3), as detailed in our response to “Comment 3e”. Furthermore, enrichment of the LC-CS signature was observed in PBMCs from a previously published cohort (Cohort 5, from a US study, Figure 6B) providing additional validation across independent datasets.

Together, these findings support the presence of a dynamic but biologically meaningful LC-CS signature in mild/moderate long COVID patients, with variability across time points likely reflecting both individual disease trajectories and technical factors affecting detection sensitivity.

We have updated the manuscript to better contextualize the temporal dynamics of the signature, and we now explicitly discuss these points in the revised manuscript, at line 206 and added the Figures R4-R6 in Extended Data Figs 1A-1C.

Notably, the upregulation of these genes intensified over time (Figure 2B and Extended Data Fig 1A), indicating persistent and progressive alterations in the gene expression profile of CD14⁺ monocytes in LC up to Month 6-8.9, while the signature appears subdued in samples from Month 9-11, possible due to lower number of cells (Extended Data Fig 1B). This late time point also showed upregulation of a subset of genes overlapping acute infection profile. The persistent upregulated genes across all time points were associated with both pro-inflammatory (e.g., *CSF1*, *IRF8*, *RELA*, *NOTCH1*) and anti-inflammatory (e.g., *TGFB1*, *SMADs*, *ENG* and *SERPINE1/PAI-1*) functions (Extended Data Fig 1C).

And at line 483 highlighting patient heterogeneity as evaluated longitudinally.

Longitudinal analysis showed the progression of LC-CS signature, further describing heterogeneity in LC disease dynamics among these patients.

c. Comparisons are made between long-COVID patients and recovered patients are made for the different time points (i.e. Fig 2c, multiple supplements). Are these comparisons made for recovered patients as a whole, or are the comparisons made within timepoints (i.e. Long Covid for T3 vs Recovered for T3). The number of recovered patients at each timepoint are quite small.

We thank the reviewer for this thoughtful question. We agree that the number of recovered patients at individual time points is limited. For this reason, we grouped all recovered samples together as one group for comparison, rather than stratifying by time point.

This decision was based on two key considerations:

1) **Lack of availability of samples.** Once patients are clinically recovered, it becomes logistically difficult to obtain PBMC samples and clinical metadata at multiple longitudinal time points.

2) **Resolution of disease-associated signals:** Recovery likely reflects a return to immunological baseline, with resolution of transcriptomic alterations associated with acute infection or

persistent symptoms. Therefore, we reasoned that the precision timing of sample collection in recovered individuals would have a minimal effect on the comparisons, particularly relative to the dynamic changes observed in LC.

We now clarify this approach with an updated Fig 1C (shown in Figure R3 above), added information in the Methods section at line 738

All recovered patient samples were considered as one category.

and address its implications in the Discussion at line 543

Our study compared molecular alterations in LCs to recovered samples (unstratified) due to limited sample size. Stratified recovered groups may additionally provide deeper insights for a matched comparison in future studies.

d. How were intervals chosen for T2-T5? These seem arbitrary and based on convenience rather than a biologic rationale.

We thank the reviewer for this valuable observation and apologize for not explaining the rationale more clearly in the manuscript.

Due to variability in the timing of patient follow-up visits (e.g. P12 was likely sampled at Day 53 post-acute infection, while P23 was probably sampled at Day 65), we grouped the samples into broader intervals (T2-T5) to enable meaningful comparisons across individuals.,

These intervals were defined as follows:

- T2: up to ~2.9 months post-acute infection
- T3: up to ~5.9 months
- T4: up to ~8.9 months
- T5: up to ~11 months

This grouping was designed to **balance biological relevance** (examine disease progression imprints, as shown in Figure R6 on page 12 of this document) with the **practical constraints** of patient availability, while still capturing different phases of post-acute immune dynamics. We have now clarified this approach in the revised manuscript in the methods at line 731.

Further samples were also classified based on time-points of collection post-acute-infection.

These time points are as following: T1- Acute infection, T2: 1.5-2.9 months, T3: 3-5.9 months, T4: 6-8.9 months and T5: 9-12 months.

e. Early and late time intervals (T2, T5) are missing for comparisons of moderate LC vs recovered in Fig 2c and Fig 3E. This gives the appearance that data are “cherry picked” since these early and late time points have less impressive gene signatures (see comment 2b above).

We thank the reviewer for this important observation and apologize for not making the rationale behind the figure sections clearer in the manuscript.

Our primary aim in Fig. 2C was to investigate the dynamics of immune-related pathways in LC samples in comparison to acute COVID-19 samples. Since immune pathways are typically downregulated following acute infection, we sought to identify the key immune pathways

that remain elevated —or are reactivated—in LC. To expand this analysis, we also include comparisons to recovered individuals to determine whether any immune pathways showed continued or selective upregulation in LC.

The most prominent immune pathway dynamics were observed at T3 and T4, which is why these are the focal points in the main figures. As this analysis was done using both HALLMARK and REACTOME gene sets, it generated a comprehensive list of significant pathway enrichment results. To improve clarity and focus in the main text, we limited the display to immune-related pathways that showed statistically significant changes, and we plotted only these comparisons where significance was reached. Comparisons that did not reach significance for the pathways shown —such as at T2 and T5—were therefore not displayed in Figure 2C.

Specifically regarding T5, as noted in our response to “Comment 3b”, cell number per sample were substantially lower at this late time point, reducing statistical power and limiting the detection of significant pathway changes. As such, from the listed pathways in Figure 2C, no pathways reached statistical significance for LC vs. recovered at T5, and this comparison was excluded from the main figure.

To ensure transparency and provide a complete overview, we have included all pathway enrichment results—including those for T2 and T5 comparisons, across all major subsets and stratifications—in Extended data table 2. We have now clarified this rationale in Methods section of the revised manuscript at line 758.

Pseudobulk of each donor per time point was calculated, followed by similar comparisons as described in previous section using DESeq2 were done. Gene Set Enrichment Analysis (GSEA) using HALLMARK and REACTOME pathways as background was calculated using fgsea R package. Further, immune related pathways showing statistically significance for any comparisons were plotted. The whole list of statistically significant pathways resulting from all comparisons is added in Extended data table 2.

Similar to the approach in Figure 2C, the initial comparisons in Figure 3E were performed against the acute time point (T1) to track the evolution of cellular states over time. In response to the reviewer’s suggestion, we have now included the results for T2 and T5 comparisons as well. These time points show broadly similar patterns as shown in comparison to T1 in terms of the MC4 cellular state. At T2, fewer cellular neighbourhoods reached statistical significance— likely due to not fully developed cellular state in most donors at this stage (as shown above in figure R6 on page 12 of this document). However, in the T5 vs Recovered comparison, we observed neighbourhoods reaching statistical significance in the enrichment of the MC4 cellular state, supporting the persistence of this state at later time points. (See figures below)

Figure R7.1: Analysis for comparison between Mild-to-moderate T2 vs Recovered. Red indicates significant upregulated neighbourhoods, blue indicates significantly downregulated neighbourhoods, grey represent not significant. MC1-MC4 represent monocyte clusters. MC4 represents the LC-CS.

Figure R7.2: Analysis for comparison between Mild-to-moderate T5 vs Recovered. Red indicates significant upregulated neighbourhoods, blue indicates significantly downregulated neighbourhoods, grey represent not significant. MC1-MC4 represent monocyte clusters. MC4 represents the LC-CS.

We have included these two new figures in the main figure 3E with the text in the Results section at line 302.

Given the well-defined and significant role of monocytes in acute SARS-CoV-2 infection^{31,32}, we next performed differential neighbourhood analysis and compared LC samples to acute and recovered samples at all time points (Fig. 3E and Extended Data Fig. 5A-5B).

Additionally, the new cohort (cohort 3) comprising of LC patients with a mild COVID-19 disease history showed enrichment of LC-CS signature (Clust1). These samples were collected from > 8 months (T4) to up to 42 months (beyond our T5 time point) since acute infection. Importantly, the cellular state was found in each of the LC patients. Please see below:

Figure R8: (A) CD14+ Monocytes from LC patients of Cohort 3. **(B)** LC-CS signature AUC scores in the 3 subclusters, showing highest activity in Clust1. **(C)** Distribution of Clust1 in each of the LC patient's monocytes.

We have updated the text at line 338 in manuscript with new figures:

To validate the LC-CS signature, we analysed scRNA-seq data from an independent cohort (Cohort 3) of eight LC patients who had experienced mild acute COVID-19 disease but presented with severe LC symptoms 8-42 months post-infection (Supplementary Table 3, see methods). All patients reported fatigue and dyspnoea; three also showed bronchial hyperresponsiveness³⁵ and were classified as Respiratory-PASC with BHR (Resp-PASC-BHR). CD14⁺ monocyte analysis revealed three main clusters, with Clust1 showing significantly elevated LC-CS signature AUC scores based on the top 141 marker genes of LC-CS identified above (Figure 4A, 4B).

f. Levels of plasma cytokines (Fig2D) do not appear to exhibit the same kinetic as suggested by pseudobulk analysis. Since plasma is from a larger cohort, this reinforces concerns raised in point 2b above. Although it is not implicitly stated, it is implied that peripheral blood leukocytes are the source of the cytokines. Accordingly, if analysis needs to be broken down by time it would be useful to show gene expression for TNF, CCL2, CXCL11 from PBMCs used to generate Fig 2b, c.

We thank the reviewer for this insightful comment. We agree that plasma cytokines levels do not fully mirror the kinetics observed in the pseudobulk transcriptomic analysis of monocytes. This discrepancy likely reflects the **systemic nature of plasma cytokines**, which represent a cumulative signal from multiple cellular sources beyond peripheral blood leukocytes. As such their levels may not directly correlate with transcriptional activity within PBMCs alone.

In response to the reviewer's suggestion, we have now **examined the gene expression of TNF, CCL2 and CXCL11 across all immune cell subsets** at all available time points. As shown in the new figure below, TNF expression is consistently upregulated in LC-mild/moderate patients across most immune subsets.

In contrast, CCL2 and CXCL11 expression is not markedly elevated in PBMCs and was not detected in all immune subsets, supporting the possibility that these cytokines are predominantly produced by non-hematopoietic sources such as endothelial, epithelial, stromal cells.

Figure R9: Gene expression of TNF, CCL2, CXCL11 in major immune cell subsets from PBMC.

We have included these additional analyses and clarified this interpretation in the revised Results Sections line 247 and have added this figure in the Extended Data Fig 2D.

TNF α gene expression was also persistently upregulated within LC mild-to-moderate samples in most immune subsets in most of time points (Extended Data Fig 2D).

4. It seems inappropriate to include recovered patients in statistical analysis for figure 2f. By definition, these patients should have low FAS scores. If gene expression has “normalized” in these patients as part of recovery they will skew the data.

We thank the reviewer for this important observation. We would like to clarify that gene expression normalization was performed at the **genome-wide level using the well-established normalization methods such as those implemented in Seurat** ensuring that statistical comparisons between conditions were not biased toward any specific group, including recovered patients. We agree, that if the FAS scores are really correlated with the pathway activity (and not with the patient status), then the recovered samples with low FAS scores should also have lower expression of genes of these pathways.

Therefore, our primary aim in Figure 2f was to assess the **association between pathway activity scores** (calculated using AUC scores of ~ 40-80 genes per pathway from pseudobulk expression per cell subset) and **FAS scores**, independent of disease status. Including recovered individuals, who by definition have low FAS scores and correspondingly lower pathway activity, strengthens this analysis by illustrating **a gradient of association across the clinical spectrum**, rather than comparing only symptomatic patients.

Indeed, the fact that recovered individuals cluster at the lower end of both fatigue and pathway activity supports the biological relevance of the relationship between these pathways and fatigue severity. Importantly, patients with higher FAS scores exhibit the highest pathway activity, consistent with this trend.

That said, we acknowledge the reviewer's concern and would be happy to provide an additional sensitivity analysis excluding recovered patients. As shown below, the direction of the correlation stays the same without the recovered samples included for all comparisons.

Figure R10: Correlations of pathways with FAS scores without recovered samples

We have added text to clarify the point at line 271

These correlations performed without patient stratification, suggest a gradient of association of these pathways across the clinical spectrum, rather than patients alone.

5. Pseudotime trajectory analysis adds little to the paper. I recommend removing it. The method “forces” a trajectory whether one is present or not. Moreover, it is not true reflection of lineage tracing since monocyte survival in the circulation is limited to a few days. In comparison, timecourses used for the analysis are on the order of many months.

We thank the reviewer for this thoughtful comment. We fully agree that **pseudotime does not reflect true lineage tracing**, particularly given the short lifespan of circulating monocytes and the extended clinical time course in our study.

Our intention, however, was not to infer cellular lineage or fate, but rather to **explore the continuum of transcriptional programs** within monocytes at a single-cell level. This approach allows us to capture the **diversity of immune states** across patients and time points in an **unsupervised, non-stratified manner**, thereby complementing our patient/timepoint-based analyses.

Importantly, this analysis provided **orthogonal validation of transcriptional modules** identified in stratified datasets, suggesting that these programs emerge consistently, independent of sample metadata. To address the reviewer's concern, we have now moved this analysis to **Extended Data Fig 5** and **substantially condensed the corresponding text**, highlighting only the core rationale and key findings at line 319 in the revised manuscript .

We additionally evaluated the transcriptional program of MC4-like cells among all CD14⁺ monocytes through trajectory analysis (See Methods) in a stratification-independent manner (Extended Data Fig 5D). The resulting three lineages showed the segregation of single cells solely based on transcriptome, with lineage 3 overlapping MC4 immune program, without prior sample stratification (i.e., acute, recovered and LC samples category) and allowed to recapitulate the pathway dynamics from stratified analysis (Extended Data Fig 5E).

We hope this clarification addresses the concern and demonstrates that this analysis was used cautiously and interpretively, without overstating biological trajectory or cellular ontogeny

6. Experiments that reanalyze published data from LC patients are both confusing and confounded by lack of healthy controls as a reference to show whether cells are “pro-inflammatory” or “pro-fibrotic.” The strength of the published data arises from patients with pulmonary symptoms versus non-pulmonary symptoms. If these data are to be leveraged, then the author's primary dataset should also incorporate respiratory versus non-pulmonary phenotyping. The ideal analysis would be to derive a LC-CS signature from the author's dataset and to demonstrate whether this gene signature maps on to PBMCs from the published dataset for respiratory vs non-respiratory phenotypes. This would help to confirm the author's results and provide a firm link to the first 5 figures of the paper. In its current form, the link is tenuous.

We thank the reviewer for this important and constructive comment, and we apologize for any confusion caused. We agree that the absence of pulmonary vs non-pulmonary symptom stratification in our primary cohort (Cohort 1) limits the interpretation of LC-CS implications in respiratory PASC. We addressed this point using two approaches; using a new cohort and examining published cohort based on reviewer's suggestion. Details of both are below.

- We incorporated a **new independent cohort (cohort 3)** from a pulmonary rehabilitation clinic in Germany. All patients in this cohort reported LC symptoms such as fatigue and dyspnoea, with three patients additionally diagnosed with bronchial hyperresponsiveness (BHR). All had experienced mild acute COVID-19 disease. Based on these clinical features, we classified the patients into respiratory-PASC-BHR and respiratory-PASC subgroups. Notably, LC-CS signature cells was found in all LC patients (Figure R8 on page 16 in this document), and its expression was **significantly higher** in respiratory PASC-BHR patients, compared to non-respiratory PASC patients. Please see below:

Figure R11: LC-CS signature AUC score in LC patient’s Clust1 cells, with higher AUC scores in respiratory PASC patients.

We have incorporated this analysis at line 345 of manuscript:

Notably, Resp-PASC-BHR patients showed significantly higher LC-CS enrichment compared to Resp-PASC individuals (Figure 4C). These results independently validate the presence of the LC-CS signature in mild-to-moderate LC patients and suggest a potential association with progression to Respiratory-PASC.

- We also went back and examined the published dataset again (cohort 5) and found additional immune cells within the PBMCs from 2 HCs. Therefore, we included cells from HCs also in the analysis and show results within the PBMCs for a better comparison. Following the reviewer’s helpful suggestion, we **re-evaluated the enrichment of the LC-CS signature within PBMC of 2 HCs and PASC patients**, where monocyte cluster 5 showed the highest enrichment (Figure 6A-6B, also shown below).

We then stratified the cells in the published PBMC dataset by PFT based **phenotype (respiratory-PASC vs nonrespiratory-PASC vs healthy controls)**, as described in the publication. This allowed us to test whether our LC-CS signature derived from our dataset could distinguish relevant patient subtypes in the external cohort, as suggested by the reviewer.

Indeed, we observed that **cells from respiratory PASC patients exhibited significantly higher enrichment of the LC-CS/MC4 signature**, particularly within monocyte clusters 5. In contrast, **non-respiratory PASC patients and healthy controls showed no such enrichment**. These findings support the relevance of the LC-CS signature to pulmonary manifestations of Long COVID and **establish a clearer link between our primary data and the published study**, thereby strengthening the translational relevance of our findings.

Figure R12: LC-CS signature enrichment within cluster 5 monocytes, cells stratified based on HCs and PASC categories.

We have now incorporated this analysis in line number 422 and added the figure panel in 6C.

Moreover, cluster 5 cells of Resp-PASC patients showed significantly higher enrichment of LC-CS signature in comparison to cells of both nonResp-PASC and HCs (Fig. 6C).

And in discussion at line 487:

However, the LC-CS signature showed highest expression in patients with clinically severe respiratory LC/PASC symptoms in two independent cohorts, irrespective of their acute COVID-19 disease history. Within the BAL samples, LC-CS-like macrophages showed a pro-fibrotic phenotype and were in higher proportions in patients with Resp-PASC.

And line 526:

It is intriguing, that while the LC-CS signature was predominantly identified in the circulating monocytes of LC-mild-to-moderate patients, it is also present in the myeloid cell populations in the BAL of LC with severe respiratory PASC. In our dataset, we did not see LC-CS in circulation of LC-severe category, except for one patient. It is therefore tempting to speculate that monocytes with the LC-CS signature remain in the circulation and are not actively recruited in the lungs of LC-mild/moderate patients, leading to fatigue and subtle impairment in pulmonary function. While they migrate in the lungs of LC-severe patients, where they contribute to the severity of the lung pathophysiology, upon receiving signals from the tissue micro-environment.

We hope that these two independent analyses, from different cohorts strengthens the link between the LC-CS signature identified in our primary dataset and its link to pro-inflammatory/pro-fibrotic profiles in patients with respiratory difficulties, in alignment with the findings from the published dataset.

Minor

Line 116. Text is missing from the sentence that ends with “characterized by elevated.”

We apologize for this issue. We have re-written this section at line 130 as:

Our findings revealed a distinct long COVID cellular state (LC-CS) within the circulating CD14⁺ monocytes, particularly enriched in patients who experienced mild-to-moderate acute infection.

Line 204 states “ CD14⁺ monocytes in these patients showed elevated ‘PD1 signalling’, suggestive of a low phagocytotic state...”. This is an over interpretation. PD1 is involved in a number of functions in monocytes (including cell death). The role of PD1 in phagocytosis has primarily been suggested in tumor associated macrophages.

We agree with the reviewer’s suggestion and have removed the over-interpretation from the manuscript to ensure a more accurate representation of the data.

The Y axis on Figure 1D is confusing. Have data values been log-transformed? Would it be clearer to display the data using a log scale?

The data were indeed log₂-transformed for each cytokine. We have now added this information to the figure 2 legend

Plot of log₂ (pg/ml) of cytokines showing significant differences compared to healthy controls.

Flow cytometry provides a nice companion to monocyte transcriptional phenotyping. In the supplemental figures gating strategies should be shown. In addition, it will be important to show data for some of the trickier markers (i.e. TGFb1) along with their isotypes. Lastly, it will be important to confirm that isotypes were used as fluorescence minus one (FMO) fashion in which all other markers received true Abs.

We thank the reviewer for these helpful suggestions. In response, we have added histogram plots in **Extended Data Fig 6** (please see below) in the revised manuscript to enhance transparency and reproducibility.

For monocyte gating, we used CD3⁺ cells as a control population. Since CD3⁺ cells do not express monocyte-specific markers, their inclusion allowed us to define negative staining boundaries and confidently gate monocyte subsets in the presence of the same antibody panel.

Regarding the use of isotype and FMO controls, we acknowledge their distinct roles. FMO controls are typically used during panel optimization to evaluate fluorochrome spillover and background fluorescence, while isotype controls are important for determining nonspecific antibody binding, particularly in cells with high Fc receptor expression such as monocytes.

In our study, we focused on intracellular markers including TGFb and IRF8, which are known to be challenging due to potential background staining.

To minimize non-specific binding and maximize marker resolution:

- We included an Fc receptor blocking step prior to antibody staining.

- We designed two independent flow cytometry panels using our spectral analyser equipped with 5 lasers to minimize the fluorochrome overlap
- For key markers identified in transcriptomic analyses (e.g. TGFb1), we incorporated isotype controls to guide accurate gating and confirm the specificity of antibody binding.

Figure R13: Histograms for the flow cytometry analysis of CALR, CD99, TGFb1, HLA-DQ, and CD120b surface expression enriched for all CD14⁺ monocytes from healthy controls (HC) compared to long-COVID patients (LC). HC, LC antibody and isotype stained are shown in blue, red and black, respectively. Shown is a representative donor of biological replicates with similar results.

These plots are added in Extended Data Fig 6 with amendment to main text at line 366

Compared to HCs, LC patients showed a significantly increased expression of HLA-DQ, CD120b, CALR, CD99 and TGF-β1 markers on CD14 monocytes (Fig. 4H and Extended Data Fig.6B).

Reviewer #2 (Remarks to the Author):

In this manuscript, Kumar et al. present the results of a multiomics profiling study performed on blood samples from a prospective cohort of patients with acute COVID-19 or sequelae (“Long-COVID” or PASC), fully recovered patients, or healthy controls, in an effort to uncover features and potential mechanisms of immunopathology in PASC. They performed single cell multiomics profiling using single nuclei RNA sequencing, scATAC-Seq, multi color flow cytometry on PBMC and plasma cytokine profiling. They additionally analyzed scRNA-Seq data from an external cohort of patients with PASC. Here, they integrated scRNA-Seq data from PBMC and bronchoalveolar lavage (BAL) cells.

The main finding is a distinct transcriptional profile in PBMC, particularly in CD14+ monocytes, in patients with PASC after mild to moderate disease during the acute stage of the infection, compared to acute disease or recovered patients. PBMC from patients with PASC after mild to moderate acute COVID-19 displayed an enrichment of genes associated with TGFbeta- and WNT-Catenin signaling. The authors describe a cellular substate in CD14 monocytes that was positively correlated with participants’ fatigue score and negatively correlated with O2 saturation. Moreover, they describe a similarity of this monocyte state (MC4) with BAL macrophages (from PASC patients) that express profibrotic signatures, indicating that this monocyte state may be mechanistically linked to fibrotic responses in patients with respiratory symptoms and PASC.

Overall, this is a complex study that offers interesting insights into transcriptional states in PASC patients. Specifically, distinguishing between PASC patients with acute-to-moderate initial disease and those with initially severe disease is important, since the clinical phenotype and the underlying pathophysiology are likely distinct. However, the study contains a number of inconsistencies and shortcomings, which compromise the overall interpretation and the potential impact of the study. Details are listed below.

We thank the reviewer for their positive assessment of our manuscript.

Specific comments:

1. The authors measured samples from a complex cohort, with both longitudinal and cross-sectional sampling, and participants with distinct outcomes, and controls. Figure 1 attempts to depict the samples from each time point, but it fails to provide a sufficient overview of the composition of the cohort. Suppl. Table 1 adds some information, but the overall description of the cohort is insufficient, and additional information on which samples were taken from the same individuals etc. should be added.

We thank the reviewer for this important comment and apologize for the lack of clarity in the original presentation of the cohort. To address this, we have revised the text and supplemental materials to provide a more comprehensive overview.

Our study design involved both **longitudinal** and **cross-sectional** sampling of participants across multiple clinical categories, including **Long COVID (LC) patients, recovered individuals, and acute COVID-19 cases.**

To improve transparency, we have provided information on the two discovery cohorts in Supplementary table 1 and 2 and now included additional figures describing each cohort’s LC

sampling distribution, range of COVID-19 disease severity as available below in **Figure R14** and in **Figure R3** on page 9 of this document, to better reflect the sampling schema and number of samples used for each time point in each stratified comparisons for transcriptomics data analysis. Specifically, we performed pairwise comparisons such as T2 vs T1, T3 vs T1, and comparisons of LC timepoints (T2 vs Recov, T3 vs Recov and so on) vs the Recovered group (comprising 8 donor samples). We also included information of LC symptoms and clinical tests performed and their numbers within each cohort (Figure R2 on page 6 of this document).

Figure R14: Distribution of LC samples from each cohort. Barplot colours represent acute COVID-19 disease severity. Dotted red line represents start of LC sample collection.

Information on longitudinal samples is added in the Results section at line 167 and figures are now included in main Figure 1A and 1C.

To comprehensively characterize the cellular, molecular and regulatory landscape of LC, we performed single-cell multiome analysis (snRNA-seq and snATAC-seq) from 78 peripheral blood mononuclear cell (PBMC) samples (Cohort 1) incorporating both longitudinal and cross-sectional data from 45 patients (Acute Infection and LC) and 6 healthy controls. (Fig. 1A and Supplementary table 1). These PBMC samples represented 18 patients with longitudinal samples of at least two time points from the same patient and 27 samples at single time points. In summary, cohort 1 included samples from individuals with ongoing SARS-CoV-2 infection, patients with LC symptoms at varying time points (1.7- 10.2 months post-infection), samples from patients recovered from their LC symptoms (Recovered) and COVID-naïve

healthy controls (HC) (Fig. 1A and 1C (top)). All patients (except acute infection time point) tested PCR-negative at the time of collections.

2. Some notes on the cohort. Some groups differ in size (e.g. n=28 PASC_mild/moderate vs n=8 recovered) and in baseline characteristics (e.g. avg. age 39 yrs in recovered vs. 52 yrs in recovered, and male:female ratio of 2 in recovered and 0.58 in PASC_mild/moderate), making it difficult to clearly attribute transcriptomic differences to the clinical outcome, as opposed to baseline differences. It is also unclear, which participants contributed several samples and how the cohorts were composed at the different time points with regards to demographic baseline parameters. Overall, I feel that it would be more powerful to compare matched cohorts.

We thank the reviewer for this thoughtful and important comment. We acknowledge the inherent limitations in cohort composition, particularly regarding group sizes and baseline demographic differences such as age and sex ratios. However, to address these potential confounding factors, we structured our analyses to mitigate their influence wherever possible.

Specifically, **Long COVID (LC) samples were not only stratified by clinical severity but also across distinct timepoints (T2–T5)**. This approach allowed for more **balanced group sizes**, particularly at timepoints T3, T4, and T5, where each group within LC-mild/moderate includes approximately **10 samples** as also reflected in **Figure R3** (page 9 of this document), compared to **8 samples in the Recovered group**.

While it was not possible to fully control the **sex ratio** in the Recovered group due to limited sample availability, we did perform additional comparisons to acute-phase samples (T1) for each LC timepoint, where the **male-to-female ratio is more balanced** (4 males and 3 females). Moreover, the **LC-associated cellular state** was consistently enriched in **both male and female** patients with the LC-mild/moderate group (see the figure below), suggesting that sex is unlikely to be the primary driver of the observed transcriptional differences.

Figure R15: Proportions of MC4/LC-CS cells within each donor from LC-mild-to-moderate, stratified based on sex. Each dot corresponds to a sample, the size of dot represents percentage of MC4/LC-CS cells within CD14+ monocytes of the sample.

To improve transparency, we have now:

- **Updated the Fig 1C (Figure R3)** to clearly indicate the distribution of samples from unique versus repeated donors at each time point.
- **Attach a table with** donor-specific demographic data (age, sex timepoint and measured clinical parameters) with this document.
- Added a clarifying description of the **Methods section** line 733 and referenced the new Extended Data materials accordingly as also described on pages 9-10 of this document.

And we have added Figure R15 in Extended Data Fig 5A with mention in text at line 305

“In both comparisons, LC samples particularly from mild-to-moderate acute infection showed a significantly increased abundance of neighbourhoods in the MC4 cluster, **in both males and females** (Fig. 3E and **Extended Data Fig 5A**).”

While we agree that matched cohort analyses offer stronger control over confounders, such matching is often impractical in real-world longitudinal studies involving post-acute infection syndromes. Nonetheless, we believe our stratified approach and additional validation steps help ensure the robustness of our findings.

3. In lines 161/162 the authors state: “In all cell subsets, genes upregulated during acute infection were markedly downregulated in LC samples from individuals with mild to moderate acute infection.” While there is a general pattern, there is clearly also a cluster of genes in the top of the ‘Day 292-311’ column (Figure 2B) that appear to be upregulated with some overlap to the acute infection.

We thank the reviewer for this valuable observation. We agree that while the overall pattern shows downregulation of acute-phase genes in LC samples with mild/moderate initial disease, there is indeed a subset of genes — particularly in the Month 9-11 samples — that show overlapping upregulation with those observed during acute infection.

This partial overlap may reflect heterogeneity within the LC cohort at later timepoints. Notably, the majority of Month 9-11 samples are from **cross-sectional collections**, and it is possible that some individual exhibit **persistent or reactivated inflammatory programs** resembling those seen during acute infection.

We have revised the manuscript accordingly at line 200 and at line 209 to clarify this point and acknowledge this nuance in the data.

“In all cell subsets, genes upregulated during acute infection were markedly downregulated in LC samples from individuals with mild-to-moderate acute infection, **until Month 6-8.9**.

This late time point also showed upregulation of a subset of genes overlapping acute infection profile.

4. In lines 166-171 the authors state: “Specifically, differential gene expression analysis in, CD14+ monocytes revealed distinct upregulation of 1737 genes in LC samples, particularly in individuals with mild-to-moderate acute SARS-CoV2 infection, compared to samples from acute infection, recovered patients and HCs. Notably, the upregulation of these genes intensified over time, indicating persistent and progressive alterations in the gene expression profile of CD14+ monocytes in LC. The persistent upregulated genes are associated with both pro-inflammatory (e.g., *CSF1*, *IRF8*, *RELA*, *NOTCH1*) and anti-inflammatory (e.g., *TGFB1*, *SMADs*, *ENG* and *SERPINE1/PAI-1*) functions.” Contrary to what is stated here, looking at the data in Figure 2B, it seems that differential gene expression is rather transient and the transcriptional changes observed at Day 101-170 and Day 191-251 are largely reversed by Day 292.

We thank the reviewer for this important observation. A similar concern was raised by Reviewer 1 (Comment 3b), and we refer to our detailed response on page 10-13.

In brief, while the LC-CS transcriptional signature is most prominent at 3-6 months and 6-9 months, several lines of evidence support its biological relevance and persistence:

- Key genes such as *TGFB1* and *CALR* show consistent upregulation across all time points and were validated at the protein level in an independent LC cohort.
- The signature is also detectable in some individuals as early as 1.7-3 months and as late as 9-11 months, suggesting temporal heterogeneity.
- We independently replicated the LC-CS monocyte signature in a new cohort (Cohort 3) and observed enrichment of these cell states in T5 samples in primary cohort (Cohort 1).
- The observed fluctuations likely reflect variability in long COVID disease trajectories. Additionally, low cell yields at T5 may have limited detection of rare LC-CS cells in pseudobulk analyses.

Together, these results support the LC-CS signature as a dynamic yet biologically meaningful feature of mild/moderate long COVID, with its variability shaped by both patient-specific progression and technical constraints.

We have clarified these points in the revised manuscript, at line 206 and added the revision figures in Extended Data Figs 1A-1C and Figure 3E

Notably, the upregulation of these genes intensified over time (Fig 2B and Extended Data Fig 1A), indicating persistent and progressive alterations in the gene expression profile of CD14⁺ monocytes in LC up to Month 6-8.9, while the signature appears subdued in samples from Month 9-11, possible due to lower number of cells (Extended Data Fig 1B). This late time point also showed upregulation of a subset of genes overlapping acute infection profile. The persistent upregulated genes across all time points were associated with both pro-inflammatory (e.g., *CSF1*, *IRF8*, *RELA*, *NOTCH1*) and anti-inflammatory (e.g., *TGFB1*, *SMADs*, *ENG* and *SERPINE1/PAI-1*) functions (Extended Data Fig 1C).

We have also included the T5 vs Recovered neighbourhood enrichment in the main figure 3E with the text in the Results section at line 302.

Given the well-defined and significant role of monocytes in acute SARS-CoV-2 infection^{31,32}, we next performed differential neighbourhood analysis and compared LC samples to acute and recovered samples at all time points (Fig. 3E and Extended Data Fig. 5A-5B).

5. Along these lines, which time point was analyzed in Figure 3F?

We thank the reviewer for this question and apologize for the lack of clarity. Below, we clarify the analyses presented in Figure 2F and 3F:

Figure 2F:

This figure includes **all samples** from **all time points** for both LC and recovered individuals. For each sample and cell type, we computed a pseudobulk gene expression profile, and subsequently calculated AUC scores for the selected pathways, each composed of approximately 40-80 genes. We then assessed the Spearman correlation between these pathway scores and the corresponding FAS score across all samples. This analysis was designed to evaluate the association between transcriptional pathway activity and fatigue severity, independent of disease stage or time point.

Figure 3F:

Here, we calculated **AUC scores at the single-cell level**, regardless of time point, to assess pathway activity across the cell cluster defined in Figure 3C. This approach enabled us to examine transcriptional heterogeneity within monocyte subpopulations, independently of longitudinal sampling.

We have updated the manuscript Methods section to include these clarifications at line 762.

Pathway correlations with clinical parameters.

For each sample and cell type (CD14 monocytes, CD8 T and NK cells), we computed pseudobulk gene expression profiles. Subsequently, AUC scores for the selected upregulated immune pathways as described in Figure 2C were calculated in each sample. The AUC scores of these pathways were then correlated with clinical parameters using spearman correlations. Only significant correlations were plotted.

And at line 271 in Results section.

These correlations performed without patient stratification, suggest a gradient of association of these pathways across the clinical spectrum, rather than patients alone.

6. Also, the authors state that TGF-beta signaling and WNT-beta catenin signaling pathways in CD14+ monocytes were positively correlated with FAS scores (lines 243-244). They should acknowledge that the correlations were rather weak (R=0.28).

We thank the reviewer for this important observation. We agree that the correlation values reported in Figure 2F (e.g. $R=0.28$) are modest in strength, though statistically significant. Notably, these pathways (e.g., TGF- β and WNT/ β -catenin signalling) were among the few that showed any meaningful correlation with FAS scores, while other pathways exhibited weaker or non-significant correlations (data not shown).

It is important to note that the correlation in Figure 2F were derived from pseudobulk gene expression profiles of entire cell subsets, such as CD14⁺ monocytes. However, as we demonstrated in later figures, the increased pathways activity is largely confined to a specific monocyte subpopulation (MC4 cluster/LC-CS). Since pseudobulk analysis averages the signals across all monocyte states (MC1-MC4), this likely attenuates the correlation signal observed at the bulk level. We used this approach as a first-pass, cell type- level analysis to explore potential links between pathway activity and FAS score.

We have now acknowledged the limited strength of the correlation and clarified this point in the revised text at line 262

Interestingly, FAS scores showed **modest positive correlations** with both ‘TGF- β signalling’ and ‘WNT- β catenin signalling’ pathways in CD14⁺ monocytes (Spearman $R = 0.28, 0.31$; p value = $3.6 \times 10^{-2}, 1.8 \times 10^{-2}$ respectively) (Fig. 2F).

7. Figure 3C shows GSEA for CD14 monocytes and NK cells for 3 different time points in the comparison with acute disease (“Vs Acute”) and 2 time points in the comparison with Recovered (“Vs Recov), missing “Day 53-78”. Yet, for CD8 T cells there are 3 time points in both comparisons. What is the reason? Also, it would be important to also compare this with Healthy controls.

We thank the reviewer for this helpful comment and apologize for the lack of clarity in the manuscript.

In Figure 2C, our initial aim was to examine the dynamics of immune-related pathways in LC samples in comparison to acute infection. Under the hypothesis that most immune pathways in LC samples would diminish post-acute infection. This approach allowed us to highlight pathways persistently or selectively upregulated in LC.

We then included comparisons with recovered individuals to determine whether these same immune pathways, or additional ones, remained elevated specifically in LC samples.

Because the transcriptomic signature was most prominent at the T3 and T4 time points, these comparisons yielded the most robust enrichment of immune-related pathway. In contrast, T5 samples—particularly in CD14⁺ monocytes and NK cells—contained fewer cells per sample (as previously discussed in response to Comment 4), which reduced statistical power and limited pathway detection. For CD8⁺ T cells, T2 was included in both comparisons (vs. Acute and vs Recovered), as sufficient cell numbers, resulting in significant enrichments were retained across time points. We calculated pathway enrichments against the entire HALLMARK and REACTOME pathway datasets, which yielded an extensive list of significant pathways. For clarity and focus in the main figure, we presented only a curated subset of significantly enriched immune-related pathways from the whole list. Therefore, comparisons for time points without statistically significant enrichment for these immune pathways were not plotted.

However, we provide a comprehensive list of all significant pathway enrichments (including those from T2 and T5 vs Recovered, all major subsets, and additional comparisons stratified by patient group, including comparisons against healthy controls— in the **Extended data table 2**. This resource allows the reader to fully explore the breadth of pathway dynamics observed across the dataset.

We have clarified this rationale and added a reference to Extended data table 2 in the Methods section at line 758.

Pathway analysis per cell type

Pseudobulk of each donor per time point was calculated, followed by similar comparisons as described in previous section using DESeq2 were done. Gene Set Enrichment Analysis (GSEA) using HALLMARK and REACTOME pathways as background was calculated using fgsea R package. Further, immune related pathways showing statistical significance for any comparisons were plotted. The whole list of statistically significant pathways resulting from all comparisons is added in Extended Data table 2.

8. Figure 3 and some of the interpretations are problematic and inconsistent. The authors describe four monocyte clusters MC1-MC4 within the monocyte space and found MC4 to display “higher activity in “TGF-beta signalling” and “WNT-beta catenin signalling”. They state that LC (PASC) samples from severe initial disease did not enrich for MC4, which is also evident in Suppl. Figure 4. In contrast, in Figure 3G, where the authors correlated MC cluster abundance per donor with FAS scores, it is clear that the majority (if not all!) donors with high percentages of MC4 of the CD14 monocytes were in fact “LC-severe” (=dark red dots). This does not square with some of the previous statements that showed a specific transcriptional program in LC-mild/moderate compared to LC-severe. It is also not in line with the results shown in Figure 4C.

We thank the reviewer for carefully pointing out this important inconsistency, and we sincerely apologize for the confusion it caused. Upon thorough review of our data and code, we discovered that the group colours in Figure 3G were mistakenly assigned due to a bug in the plotting script—specifically, the group labels were randomly applied rather than accurately mapped to their respective patient categories for this figure alone, as defined in the legend.

We have now corrected the issue and the updated version of figure 3G accurately displays the correct group colours (see the figure below). As expected, donors with the highest proportion of the MC4 cellular state are predominantly from the LC-mild/moderate group (now correctly shown in bright red). This correction restores consistency with our earlier findings, including those shown in updated Extended Data Fig 5D, and supports our interpretation that the MC4 transcriptional program is “predominantly” observed in the LC-mild/moderate subgroup.

We would also like to clarify that this was solely a visualization error. We have also uploaded the corrected code on github. The correlation coefficients shown in the figure were calculated using all samples, regardless of group assignment, and are therefore unaffected by the coloring issue. As such, the conclusions drawn from these analyses remain valid.

We appreciate the reviewer’s close attention to detail, which enabled us to identify and correct this issue.

Figure R16: Updated Main Figure 3G with correct color labels

9. Similar to the previous comment, Figure 4H (left most panel, “TGFB1”) suggests that TGFB1 was indeed increased in the ‘LC-severe’ samples, which contradicts prior statements that this was a specific feature of ‘LC-mild/moderate’.

We thank the reviewer for this insightful comment. Figure 4H represents protein-level expression of TGFB1 measured in an independent validation cohort of both mild-to-moderate and severe patients, compared to Healthy Controls.

We agree that TGFB1 alone, at protein level, does not completely distinguish between LC-severe and LC-mild/moderate groups. Rather, it reflects one component of a broader transcriptional and functional cellular program. As noted in our main analysis, the LC-associated cellular state (LC-CS), characterized by coordinated activation of multiple genes and pathways—including but not limited to TGFB1—shows differential enrichment based on initial disease severity.

We have now clarified this point in the Discussion section at line 520 to better reflect this distinction between individual protein markers and the composite transcriptional signature.

While the flow cytometry results showed that all LC patients expressed higher levels of TGF-β1, we showed that it is the LC-CS that segregates the transcriptomic imprint of LC patients.

10. It is also not clear when LC samples were compared to samples from patients with acute disease or those who had recovered, like in Figure 2C, and when they were compared to healthy controls, like in Figure 4G-I. This should be largely consistent throughout the study or specifically stated.

We thank the reviewer for pointing out this important clarification. We apologize for the confusion regarding the comparison groups used across figures.

Figure 4F-I refer to an independent validation cohort, distinct from the single-cell and cytokine profiling cohorts. In this validation cohort, majority of healthy controls were

individuals with a history of SARS-CoV-2 infection but without Long COVID symptoms. Therefore, for these analyses, the comparison allowed us to focus on features specific to persistent symptoms in LC samples rather than general post-infection effects.

In contrast, in **Figure 2C** and related analyses, particularly involving all transcriptome datasets, comparisons were made to either acute or recovered patients and also examined against COVID-naïve healthy control group. This approach was chosen to better delineate the trajectory of immune recovery or persistence of transcriptional alterations following infection.

We have revised the manuscript to clearly state the rationale behind the selection of comparison groups in each section and to ensure consistency in terminology and context. This clarification has been added exclusively to every Results section mentioning comparisons at line 174

samples from patients recovered from their LC symptoms (Recovered) and COVID-naïve healthy controls (HC)

at line247

Among the cytokines measured, CXCL11/ITAC, CCL2/MCP1 and TNF α showed persistently elevated expression in LC samples compared to HCs (COVID- naïve) up to more than 8 months post-infection.

And also, at line 351

Having characterised the LC-CS within the CD14⁺ monocytes, we next evaluated the protein expression of marker genes associated with LC-CS. We performed flow cytometry staining from an independent set of 40 LC patients and 10 HCs (COVID-experienced).

11. In Figure 6J,K the authors analyzed the response of PBMC from patients and recovered donors. They categorized the donors based on the content in CD14 monocytes with predefined features that the authors had termed LC-CS (Long COVID cellular state) into those containing >10% or <10% of LC-CS cells. They then found differences with regards to TGF-beta signaling, WNT-beta signaling and other pathways that were previously identified as defining features of LC-CS. When categorizing samples into those with a high and low content of cells with predetermined features and then finding differences in these exact features it seems like “self-fulfilling”. It would make much more sense to compare functional responses of PBMC from recovered patients to LC / PASC patients and healthy controls.

We thank the reviewer for their keen observation and their valuable suggestion.

Our intention in this section was to explore the functional behaviour of monocytes with LC-CS (Long COVID-associated cellular state) features **in response to an external stimulus** (*Pseudomonas* stimulation), and to gain mechanistic insight into the immune response associated with this cellular phenotype. To this end, we classified donors based on the proportion of CD14⁺ monocytes with LC-CS characteristics (>10% vs <10%), and compared their transcriptional responses in both unstimulated and stimulated conditions. The pathway analysis shown in figure 6K with enrichments of WNT-beta signalling and TGF-beta signalling **observed in the un-stimulated samples, were plotted to merely show comparison of before and after stimulation differences.** Particularly, this analysis highlights the functional response differences post-stimulation, compared to un-stimulated condition.

We acknowledge the reviewer’s concern that comparing predefined LC-CS high vs. low groups for pathways previously used to define LC-CS may introduce circular reasoning. However, our analysis aimed to examine whether the presence of LC-CS cells predicts functional immune alterations **under stimulation**, rather than merely recapitulating marker expression as done in previous analysis (Figure 3 and Extended Data Fig. 5). Of note, **all recovered donors fell into the LC-CS low category**, highlighting a biologically relevant distinction.

Notably, this analysis showed marked downregulation of pathways such as interferon signalling and upregulation of regulatory markers (e.g. *HMOX1*), specifically upon stimulation in the LC-CS high group. Our results in this section show that the dysregulated downstream interferon signalling persists **even in the stimulated monocytes** of the LC-CS high donors as shown in Figure 6K. In contrast, within LC-CS low groups, IFN signalling genes are upregulated in response to stimulation. This suggests that patients with higher proportions of LC-CS, experience high fatigue and may also have a compromised functional immune response to an infection.

As per reviewer’s suggestion, we performed an additional analysis comparing PBMC responses between LC-mild/moderate patient’s samples and recovered donor samples, independent of LC-CS classification. This analysis similarly revealed a downregulated inflammatory, interferon gamma signalling response and downregulation of IL10 signalling **upon stimulation** in the LC group. Additionally, downregulation of IL6 JAK STAT3 signalling, antigen presentation pathways was also found in LC group, while cell cycle-based pathways were upregulated. Please see below:

Figure R17: Pathway enrichment of stimulated samples only, in comparison of LC-mild-to-moderate vs Recovered.

These complementary approaches allowed us to evaluate both the impact of LC-CS cell abundance and the overall group-level functional differences between LC and recovered

individuals. We have now included these new results in the revised manuscript as Extended Data Fig 8E, to show both comparisons at line 448.

We performed joint analysis of stimulated and unstimulated samples using two approaches (Figure 6I). First, we compared the stimulated CD14⁺ monocytes from LC-mild-to-moderate and recovered samples. Pathway analysis of this comparison showed significant downregulation of “Inflammatory response” and “Interferon gamma signalling”, “IL10 signalling”, “cytokine signalling” and “IL6 JAK STAT3 signalling” (Extended Data Fig 8E).

And line 455

All recovered samples were within the low LC-CS donors’ category, along with samples from LC-severe category.

12. Finally, the fact that the authors find similarities between blood monocytes from PASC patients and BAL macrophages from PASC patients is interesting. However, I wonder whether these are actually physiologically linked. The authors have previously argued that LC-CS (or MC4 etc.), with elevated TGF-beta and WNT-signaling are a specific feature of ‘LC-mild/moderate’, hence, LC patients with a mild disease course. These are more likely to experience fatigue rather than dyspnoea and reduced oxygenation, which is more frequently observed after severe COVID-19. This is also evident in their data (Suppl. Table 1). BAL samples from LC/PASC patients were likely taken from patients with respiratory symptoms (‘LC-severe’). So, if the ‘LC-CS’ / MC4 is found primarily in ‘LC-mild/moderate’ how does this connect to profibrotic macrophages and respiratory PASC in ‘LC-severe’ patients?

We thank the reviewer for this important comment, which underlines the complexity and heterogeneity of the Long COVID (PASC) phenotype. In our work, we identified the LC-CS/MC4 monocyte population as pre-dominantly enriched in the peripheral blood of LC patients who experienced a mild/moderate acute COVID-19 disease course. The mild/moderate classification of LC patients is solely based on their acute infection WHO score and does not indicate the LC disease severity. This cellular state was characterized by elevated TGF- β and WNT signalling, among other features. Notably, this signature was also detected in PBMCs (Figure 6B) and BAL myeloid cells (Figure 6F) from an independent, previously published cohort. As the reviewer correctly points out, it is indeed that in the published study BAL samples were collected from hospitalized patients. However, **severity of LC symptoms** is not exclusively dependent on acute COVID-19 disease history, as described below.

In this revision, we introduce a new cohort (cohort 3) (Figure R2, R8 and R14 of this document), which consists of patients from LC-mild-to-moderate COVID-19 disease history but show severe LC symptoms (more information in supplementary table 3). These patients show multiple domains affected, with three patients showing bronchial hyperresponsiveness (BHR), while all patients showed fatigue and dyspnoea. Therefore, we classified the 3 patients with BHR as respiratory-PASC-BHR patients, while others were classified as respiratory-PASC patients. We next performed scRNA-seq analysis of circulating monocytes from these patients. As shown in figure R8 (page 16) and figure R11 (page 20) of this document, all these LC patients showed enrichment of LC-CS signature, while resp-PASC-BHR patients showed highest expression (single cells scoring high AUC scores) compared to respiratory-PASC. This

provides evidence that the LC-CS, although mostly enriched and identified in LC-mild/moderate category is linked to fatigue and dyspnoea. But in patients with severe respiratory PASC, it is also linked with the presence/ emergence of LC-CS. We cannot exclude however that this signature may also be present in BAL from 'LC-mild/moderate', but unfortunately that data is not available.

In the re-analysis of the published dataset (Cohort 5), we included HCs that were also available from the study. We found that cluster 5 of PBMC monocytes—enriched for the LC-CS signature (Figure 6A and 6B) was particularly showing higher levels in patients with respiratory PASC, patients with abnormal lung function test parameters (Figure R12 of this document). The same LC-CS signature was not as highly expressed in non-respiratory PASC or healthy controls. This observation, along with observations from the new cohort suggests a broader relevance of this cellular state in Respiratory-PASC pathophysiology, beyond initial disease severity stratification.

It is possible that this cellular state may also exist in LC-severe individuals, particularly with Respiratory PASC phenotype, in the lung, even if not detected at high frequency in peripheral blood. We speculate that in LC-severe patients, monocytes exhibiting the LC-CS/MC4 signature actively migrate to lungs, where they persist and acquire additional functional programs in response to the local microenvironment.

Consistent with this idea, we observed that the LC-CS transcriptional signature correlates with fatigue scores across all samples, regardless of initial COVID-19 severity (Figure 2F and 3G). Furthermore, mild/moderate LC patients in our cohort also experienced dyspnoea, suggesting that this immune cell phenotype is associated with both systemic symptoms (e.g., fatigue) and possibly mild respiratory impairment.

This also supports the hypothesis that LC-CS-like cells have the potential to acquire a profibrotic phenotype upon entering the lung and encountering inflammatory cues, including T cell–derived signals, as described in the original publication. Previous studies reporting respiratory PASCs have similarly reported the role of monocyte derived profibrotic macrophages in the lungs contributing to pulmonary fibrosis. Taken together, our findings suggest that the LC-CS state may serve as a systemic precursor to profibrotic monocyte/macrophage activity in the lungs, particularly in LC-severe patients. In LC-mild/moderate patients, in contrast, this state may remain confined to the circulation and contribute to fatigue and dyspnoea. Importantly, LC-CS may play a role in patients with Respiratory-PASC, irrespective of their acute COVID-19 disease history.

We have revised the Discussion in the manuscript to reflect these clarifications at line 526

It is intriguing, that while the LC-CS signature was predominantly identified in the circulating monocytes of LC-mild-to-moderate patients, it is also present in the myeloid cell populations in the BAL of LC with severe respiratory PASC. In our dataset, we did not see LC-CS in circulation of LC-severe category, except for one patient. It is therefore tempting to speculate that monocytes with the LC-CS signature remain in the circulation and are not actively recruited in the lungs of LC-mild/moderate patients, leading to fatigue and subtle impairment

in pulmonary function. While they migrate in the lungs of LC-severe patients, where they contribute to the severity of the lung pathophysiology, upon receiving signals from the tissue micro-environment.

13. Overall, the data appears a bit contradictory regarding the described dichotomy between transcriptional features in 'LC-mild/moderate' vs 'LC-severe' and the authors may want to clarify and solidify this a bit more.

We thank the reviewer for highlighting this important point regarding the apparent dichotomy in transcriptional features between 'LC-mild/moderate' and "LC-severe' patients. We appreciate the opportunity to clarify and have revised the manuscript to provide a more cohesive narrative that reflects the complex and evolving nature of LC pathophysiology.

While the LC-CS transcriptional signature is predominantly enriched in PBMCs of LC-mild/moderate patients, we agree that certain elements appeared inconsistent or underexplained. We now propose a refined framework in which:

1. The LC-CS state represents a **circulating monocyte phenotype** linked to fatigue and subtle respiratory dysfunction, particularly prevalent in LC-mild/moderate patients.
2. In LC-severe patients, we hypothesize that LC-CS monocytes **are actively recruited** to lungs and may transition into more pro-fibrotic states upon recruitment to sites such as the lungs—driven by local microenvironmental cues.
3. This model explains the differential detection of LC-CS in PBMCs across patient groups and integrates symptom severity with transcriptional and functional changes

To support and clarify this framework, we have added the following analyses:

- **Temporal signature consistency:** We now include a plot showing genes consistently upregulated in LC-mild/moderate CD14+ monocytes across all time points, highlighting the persistence of the LC-CS signature. We also show the dynamic nature of cellular state development in different patients. (Details can be found on pages 10 - 13 of this document).
- **Cellular Neighbourhood and Trajectory Analyses:** We performed additional comparisons of timepoints T2 and T5 versus Recovered controls (similar to Figure 3E), confirming a sustained enrichment of LC-CS-like monocyte neighbourhood at later timepoints (T5). (Details can be found on page 15 of this document); In addition, we performed a trajectory inference analysis independent of patient stratification. This identified a Long COVID-associated trajectory (Lineage 3) predominantly populated by LC-mild/moderate cells, though one LC-severe patient also showed partial enrichment, highlighting biological overlap. (Details can be found on page 19 of this document)
- **Independent validation in new cohort of LC patients:** We show LC-CS in monocytes of all new LC patients with mild COVID-19 disease history (Cohort 3). Additionally, patients with Respiratory-PASC-BHR showed significantly higher enrichment of this cellular state compared to Respiratory PASC patients (Details can be found on page 16 and 20 of this document).

- **Respiratory PASC association in public data:** In the public dataset (Cohort 5), PBMCs from respiratory PASC patients showed significantly higher LC-CS signature enrichment than both HCs and non-respiratory PASC patients, similar to observations in Respiratory PASC patients from LC-mild/moderate group. (Details can be found on page 20,21 and 35 of this document)
- **Functional evidence:** We directly compared stimulated PBMCs from LC-mild/moderate patients with those from Recovered controls. This revealed a reduced inflammatory and interferon gamma signalling response in LC patients, consistent with a functional altered monocyte state. (Details can be found on page 34 of this document)
- **Clarified Controls & Consistency:** We have clarified and standardized the comparison groups throughout the manuscript using Recovered patients and COVID- experienced HCs where appropriate to remove confounding COVID-19 imprinting and better isolate LC-specific signatures. (Details can be found on pages 32-33 of this document); We have corrected the colour coding in Figure 3G to accurately reflect the patient group. This resolves the inconsistency the reviewer helpfully pointed out. (Details can be found on page 31,32 of this document)
- **Sex-independence of LC-CS:** We have shown that the LC-CS signature is not sex-dependant and is observed in both male and female patients. (Details can be found on page 26-27 of this document)
- **Technical Clarification:** We note that low cell yields at T5 may have limited pseudobulk detection of rare LC-CS cells. Nevertheless, we showed enrichments of LC-CS is also observed at T5 in comparison to Recovered (Details can be found on page 11 and 15 of this document).

Together, these clarifications strengthen the interpretation of LC-CS as a central, dynamic monocyte program in long COVID. This circulating state may evolve differently in tissues in severe cases. This integrated view aligns with the diverse clinical manifestations of LC and resolves the perceived dichotomy.

We have updated the manuscript text accordingly in several sections of both the Results and Discussion to incorporate the revisions, particularly on pages 6,8, 9, 11, 12 and 13 of the manuscript

We would like to thank all the reviewers for their positive assessment and appreciation of our findings and revisions. We also thank the reviewers for their critical review, which helped us improve the manuscript.

We have answered each query in blue, while changes made in the manuscript are shown in purple.

Reviewer #2

(Remarks to the Author)

Kumar et al. have significantly improved their manuscript and answered most of my questions and responded adequately to most previous comments. It still remains a complex study and some inherent weaknesses such as imperfect matching of cases and controls in the cohorts, uneven distribution of samples and cell yields over different conditions and time points remain an important limitation. The authors have made significant efforts to address some of these issues and to mitigate potential effects of confounding. They have also added new information and figures to improve the clarity and transparency of their results and their methods.

However, there are still some issues that require clarification.

1. The study falls short of showing evidence of causality (which I acknowledge is challenging in observational settings) or directionality. I feel that this aspect still requires some more critical discussion.

We thank the reviewer for raising this important point. We agree that, given the observational nature of our study, we cannot infer causality. To clarify this, we have now added the following text to the discussion (line 552):

“Although our findings suggest an association between LC-CS and worsening LC symptoms, they do not directly establish causality. This will be an important next step, which can be addressed through targeted wet-lab experiments and mechanistic studies.”

2. In this regard, given the longitudinal data, do the authors have any evidence regarding the time point of MC-4 appearance and abundance in relation to the onset or resolution of LC symptoms?

We thank the reviewer for this important question. In our cohorts, the MC-4 signature is most consistently observed >3 months post-acute infection, although we also note considerable patient-specific variability.

We present an updated plot for patients with longitudinal samples in Figure R-1. In this figure, the mean gene expression of LC-CS signature genes is calculated across all monocytes for each sample, with the percentage of cells displaying the LC-CS signature within each sample shown above. We also indicate the sampling day for each sample, calculated from the onset of acute COVID-19 disease. In this figure, for example:

- P1 shows a gradual increase in MC-4 abundance and gene expression from 62 days (< 3 months) to 189 days (>7 months), in parallel with worsening fatigue.

- P2 displayed minimal MC-4 expression and minimal percentage of this cellular state at 71 days (< 3 months) with fatigue, which resolved completely by 232 days (> 7.5 months) alongside the disappearance of the fatigue.
- P3 reported extreme fatigue at both available time points, with a persistent MC-4-like cell percentage.
- P4 showed a striking fluctuation in MC4 abundance, but fatigue was reported at both visits.
- P5 also had acute phase data available: ~ 2% of MC4-like cells during acute infection, decreasing to 1% < 3 months post-infection with mild fatigue, but rising sharply to ~ 86% on 107 days (> 3 months) with fatigue and low O2 levels.

Together, these longitudinal observations indicate that MC-4 dynamics vary across individuals, but in many cases, their expansion coincides with persistent (> 3 months) or worsening long COVID symptoms.

Figure R-1. Longitudinal time points of patients with LC-mild-to-moderate, showing variation in the percentage of MC4-like cells and gene expression. The days represent the time since PCR-positive acute COVID infection. Each visit (except the acute one) had a PCR-negative status. The numbers on top of each column indicate the percentage of MC4-like cells from that sample.

We have added this figure showing LC-CS patient heterogeneity as Extended Data Figure 5A and added its reference at line 308:

“Given the well-defined and significant role of monocytes in acute SARS-CoV-2 infection^{34,35}, we next performed differential neighbourhood analysis and compared LC samples to acute and recovered samples at all time points(Fig. 3E and Extended Data Fig. 4G, 5A-5B)”

3. As mentioned, the correlations between MC4 (LC-CS) monocytes and clinical parameters such as Fatigue score or pO2 were weak. In fact, the majority of patients in the cohort had very low (or undetectable?) numbers of MC4 cells, as shown in Figure 3G. In contrast, these data points (samples with 0% MC4) appear to be missing in Figure 3H, where the y-axis is cut just below 10%. These data points should be included in the graph and in the statistical analysis. Even in the current state of Figure 3H / 3G, there seems to be a lot of overlap

between the different categories, and many patients have low or no MC4 monocytes, which poses the question whether MC4 (LC-CS) are indeed causally linked to PASC / Long-COVID or whether this is an epi- or side phenomenon.

We thank the reviewer for their comment. We agree that correlations do not imply causation and have clarified this in the discussion.

Regarding the figures:

- In Figure 3H, **the y-axis does not represent the proportion of MC4 cells** but the range of FAS scores, with 10 as the minimum value.
- Samples were stratified into MC4 high (> 10% of monocytes with an MC4-like signature) and MC4 Low (<10%). We chose this cutoff as a stringent threshold to indicate enrichment and possible systemic dysregulation.
- Samples with 0% MC4 cells were present in all categories. These data points are now explicitly shown in Figure R-2 and Extended Data Fig. 5F, which bins samples by MC4 percentage across the Long COVID and recovered groups.

From this analysis, we note that 5-10% and >10% MC4 cells are observed almost exclusively in LC-mild/moderate cases, whereas 0% MC4 cells occur across all categories (likely due to technical limitations in capturing this minor population). LC-severe and recovered samples were generally within the 0-4% range.

Figure R-2. MC4% binned and proportion of samples from each category.

We agree with the reviewer that these findings do not establish causality. We therefore refer to MC4 as an LC-associated cellular state (LC-CS) that may reflect systemic dysregulation and a possible link to symptom severity. This interpretation is supported by the presence of MC4 signatures in patients with respiratory PASC-BHR (cohort 3) and Respiratory PASC with lung dysfunction (cohort 5). Establishing causal roles and mechanisms will require dedicated experimental studies, which we highlight as an important next step.

We have added the above figure to Extended Data Figure 5F and referenced the figure at line 339:

“Samples with a high proportion of MC4-like cells exhibited significantly higher fatigue score compared to LC patients with low MC4 and to recovered individuals (Fig. 3H and Extended Data Fig. 5F)”

And at line 554:

Additionally, the role of enriched pathways in LC-CS and their drivers across tissues remains to be addressed in future experimental studies.

Overall, the study is impressive in size and complexity and it adds a wealth of interesting data and insights into potential links between altered monocyte states and post-acute sequelae of COVID-19, which is definitely an area of high interest. But it may require more critical discussion of the limitations, specifically with regard to causality and directionality of their findings.

We thank the reviewer for their supportive assessment and constructive feedback. In this revision, we have addressed the limitations more explicitly by

- Clarifying in the discussion that our findings describe associations between LC-CS and long COVID symptoms, without establishing causality or directionality.
- Updating the longitudinal figure to illustrate patient-specific heterogeneity, including the sampling times, along with calculated cell proportions from each sample, to closely examine MC4 dynamics.
- Addition of the distribution of MC4 percentages across patient categories to provide a clearer view of variability.

Reviewer #3

(Remarks to the Author)

I have been asked to consider the authors rebuttals and revisions in response to comments from Reviewers #1 and #2. Broadly, I believe that the authors have done a thorough and well-considered job of responding to both reviewers, including the incorporation of a new cohort and a series of new analyses, all of which have strengthened the main conclusions. There are still gaps, but I believe they are inherent limitations in the samples/data available that do not impact the overall conclusions. I have a few specific comments, following the rebuttal:

Reviewer #1, comment 4, regarding figure 2f - I agree that the figure display statistics should not include the Recovered group. The authors demonstrate in the rebuttal that the interpretation of this data does not change if this group are excluded, so I suggest you change the statistics in 2f to exclude the Recovered group, but leave the blue dots on the plot as a comparator. Amend the legend to reflect this. You could also colour the R and P value text in the figure to red, making it apparent that the statistics refer to only one of the groups on the plot.

We thank the reviewer for this helpful suggestion. Following the advice, we have kept the Recovered group displayed as a comparator but recalculated the statistics without this group. To make this clear, we amended the legend of Figure 2 as follows:

“Statistics calculated using only patient samples are highlighted in red; statistics including all samples (patients and recovered) are shown in blue. P: Patient only, P&R: Patient and Recovered”.

Figure R-3: Updated Figure 2F with statistics for only patients in red and both patients and recovered in blue.

And also re-organised the corresponding text in Results at line 265:

Interestingly, FAS scores showed modest positive correlations with both ‘TGF- β signalling’ (Spearman $R = 0.28, 0.2$; p value = $3.6 \times 10^{-2}, 1.8 \times 10^{-1}$ respectively) and ‘WNT- β catenin signalling’ pathways in $CD14^+$ monocytes (Spearman $R = 0.31, 0.3$; p value = $1.8 \times 10^{-2}, 3.6 \times 10^{-2}$ respectively), calculated on all samples or patient samples alone (Fig. 2F). Similarly, in $CD8^+$ T cells and NK cells, Interferon $\alpha\beta$ induction related pathways show positive correlation with FAS Scores (Spearman $R = 0.44, 0.38, 0.27$ and 0.29 ; p value = $9.7 \times 10^{-4}, 1 \times 10^{-2}, 4.9 \times 10^{-2}$ and 5.8×10^{-2} in CD8 T; Spearman $R = 0.31, 0.24$; p value = 2.3×10^{-2} and 1.1×10^{-1} in NK respectively) (Fig. 2F). $CD8^+$ T cells also showed significant correlation of ‘WNT- β catenin signalling’ with FAS scores (Spearman $R = 0.45, 0.38$; p value = 7.1×10^{-4} and 9.4×10^{-3} respectively). Additionally, in NK cells, TLR signalling cascade pathways were correlated with FAS scores (Spearman $R = 0.29, 0.28, 0.31$ and 0.32 ; p value = $3.2 \times 10^{-2}, 0.7 \times 10^{-2}, 2.4 \times 10^{-2}$ and 3.2×10^{-2}), while ‘TNF α signalling via NF- κ B’ pathway did not show a significant correlation. These correlations show a gradient of association of these pathways across the clinical spectrum.

Reviewer #1, comment 3b and Rev#2 comment 4 are the same, warranting particular attention. On balance, I think that the issue is likely to have arisen from a relative underpowering of the T5 group (being both smaller in patient number and with fewer

monocytes sequenced). This isn't ideal, but it is, I believe, acceptable when viewing the study pragmatically.

We thank the reviewer for their positive assessment of our revisions and the manuscript.

Additional comments:

While not raised by reviewers 1 and 2, I believe there is one important element that requires attention, specifically, the definition used for PASC/long COVID - I am concerned that the participants in this study may not meet many of the common definitions for 'long COVID': 1) The study includes participants with 'long COVID' at less than 3 months post acute disease, but most definitions require 3 months to have elapsed to be certain the 'acute' phase is over. 2) Do your cohorts meet the NASEM criteria for symptoms lasting >3 months? Using appropriate definitions is essential, and I fear that the current data does not indicate whether participants in this study meet these current definitions. This needs to be clarified. Reviewers 1&2 may not have identified this as the new Figure 1 graphics make this most apparent. I believe that including samples from earlier than 3 months post acute disease is still informative, if those participants are known to have maintained disease for at least 3 months (meeting the NASEM criteria). I sympathise with the authors that these definitions are likely to have changed during the course of conducting this analysis, but I view it as essential to the ability to interpret this data that it is clear if/how these cohorts meet these definitions.

We thank the reviewer for raising this important point and for acknowledging the challenges arising from evolving definitions of Long COVID/Post-COVID during the course of this study.

Our cohorts 1-4 were recruited in line with the German S1 Guidelines (as described in the previous revision), which, similar to the NICE guidelines (<https://www.nice.org.uk/guidance/NG188>), define Long COVID as symptoms persisting 4-12 weeks after infection and post-COVID syndrome as symptoms persisting >12 weeks. Since the recruited samples (Cohorts 1 and 2) were up to spring 2022, the **NASEM definition (2024) was not yet available**. Therefore, in this study, we examined both the early (<3months) and later (>3 months) phases of disease, as defined by the German S1 Guidelines.

Importantly, **cohorts 3 and 4 fully meet the NASEM criteria (>3months)**. Additionally, our key finding of LC-CS in Cohort 1 was most prominent beyond 3 months(T3), and was further validated in Cohort 3, demonstrating the robustness across cohorts. For Cohort 5 (up to 3 months), LC-CS was also observed in PBMCs of patients with respiratory PASC, suggesting relevance across both early and later phases.

To directly address the concern regarding <3-month samples (T2):

- In Cohort 1, >70% of patients with T2 samples showed persistent symptoms beyond 3 months, thereby meeting the NASEM criteria. Re-analysis excluding the few patients who did not progress showed minimal differences in monocyte expression profiles (Fig. R-4, analogous to Main Figure 2B).

- Similarly, in Cohort 2 at T2, >70% of patients developed post-COVID syndrome, and excluding the others did not alter cytokine findings.

Together, these analyses demonstrate that our results remain robust even when restricted to patients fulfilling the NASEM definition.

Figure R-4: Monocytes gene expression profiles as shown in Main Figure 2B.

Figure R-5: Significant cytokines after excluding the patients from T2, who did not develop post-COVID syndrome.

We have clarified this in the manuscript

- **Results (line 147):** “Patients were recruited in accordance with German S1 guidelines and Delphi consensus criteria of long COVID (4-12 weeks) and post-COVID syndrome (>12 weeks)^{21,22}.”
- **Discussion (line 484):** “Particularly, our study examined both the early (i.e. < 3 months) and later time points (> 3 months). While recruitment predated NASEM definition, more than 70% of patients from the early time point (i.e., < 3 months) showed persistent symptoms, well over 3 months, thereby fulfilling the NASEM criteria for symptoms lasting >=3 months.”
- **Methods (line 575):** “EDTA blood was collected from patients with Long COVID (>4 weeks) and post-COVID-19 symptoms (>12 weeks) in accordance with German S1 guideline for Long COVID and Post-COVID syndrome at Hannover Medical school (MHH) or the Siloah hospital, Hannover, between March 2020 and April 2022.”

In summary, I believe that the article makes a series of important discoveries that will be both valuable additions to the literature on long COVID, and potentially be of interest to the wider immunology field as an example of the immunopathology evident in a sequelae of respiratory viral infection. I find the concept of underpinning monocyte dysregulation that manifests in different tissues, owing to trafficking to different sites (i.e. fatigue versus cardiorespiratory symptoms), particularly interesting and fitting with other literature. This gives me confidence that the manuscript would be a welcome and valuable addition to the field.

We thank the reviewer for their encouraging and positive assessment of our findings.

Reviewer #4

(Remarks to the Author)

Kumar and colleagues applied snRNA, snATAC-seq, and targeted proteomics to PBMC and plasma samples of two discovery cohorts and scRNA-seq to PBMC and bronchoalveolar lavage fluid samples of three separate validation cohorts to assess different factors and cellular states correlating with active long COVID (LC) in patients suffering from fatigue and respiratory symptoms following COVID-19. A subset of the discovery cohort samples was stimulated using *P. aeruginosa*, followed by single-cell analysis. They found a LC cellular state (LC-CS) in circulating CD14⁺ monocytes and increased TNF, CCL2 and CXCL11 concentrations in patients with active LC who had previously experienced mild-to-moderate acute COVID-19. In two independent LC cohorts of patients with respiratory symptoms, this LC-CS was enriched in blood CD14⁺ monocytes and also found in myeloid cells isolated from bronchoalveolar lavage fluid. Monocytes of individuals with high percentages of LC-CS CD14⁺ monocytes showed an abnormal response to in vitro stimulation with *P. aeruginosa*. Overall, this is an interesting study characterising monocytes in LC patients. The authors are experts in multiome analysis of immune cells. The epigenetic insight into monocytes and immune cells of LC patients is particularly noteworthy.

We thank the reviewer for their positive assessment of our manuscript.

Major points

(1) The authors characterised different LC-CS in monocytes. The LC-CS in MC4 was very different between months 3 to 9 to that of the other monocyte subsets (Fig. 3E). Are these inter-monocyte differences driven mainly by the pathways shown in Fig. 3F, i.e. TNF, WNT-beta-catenin, and TGF-beta signaling or by other factors?

We thank the reviewer for this insightful question. To clarify, **we refer specifically to the MC4 cluster as the LC-CS in monocytes**. This designation is supported by the significant expansion of MC4 neighbourhoods, along with the significantly higher AUC scores for TGF-beta signalling and WNT-beta-catenin signalling in this cluster.

From our trajectory analysis and flow cytometry analysis, as shown in Extended Data Figure 5E and Main Figure 4E, respectively, we observe a downregulation of TNF signalling coupled with increasing TGF- β and WNT- β -catenin activity as cells adopt an MC4 program. While these pathways appear central, the precise mechanisms driving MC4 enrichment remain unsolved and will require future experimental validations.

We have added the following text to the discussion (line 554):

“Additionally, the role of enriched pathways in LC-CS and their drivers across tissues remains to be addressed in future studies.”

(2) Cytokines and chemokines are very tightly regulated. By using the Quanterix HD SP-X Imaging and Analysis System together with a cytokine and a separate chemokine panel, the authors describe increased concentrations of TNF, CCL2 and CXCL11 in patients with active LC who had previously experienced mild-to-moderate acute COVID-19. Did the authors verify their measurements using another assay?

We thank the reviewer for this important point. While we did not measure these cytokines using an additional assay in our cohort, we note that TNF, CCL2 and CXCL11 were **significant across LC categories**, with TNF correlations to O2 particularly evident in LC-mild-to-moderate cases.

Importantly, several independent studies from different cohorts report consistent findings. For example, TNF elevation was confirmed in a study of >8,000 patients (PMID: 35732153) and in > 100 patients (PMID: 35701186). The latter also reported CCL2, which was further validated in another cohort of >150 samples (PMID: 36458566). These studies are from independent cohorts and cover different long COVID phenotypes, collected and examined in different labs from different countries.

To directly address the reviewer’s concern of measurements using a different assay, we also found studies where OLINK measurements were performed in samples from Long COVID patients (PMID: 35929818) and showed a significant increase in TNF, CXCL11 and CCL2. Similarly, another publication with the OLINK panel also reported TNF-driven inflammation and showed an increase in TNF and CXCL11 (CCL2 was not included in the panel) (PMID: 37296110).

These converging results across independent cohorts and measured using at least two different technologies strengthen the robustness of our findings. We have added the following clarification at line 490:

“Our findings revealed sustained elevation of pro-inflammatory mediators, including TNF α , CCL2 and CXCL11 in the plasma of LC patients, up to 9 months post-infection, consistent with reports from independent cohorts measured using different assays^{8,56-59}.”

(3) The curves in Fig. 3G for MC4 seem to correlate with some outliers rather than with the majority of data points. How would the analyses look like if the most extreme outliers were excluded? How did the authors determine high vs. low MC4 samples? Would the percentage of high vs. low MC4 samples still correlate with the FAS score, as shown in Fig. 3H, if the authors left out the apparent outliers?

We thank the reviewer for this important point. To address it, we note that

- We used Spearman correlations, which are less sensitive to outliers. Importantly, the samples with very high MC4 percentages are not technical outliers but come from **different patients, sampled at different timepoints and from different experimental runs**, reflecting biological variability.
- As shown in Figure R1 (longitudinal) and Figure R2 (distribution by category), these high percentage MC4 samples are predominantly from LC mild-to-moderate patients, whereas LC-severe and recovered samples all fall mostly within < 0-5% range.
- We defined MC4-high as >10% of CD14⁺ monocytes with an MC4-like expression profile, a cutoff chosen to reflect enrichment and systemic dysregulation.
- To test robustness, we repeated the analyses excluding samples with > 75% MC4-like cells (5 samples, each from a different patient). The overall trend remained in the same direction, though correlations became statistically weaker, as expected with reduced sample size. Importantly, categorical comparisons of MC4-high vs. MC4-low still showed significant differences in FAS scores after excluding these samples.

Figure R-6: Sensitivity test of correlations, by excluding the 5 samples with >75% MC4-like cells (left) and categorical analysis, by removing the 5 samples (right).

Together, these findings indicate that the observed MC4-high cases represent true biological variability rather than statistical outliers. Even after removing the most extreme cases, the association between MC4 proportion and fatigue remains evident, supporting the robustness of our conclusion.

(4) The epigenetic analysis of the identified different monocyte subsets in Fig. 5 is interesting. Based on the landscapes shown of IER3 and LMNA, there seem to be rather minimal differences between MC4 and the other monocyte subsets. Yet, the signal counts appear to be significantly different. Could the authors make this connection between landscapes and calculated signal counts more intuitive to appreciate?

We thank the reviewer for this helpful comment. To make the connection between chromatin landscape and RNA levels more intuitive, we generated coverage plots for the highlighted peaks for IER3 and LMNA in Figure 5F-5G. These plots illustrate the differences in accessibility at specific regions that underlie the observed signal count differences.

IER3 peaks

LMNA peaks

Figure R-7: Coverage plots of significant peaks linked to expression of the gene across different cell subsets.

We have added these figures to Extended Data Figure 6C and referenced them in the manuscript at line 405.

“This allowed us to pinpoint TFs with binding sites in the open chromatin regions that regulate the expression of the corresponding linked peaks (Fig. 5F-5G, Extended Data Fig. 6C and Supplementary Table 6).”

(5) I understand the monocyte clusters identified in BAL myeloid cells and shown in Fig. 6 are different from the ones described in the previous figures. The BAL monocyte cluster C1 appears to correlate most with the PBMC monocyte cluster MC4, and C1 appears to show a profibrotic gene signature. Is this profibrotic signature mainly based on TGF-beta? If yes, this might not be a solid signature molecule indicating a profibrotic state, as depending on its cytokine microenvironment, TGF-beta can promote type 3 inflammation or immune regulation, both without causing fibrosis. Does MC4 also show a profibrotic gene signature?

We thank the reviewer for this insightful comment. Based on the published work by Li et al. (PMID: 39018367) and the datasets analysed here, the likely cytokine driver of fibrosis in the lung microenvironment is IFN- γ rather than TGF- β . Consistent with this, our analysis of TGF-beta signalling AUC scores did not reveal striking differences that distinguished the BAL C1 cluster from the other myeloid clusters (Figure R-8 (left)). Furthermore, comparison of the TGF-beta signalling gene set with the pro-fibrotic signature defined by Wendisch et al. (Cell, 2021) showed no gene overlap.

These findings suggest that the pro-fibrotic phenotype in BAL is tissue-specific and likely shaped by the local cytokine milieu (e.g. IFN- γ). As shown in Extended Data Figure 7E, a figure analogous to Wendisch et al (Cell, 2021), C1 exhibited enrichment of the pro-fibrotic signature. In our cohort 1, circulatory monocytes, MC4 does not show a strikingly different pattern than other monocyte subclusters (Figure R-8 (right)). To examine this more broadly, we compared all significantly expressed genes in MC4 ($p_{adj} < 0.05$, $> 2k$ genes) with the pro-fibrotic signature (>200 genes) and found only $\sim 1.3\%$ overlap (~ 40 genes). Instead, MC4 was characterised by the upregulated expression and open chromatin accessibility of pro-angiogenic and adhesion markers, such as *VEGFA*, *ENG*, *TGFB1*, *RXRA*, *ICAM1*, and *ITGA5* (Figure 5C).

Together, these results indicate that the C1 cluster represents a pro-fibrotic program acquired under tissue-specific conditions, whereas circulating MC4 do not show this program. Other cytokine and cellular interaction cues within the tissue micro-environment may also contribute to shaping the fibrotic phenotype of C1 cluster cells, and we emphasise that functional validation will be required to confirm their roles.

Figure R-8: AUC score of TGF-beta signalling in all clusters as shown in Figure 6D (left). AUC score within circulating monocyte subclusters of cohort 1 of pro-fibrotic signature collected from Wendisch et al. (Cell, 2021) (right).

We have adjusted the discussion to remove any speculation of TGF- β -driven fibrosis from line 524 and replaced it with the following text.

The activity of these pathways in CD14⁺ monocytes was significantly correlated with the FAS score, suggesting their link with the pathology of LC.”

Minor points

(a) In the legend to figure 5, Asterix should be an asterisk.

Thank you for the correction. This has been corrected.

(b) In Fig. 6I, it should read *P. aeruginosa* instead of *P. auerginosa*.

Thank you for the correction. This has been corrected.

Reviewer #2 (Remarks to the Author):

The authors have addressed my previous comments through additional analyses and textual revisions, with one exception. I believe it remains important to acknowledge, both in the Results and Discussion sections, that the observed associations between LC-CS (MC4) and O₂-saturation or fatigue scores are relatively weak, and that MC4 was undetectable in a substantial proportion of samples. This clearly suggests that additional or alternative mechanisms contribute to PASC. While the authors acknowledge this point in their response to my comments, it is not clearly reflected in the manuscript itself. Explicitly stating this limitation, would not detract from the careful and comprehensive characterization of monocyte states in PASC presented in this well-conducted study. Rather, it would strengthen the critical discussion of the results. Overall, the manuscript and the accompanying data represent a valuable contribution to the growing body of work on potential immune mechanisms underlying PASC.

We thank the reviewer for the thoughtful comments and apologise for not having stated the rationale as clearly as expected. We have now clarified this point in the Results section at line 321

We found a modest but statistically significant positive correlation between MC4 proportions and the FAS score (Spearman's $R = 0.31$, p value = 2.0×10^{-2}) and a negative correlation with pO₂ (Spearman's $R = -0.28$, p value = 3.1×10^{-2}). In contrast, the MC1-like cellular state showed a statistically significant negative correlation with the FAS score (Spearman's $R = -0.27$, p value = 4.0×10^{-2}) (Fig. 3G). The modest strength of these correlations likely reflects underlying patient heterogeneity (Extended Data Fig. 5F).

And in the discussion at line 530

The proportions of LC-CS also showed modest correlations with the FAS-Score and blood pO₂ levels, suggesting a potential mechanistic involvement of this cellular state in the pathophysiology of LC.

And at line 553

Although our findings suggest an association between LC-CS and the severity of LC symptoms, these correlations are relatively modest and therefore do not establish causality or rule out alternative mechanisms contributing to symptom development.